# New insights into 2021 La Palma eruption degassing processes from direct-sun spectroscopic measurements

Noémie Taquet[1,2,3], Thomas Boulesteix[2], Omaira García[1], Robin Campion[4], Wolfgang Stremme[5], Sergio Rodríguez[6], Jessica López-Darias[6], Carlos Marrero[1], Diego González-García[7,8], Andreas Klügel[9], Frank Hase[10], M. Isabel García[6], Ramón Ramos[1], Pedro Rivas-Soriano[1], Sergio Léon-Luis[1,3,11], Virgilio Carreño[1], Antonio Alcántara[1], Eliezer Sépulveda[1,3], Celia Milford[1], Pablo González-Sicilia[1,3], Carlos Torres[1]

[1] Izaña Atmospheric Research Center (IARC), State Meteorological Agency of Spain (AEMET), Tenerife, Spain
[2] Consejo Superior de Investigaciones Científicas, Volcanology Research Group, IPNA-CSIC, Tenerife, Canary Islands, Spain
[3] TRAGSATEC, Madrid, Spain
[4] Universidad Nacional Autónoma de México, Instituto de Geofísica, Mexico City, Mexico
[5] Universidad Nacional Autónoma de México, Instituto de Ciencias de la Atmósfera y Cambio Climático, Mexico City, Mexico
[6] Consejo Superior de Investigaciones Científicas, Group of Atmosphere, Aerosols and Climate, IPNA CSIC, Tenerife, Canary Islands, Spain
[7] Institute of Earth System Sciences (Section of Mineralogy), Leibniz University of Hannover, Hannover, Germany
[8] Department of Mineralogy and Petrology, Universidad Complutense de Madrid, Madrid, Spain
[9] Department of Geosciences, University of Bremen, Bremen, Germany
[10] Institute for Meteorology and Climate Research, Karlsruhe Institute of Technology, Karlsruhe, Germany
[11] Departamento de Física, Universidad de La Laguna, San Cristóbal de La Laguna, Santa Cruz de Tenerife, Spain

*Correspondence to*: Noemie Taquet (noemi.taquet@gmail.com) and Omaira Garcia (ogarciar@aemet.es)

**Abstract**. In a world increasingly impacted by climate change and natural hazards, atmospheric monitoring networks are essential for informed decision-making. During the 2021 La Palma eruption, we combined existing and rapidly deployed instruments to monitor volcanic gas emissions up to 140 km from the source. We used direct-sun measurements from low- (EM27/SUN) and high- (IFS-125HR) resolution Fourier Transform InfraRed (FTIR) spectrometers contributing to key atmospheric global networks. In La Palma, the EM27/SUN was combined with a Differential Optical Absorption Spectroscopy (DOAS) instrument. We present new FTIR retrieval methods to derive the $SO_2$, $CO_2$, CO, HF, HCl relative abundance in the plume from both low- and high- resolution solar absorption spectra. Using Sentinel-5P TROPOMI data, we derived $SO_2$ fluxes and estimated total emissions of $1.8 \pm 0.2$ Mt $SO_2$, $19.4 \pm 1.8$ Mt $CO_2$, $0.123 \pm 0.005$ Mt CO, $0.05 \pm 0.01$ Mt HCl, and $0.013 \pm 0.002$ Mt HF over the course of the eruption. These results are consistent with mass balance derived from petrologic degassing estimates. This study demonstrates that high- and low-resolution FTIR and DOAS spectrometers, integrated within global monitoring networks, can provide quantitative constraints on volcanic gas composition and fluxes over large distances. Such capabilities are directly applicable to volcanic crisis monitoring, complement satellite observations, and support improved assessments of volcanic impacts on atmospheric composition at regional scales.

## 1. Introduction

Volcanic emissions of greenhouse gases and pollutants remain poorly constrained due to the limited number of volcanoes well monitored for gas emissions. The present knowledge relies on either short-term records at permanent stations or on discrete campaigns of measurements, mostly during eruptive crises. Characterizing volcanic degassing processes is essential to improve our understanding of the multi-species volcanic gas

emissions across various geodynamic settings and their long- and short-term impact on the atmospheric
composition.

The abundance and composition of dissolved volatiles control the buoyancy and viscosity of magmas,
making them a primary driver of eruptive dynamism and duration (Longpré et al., 2025). Water ($H_2O$) and
carbon dioxide ($CO_2$) are the most abundant species in volcanic degassing, followed by sulfur dioxide ($SO_2$) and
halogen-derived species (mainly halides). They show different solubility in magma, which depends mainly on
pressure, temperature and redox conditions (Gennaro et al., 2020; Cassidy et al., 2022). $CO_2$ and $H_2O$ are
usually among the deepest exsolved gas species, followed by $SO_2$ and halogens in sub-surface. Therefore, the
exploration of their pre- and co-eruptive relative abundance can reveal critical information on pressurisation of
the magma plumbing system, as well as on ascent rates and volatile exsolution pathways (Voigt et al., 2014;
Taquet et al., 2019). The temporal evolution of the $\Delta CO_2/SO_2$ ratio and halogen-derived species-to-$SO_2$ ratios in
volcanic plumes have often been used to infer the respective contribution of deep to shallow magmatic
processes in the transitions in eruptive dynamism such as changes in the bubble contents in the magma chamber,
replenishment, magma batches mixing or fractional crystallisation (Harris and Rose, 1996; Shinohara et al.,
2003, 2008; Werner et al., 2012; La Spina et al., 2015). Volcanic plume compositions, when combined with
seismic and structural data, help constrain volatile fluxes, magma ascent rates, and the architecture of the
magmatic plumbing system. Integrating gas measurements with petrological constraints from matrix, melt
inclusions (MI), and fluid inclusions (FI) enables reconstruction of pre-eruptive volatile contents and degassing
pathways, which are key to modeling eruption dynamics (e.g.: Ubide et al., 2023; Longpré et al., 2025).

The 2021 Cumbre Vieja (La Palma) fissure eruption (from 19 September to 14 December 2021, VEI
3), called Tajogaite, was the first subaerial eruption in 50 years in the Canary Islands archipelago and thus the
first opportunity to directly assess the amount and composition of volcanic degassing during an eruption in
Canary Islands (Burton et al., 2023). It was preceded by up to 12 low intensity seismic swarms between October
2017 and September 2021, occurring at depths between 20 and 30 km, without evidence of surface deformation
(Torres-Gonzalez et al., 2020; Mezcua and Rueda, 2023). Some of these seismic swarms were accompanied by
changes in flux or composition of trace gases ($CO_2$, He, Rn) in soil or at the Dos Aguas cold spring located in
the Caldera de Taburiente to the north (Torres-Gonzalez et al., 2020; Padrón et al., 2022). These observations
were interpreted as evidence of magma migration from a deeper upper mantle reservoir to a shallower sub-
crustal reservoir (Padrón et al., 2022). On 11 September 2021, a new seismic swarm occurred at ~10 km depth
and intensified over the following days, accompanied by ground inflation reaching 30 cm (De Luca et al., 2022).
Subsequently, the seismicity migrated towards the surface and the Tajogaite eruption started on 19 September
2021. Several craters opened and grew along a NW-SE eruptive fracture (Muñoz et al., 2022) on the western
flank of the Cumbre Vieja Ridge (CVR). The eruption exhibited simultaneously multiple eruptive styles at
various summital and flank vents, including more than 100 m-high hawaiian lava fountains, strombolian
spattering activity, ash venting, vulcanian explosions and significant effusive activity. Over the 85 days of its
activity, it produced a $\sim 1.8 \times 10^8$ m$^3$ lava flow field (Civico et al., 2022) covering an area of 12 km$^2$, and a
tephra blanket with a total estimated volume of $\sim 2.3 \times 10^7$ m$^3$ (Bonadonna et al., 2022), provoking the
evacuation of several thousands of people and the destruction of ~3000 buildings (Copernicus EMSR546,
PEVOLCA reports). During the course of the eruption, volcanic gases were injected between 1000 and 6000 m
a.s.l. (Bonadonna et al., 2022; Milford et al., 2023; Hedelt et al., 2025) and were transported over North Africa
and Europe, as well as across the Atlantic to the Caribbean on several occasions (Hedelt et al., 2025). Total $SO_2$
emissions were estimated to be about 1.84 Mt (Milford et al., 2023) using the daily mass estimates derived from
the TROPOspheric Monitoring Instrument (TROPOMI) measurements and provided by MOUNTS-Project
(mounts, Valade et al., 2019).

Geophysical and geochemical co-eruptive observations revealed insights into the structure of the
plumbing system (d'Auria et al., 2022; Dayton et al. 2023) and melt evolution during the eruption (Day et al.,
2022; Ubide et al., 2023; Dayton et al., 2024; Longpré et al., 2025). Co-eruptive seismicity defines to clusters
(d'Auria et al., 2022; Del Fresno et al., 2023), the shallowest one ranging between 5 and 15 km depth starting on
26 September 2021 and remaining until the end of the eruptive period, and the deeper ranging between 20 and
25 km depth occurring from 1 October to 13 December 2021. Additionally, a temporal progression in the melt

chemical composition was observed: the initial erupted magma exhibited a tephritic composition (MgO ~6 wt% and TiO$_2$ ~4 wt%) and was gradually (< day 20; Day et al. 2022) replaced with a basanitic magma (MgO ~8 wt% and TiO$_2$ 3.7 wt%) for the rest of the eruption (Day et al., 2022; Ubide et al., 2023). This type of transition reflects a behavior similar to that previously documented for the 1949 and 1971 Cumbre Vieja eruptions (Klügel et al., 2000), and was interpreted as mixing between a resident mush and deep fresh basaltic magmas in the shallow reservoir. Such changes in magma composition could contribute to changes in eruptive dynamism and might be reflected in surface gas composition changes. In fact, variability in the eruptive dynamism was observed through seismic and deformation monitoring (Del Fresno et al., 2023; Charco et al., 2023), tephra analysis and geochemical lava and ash studies (Bonadonna et al., 2022; 2023; Birnbaum et al., 2023; Longpré et al., 2025). In the early phase of the Tajogaite eruption, rapid cone growth and vent openings were accompanied with explosive tephra ejections. On 25 September, a significant cone collapse was accompanied with increasing explosive activity with evidence of white xeno-pumice fragments in tephra (Day et al., 2022; Romero et al., 2022). Lava became more fluid after the transition from tephritic to basanitic composition. By late October-November, plume height stabilized at 2500–3500 m a.s.l. (Córdoba-Jabonero et al., 2023), with lower SO$_2$ emissions (Milford et al., 2023). The final weeks saw intense activity, collapses, structural changes and vents reconfiguration (Gonzalez, 2022; Walter et al., 2023).

To date, only a few studies have reported the composition of the gas plume measured during the Tajogaite eruption, and none have provided a multi-species time series of estimated emission fluxes over the entire eruptive period. Ericksen et al. (2024) derived CO$_2$ volcanic emission fluxes from drone-borne SBA-5 infrared CO$_2$ sensors measurements and also measured $\Delta CO_2/SO_2$ ratios using ground-based Multi-GAS instruments localized near the vent. Burton et al. (2023) reported the first time series of the $\Delta CO_2/SO_2$ ratio of the gas plume, employing ground-based FTIR spectrometry techniques using incandescent ash plumes, lava fountaining and lava flow as thermal sources, and occasional solar absorption measurements. They also reported drone-borne and ground-based MultiGAS in-plume measurements. Recently, Asensio-Ramos et al. (2025) reported the first time series of $\Delta CO_2/SO_2$, SO$_2$/HCl and $\Delta CO/\Delta CO_2$ ratios measured at the base of the eruptive column using open-path FTIR measurements with lava fountaining and lava flows as thermal source. Using the surface gas measurements, petrological data and estimates of lava emission rates, these authors reveal evidence of exceptional CO$_2$-rich gas emission with respect to the emitted lava volume during the eruption. Recent studies showed the presence of particularly SO$_2$- and CO$_2$-rich compositions of deeply entrapped-MI in volcanic rocks from the Canary Islands (Longpré et al., 2017; Taracsak et al., 2019), which may be linked to mantle metasomatism (Hansteen et al., 1991; 1998).

This study presents a comprehensive time series of $\Delta CO_2$, $\Delta CO$, HCl, and HF to SO$_2$ molar ratios measured in the Tajogaite volcanic plume between 21 September and 14 December 2021, spanning the full duration of the eruption. The measurements were conducted at distances of 15 km and 140 km from the vent using ground-based direct-sun FTIR and DOAS instruments, integrated into global atmospheric monitoring networks. Ground-based FTIR and UV direct-sun methods provide multi-species and time-resolved total column measurements of the main volcanic gases, regardless of the plume altitude, while ensuring operators and instruments safety (Butz et al., 2017; Taquet et al., 2023). They have the advantage of using the sun as a common and both homogeneous and constant-intensity source (at the timescale of a single measurement), providing solar spectra in a wide spectral range and with a high signal-to-noise ratio. We took advantage of the instrumentation installed at the Izaña Atmospheric Observatory (IZO) in Tenerife. Its high altitude and geographical location were ideal for repeatedly directly capturing the volcanic plume including in situ surface measurements, thereby enhancing the temporal density of our dataset. We estimated daily SO$_2$ volcanic emission fluxes from space-based TROPOMI/Sentinel-5P measurements, and used the measured species-to-SO$_2$ ratio to derive the emission fluxes of the other volcanic species and their total emissions. Our results are interpreted in the light of petrological (including new melt inclusions and matrix glass compositions presented in this study) and geophysical data taken from the literature.

## 2. Gas and particulate matter measurement sites and instrumentation

A comprehensive network for the monitoring of trace gases, aerosols and ash fallout was operative for
air quality monitoring and scientific research during the eruption. Monitoring efforts relied on a combination of
permanent stations which are part of international atmospheric research and air quality monitoring networks,
such as those in the Canary Islands Government Air Quality Monitoring Network (AQMN) and the facilities at
IZO, as well as additional equipments specifically installed for attending the volcanic emergency. In this
framework, the State Meteorological Agency of Spain (AEMET), through the Izaña Atmospheric Research
Center (IARC) and the Territorial Delegation of AEMET in the Canary Islands (DTCAN) and in collaboration
with the Spanish National Research Council (CSIC) and other institutions, deployed scientific instrumentation
on La Palma. The objectives of the deployment were: 1) real-time monitoring and characterization of the
vertical structure of the eruptive plume, carried out through the implementation of an aerosol profiling network
in the context of the European Aerosol, Clouds and Trace Gases Research Infrastructure (e.g.: ACTRIS, 2021;
Barreto et al., 2022; Álvarez et al., 2023); 2) complementing the air quality network observations managed by
the Government of the Canary Islands (Milford et al., 2023, and references therein); and 3) investigating the
physicochemical composition of the volcanic plume, its links with the evolution of the eruptive process and
studying the ash-gas-aerosol interactions (e.g.: Garcia et al., 2022; Cordoba-Jabonero et al. 2023; Cuevas et al.,
2024, and references therein).
We conducted remote sensing and surface gas and ash measurements during the entire eruptive period
at two stations localised in La Palma (FUE) and Tenerife (IZO) islands (Fig. 1, upper panel) to assess the co-
and post-eruptive compositional variability of the Tajogaite volcanic plume. In addition, aerosol and surface
$SO_2$ measurements were conducted at two locations on La Palma (Los Llanos, El Paso) and Tenerife (IZO).
Mobile in-situ plume measurements in La Palma using MultiGas were also performed during episodes of plume
grounding driven by favorable meteorological conditions. Figure 1 displays a map of the FUE and IZO stations,
as well the MultiGas, aerosols and $SO_2$ measurement sites in La Palma concurrently with a typical $SO_2$ plume
as detected by space-based TROPOMI/Sentinel-5P sensor. The instruments at each site and the measurement
periods are summarised in Table 1 and detailed below.

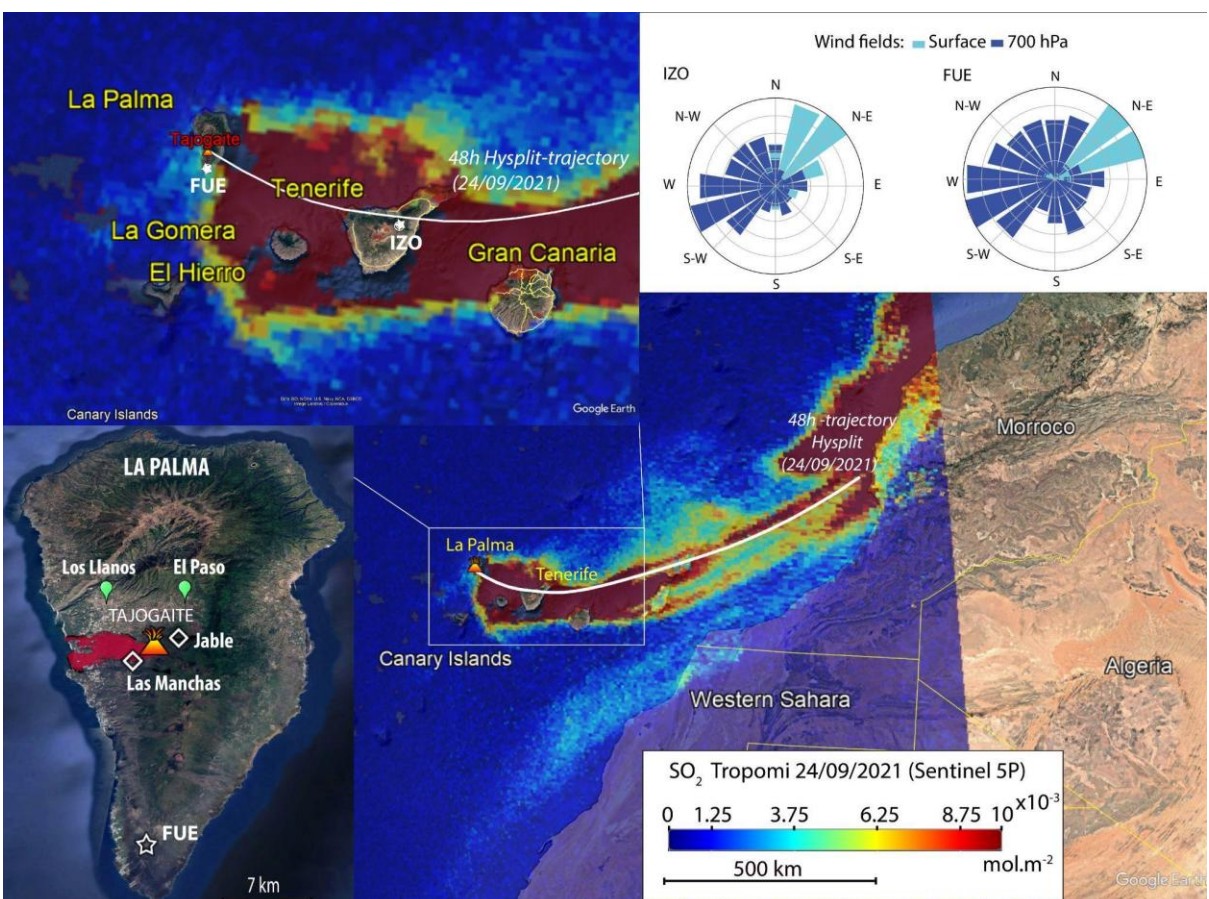


**Figure 1: Location of our measurement stations in Canary Islands during the 2021 La Palma eruption (FUE and IZO**
**represent the Fuencaliente and Izaña stations, respectively, marked by white stars). SO$_2$ data from**
**TROPOMI/Sentinel-5P sensor are shown in the map for 24 September 2021, illustrating the typical plume dispersion**
**over hundreds of kilometres. The instruments implemented at FUE and IZO stations are summarised in Table 1.**
**Wind rose diagrams for surface and 700 hPa levels (corresponding to the average of the plume altitude during the**
**eruption) are also presented for the IZO and FUE stations (upper right panel), considering the entire eruptive period**
**and the ECMWF Reanalysis v5 (ERA5) data (ECMWF: https://www.ecmwf.int/en/forecasts/dataset/ecmwf-**
**reanalysis-v5). The base layer was sourced from Google Earth (© Google), while the SO$_2$ distribution map was**
**derived from TROPOMI data accessed through the Sentinel Hub platform. The upper left panel presents a zoom on**
**La Palma Island including all of the surface (white hollow diamonds for MultiGAS), aerosols (green) and columns**
**gas (white stars) measurement sites from which data is used for this study. The Tajogaite eruption lava flow field (red**
**shaded area) was taken from the European Environment Agency Copernicus Emergency Management Service**
**(https://emergency.copernicus.eu/mapping/list-of-components/EMSR546).**
**Table 1: Solar FTIR - DOAS and surface gas and particulate matter in situ measurements conducted at the FUE,**
**IZO and La Palma stations from 21/09/2021 to 21/01/2022. Details on aerosols in situ measurements are given in**
**Rodriguez et al. (submitted).**

| Station (Island) (geographical coordinates) altitude distance from the eruptive fissure | Instrument (Networks) | Measurement period | Fraction of measurement days capturing the volcanic plume (or post eruptive diffuse emissions) |
|---|---|---|---|
| FUE (La Palma) (28.49ºN,17.85ºW) 630 m a.s.l. ~ 15 km | EM27/SUN#SN143 (COCCON) | 25/09/2021 - 21/01/2022 | 21/59 (co-eruptive) 1/11 (post-eruptive) |
| | Combined EM27/SUN#SN143-DOAS | 10/10/2021 - 10/12/2021 | 14/32 (co-eruptive) |
| IZO (Tenerife) (28.31ºN, 16.50ºW) 2373 m a.s.l. ~ 140 km | EM27/SUN#SN085 (COCCON) | 20/09/2021 - 31/01/2022 | 4/38 (co-eruptive) 0/9 (post-eruptive) |
| | IFS-125HR (NDACC) | 19/09/2021 - 31/01/2022 | 11/48 (co-eruptive) 0/13 (post eruptive) |
| | In situ UV fluorescence analyzers (SO$_2$) (GAW WMO network) | 21/09/2021 - 31/12/2021 | 26/83 (co-eruptive) 1/16 (post eruptive) |
| | In situ Picarro (CO$_2$, CO) (GAW WMO network) | 19/09/2021 - 31/12/2021 | 26/85 (co-eruptive) 1/16 (post eruptive) |
| | Aerosol samplers | 19/09/2021 - 31/12/2021 | 26/85 (co-eruptive) |
| El Paso (La Palma) (28.6590ºN, 17.8481ºW) 860 m.a.s.l. | Aerosol samplers | 27/09/2021- 19/10/2021 | 18/22 (co-eruptive) |
| Los Llanos (La Palma) (28.6586ºN, 17.913100ºW 343 m.a.s.l. | Aerosol samplers | 20/10/2021- 07/01/2022 | 52/55 (co-eruptive) |

## 2.1. The Fuencaliente station (FUE, La Palma Island)
In the context of AEMET responsibilities, as a State Agency, for continuous monitoring of the
meteorological and climatic conditions and of atmospheric composition, a specific instrumental deployment has
been set up in La Palma. In particular, a new station for gas and particle monitoring was implemented at the San
Antonio Volcano visitors center of Fuencaliente, at the southern tip of La Palma Island, ~15 km from the
eruptive fissure of the Tajogaite volcano (Fig. 1). The FUE station included a wide range of instruments such as
a sun-lunar Cimel CE318T photometer, contributing to the Aerosol Robotic Network (AERONET), for aerosol
column measurements, a Lufft CHM15k ceilometer for aerosol and cloud vertical profiling and an all-sky
camera for weather monitoring (Román et al., 2021) and a tephra trap.

A few days after the beginning of the eruption (on 25 September 2021), we deployed an EM27/SUN
spectrometer (developed by the Karlsruhe Institute of Technology (KIT), in collaboration with Bruker Optics,
Germany), which is the standard instrument of the Collaborative Carbon Column Observing Network
(COCCON, Frey et al., 2019) dedicated to the measurement of greenhouse gases. This portable Fourier
Transform Infrared (FTIR) spectrometer, equipped with a Quartz beamsplitter and two InGaAs photodetectors,
provides low-spectral resolution (0.5 cm$^{-1}$) solar absorption spectra in the Near-Infrared (NIR) range (from 4000
to 11000 cm$^{-1}$), allowing the analysis of COCCON standard species ($CO_2$, $CO$, $H_2O$, $CH_4$). It records double-
sided forward-backward interferograms with a scanner velocity of 10 kHz and typically averages ten scans, so
that a spectrum is acquired approximately every minute. The spectral range of this instrument also allows
obtaining other gas species of interest for volcanology and air quality studies, such as halogen halides ($HCl$, $HF$)
(Butz et al., 2017). From 10 October 2021 to 10 December 2021, following Butz et al. (2017) approach, we
combined the EM27/SUN with a UV-Vis DOAS spectrometer (model Avantes ULS2048). The DOAS
instrument had a 50 μm wide slit entrance, and allows recording spectra in the 270-425 nm spectral range with a
spectral resolution of 0.4 nm. We used a 200 μm wide quartz-made optical fibre. Both instruments shared the
incident sun radiation from the EM27/SUN solar tracker to add simultaneous measurements of $SO_2$ with the
same measurement configuration (Fig. 2).

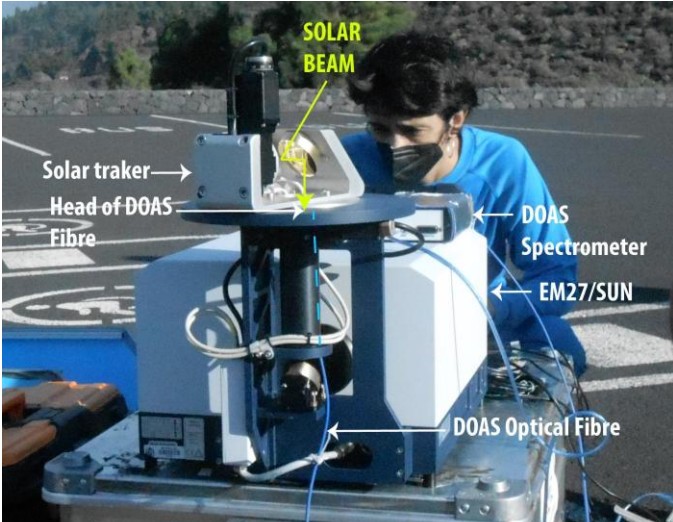

**212 Figure 2: Photograph of the combined EM27/SUN-DOAS direct-sun measurements set-up implemented at the FUE**
**213 station during the Tajogaite eruption (La Palma Island). The DOAS optical fibre is introduced and attached in the**
**214 FTIR sunlight collection tube, pointing towards the solar tracker mirrors. The yellow lines schematize the incident**
**215 sunlight optical path. Photograph taken by R. Campion.**

The DOAS fibre was inserted and attached coaxially into the tube directing the light from the solar tracker
toward the EM27/SUN spectrometer entrance (Fig. 2). By this way, it allows collecting the maximum light
intensity with a minimal disturbance of the solar beam transmitted to the EM27/SUN. The fibre was connected
to the DOAS spectrometer, installed in a protective case sheltered from solar radiation near the EM27/SUN
instrument. DOAS direct-sun absorption spectra were routinely recorded using the MobileDOAS software
(unpublished acquisition program developed for mobile DOAS measurements by C. Fayt and A. Merlaud from
the BIRA-IASB institute) with an integration time of about 30 seconds, with on average 20 scans. The details of
the spectral DOAS and EM27/SUN spectral analysis and retrievals are given in section 3.
**2.2. The Izaña Atmospheric Observatory (IZO, Tenerife Island)**

The proximity of the island of Tenerife to La Palma and the location of the IZO station in the free
troposphere (2373 m.a.s.l) resulted in this international reference observatory to be affected several times by the
Tajogaite volcanic plume. This allowed a more in-depth study of various aspects of the volcanic eruption from a
multi-instrumental perspective. Given its strategic location and its excellent atmospheric conditions, IZO indeed
has a comprehensive state of the art program for atmospheric composition measurements. Uninterrupted
meteorological and climatological observations started in 1916 and, since 1984, IZO has contributed to the
GAW-WMO (Global Atmosphere Watch, World Meteorological Organization) program and to multiple
international networks and databases (WDCGG, WOUDC, NDACC, TCCON, COCCON, AERONET, BSRN,
MPLNET, E-GVAP, NOAA/ESRL/GMD CCGG, etc.; Cuevas et al., 2024, and references therein). Within
IZO's atmospheric research activities, the station is equipped with high-resolution IFS-125HR and low-
resolution EM27/SUN FTIR spectrometers, which provide ongoing long-term solar absorption measurements
since 1999 and 2018, respectively. The EM27/SUN spectrometer is the same instrument model as that
implemented at the FUE station allowing the analysis of $CO_2$, CO, HF and HCl species, as previously described.
The IZO FTIR spectrometers routinely contribute to the Network for the Detection of Atmospheric
Composition Change (NDACC, https://ndacc.larc.nasa.gov, last access: March 2025), Total Carbon Column
Observing Network (TCCON, https://tccon-wiki.caltech.edu, last access: March 2025), and COCCON
(https://www.imk-asf.kit.edu/english/COCCON.php, last access: March 2025) (Schneider et al., 2005; García et
al., 2021). As part of NDACC activities, direct solar mid-infrared (MIR) absorption spectra are measured in the
range of 700 to 4500 $cm^{-1}$, with a spectral resolution of 0.005 $cm^{-1}$. NDACC operations involve co-adding
several scans to increase the signal-to-noise ratio, resulting in each spectrum acquisition taking several minutes.
García et al. (2021) provide further details about the IZO FTIR program. The IZO IFS-125HR MIR solar spectra
were used to analyse the $SO_2$ species alongside HCl and HF, which were also measured from the EM27/SUN
spectra (unlike $SO_2$). This approach further allowed us to evaluate the uncertainties associated with our new
retrieval methods for the HF and HCl species (see section 3 and Appendix A). The details of the spectral
analysis and retrievals are given in section 3.
Moreover, as part of the GAW-WMO program, continuous surface measurements of $CO_2$ (since 1984),
$SO_2$ (since 2006), and CO (since 2008) are performed at IZO. Different in situ analyzers and measurement
techniques have been used for measuring these gases: $CO_2$ with non-dispersive infrared (NDIR) gas Licor
analyzers, CO with gas chromatography (GC) Trace Analytical RGA-3 instruments and $SO_2$ with ultraviolet
(UV) fluorescence analyzers (Thermo 43C-Trace Level). Since 2015, $CO_2$ and CO have also been monitored
using a cavity ringdown spectroscopy (CRDS)-based Picarro G2401 instrument. These observations are carried
out following the strict GAW-WMO measurement protocols and their quality is periodically assessed by
external audits by the World Calibration Center for surface Ozone, CO, Methane and $CO_2$ (WCC-Empa). The
bias for the $CO_2$ and CO measurements in the frame of the GAW-WMO network is ± 0.1 ppm and ± 2 ppb,
respectively (WMO, 2018). For $SO_2$, the uncertainties are expected to be around ± 0.2 ppb (manufacturer
specifications; see also Cuevas et al., 2024 and references therein). This continuous gas monitoring captured the
Tajogaite plume composition on several occasions, when meteorological conditions allowed rapid and direct
transport to the IZO station.
**2.3. Retrieval of $SO_2$ volcanic emission fluxes from TROPOMI data**
The $SO_2$ flux was retrieved by processing the images of the TROPOMI hyperspectral UV-SWIR sensor
on-board the Sentinel-5P satellite. The images were processed by the traverse method, initially developed for the
coarser resolution TOMS satellite images by Bluth et al. (1994) and later adapted to more recent sensors such as
OMI and TROPOMI. The traverses are drawn across the plume semi automatically and the $SO_2$ flux is
calculated using the equation:
$F = \Sigma Xi * Li * sin(\Theta) * v$
where Xi is the $SO_2$ Vertical Column Density (VCD), Li is the length of the pixel, $\Theta$ is the angle between the
pixel row and the wind direction, and v is the plume transport speed. The $SO_2$ VCD was interpolated at plume
height between the $SO_2\_1km$ and the $SO_2\_7km$ subproducts of the version 3 of the TROPOMI $SO_2$ product,
described in Theys et al. (2021). The plume speed was obtained from the Global Data Assimilation System
model of the NOAA, through the READY Archived Meteorology portal
(https://www.ready.noaa.gov/index.php). For the flux calculation, we used the average wind speed at the plume
altitude over the analysed plume portion. The plume altitude was estimated from visual observations such as
photographs, distal webcam images (from Roque de los Muchachos) and HYSPLIT trajectory simulations,
picking the injection altitude that best reproduces the general plume direction observed on the TROPOMI
image, and confirmed with the AEMET/IGN estimates for the coincident days. The $SO_2$ fluxes were finally
estimated using the average of several traverses (usually a few tens and, in some occasions, up to two hundreds,
depending on the coherence of the plume and the wind field that transports it). The traverse method does not
work in cases of plume stagnation in a low wind environment and when the plume is split into several directions
due wind shear. These situations happened during about 30% of the time of the eruption, causing some gaps in
the $SO_2$ flux time series. We also excluded images where the plume was only partially captured.
**2.4 Mobile MultiGAS measurements**
During the eruptive period, mobile surface MultiGAS measurements ($SO_2$, $CO_2$, $H_2O$, $H_2S$) were
carried out into the volcanic plume, between 28 September and 10 October 2021, when meteorological
conditions allowed it to be sampled at ground level at a high concentration. The instrument comprises an
MSR145 datalogger, an Edinburgh Gascard NG for $CO_2$ (0–1000 ppm) with a pump, a City Technology T3ST/F
electrochemical sensor for $SO_2$ (0–50 ppm) and a City Technology T3H electrochemical sensor for $H_2S$ (0–20
ppm). $SO_2$ concentrations up to 7 ppm were measured at distances of about 2 km, East and West of the vent
(Fig. 1). Time series of concentrations of the different gas species were cross-correlated by adjusting the time-
lag (usually between 5 and 9 seconds) and smoothing parameter until the best R-squared correlation coefficient
was obtained. The measurements presented here have R-squared higher than 0.75.
**2.5 Sulfates aerosol measurements**
Samples of aerosols, or particulate matter (PM), smaller than 10 **μ**m ($PM_{10}$) were collected at two sites
in La Palma, at El Paso and at Los Llanos de Aridane and at IZO in Tenerife island. We used high volume
samplers (30 $m^3.h^{-1}$) and quartz microfiber filters (150 mm diameter). Sulfate concentrations were determined
by ion chromatography (Metrohm™ 930 Compact IC FLEX), after a leaching extraction in deionized milli-Q
grade water of the sample by methods described in Rodríguez et al. (2012).
**2.6 Volcanic glass S, Cl and F contents and sulfide droplets composition**
We report 14 new compositions of MIs hosted in olivine, clinopyroxene and amphibole (kaersutite)
crystals (Appendix B3; Supplementary Table). We also report Cl, F, and S contents in tephra glasses that were
measured alongside major elements during the analytical session described in Gonzalez-Garcia et al. (2023),
although only the major element data were published in that study. The volatiles were analysed using a Cameca
SX-100 electron microprobe (EPMA) at the Department of Geosciences of the University of Bremen
(Germany), with an acceleration voltage of 15 kV, beam current of 40 nA and defocused beam of 10 μm,
following the methods described in Gonzalez-Garcia et al. (2023). The instrument was calibrated with a natural
fluorite for F, pyrite for S, and Smithsonian scapolite for Cl. Counting times on peak were 120 s for F and 60 s
for S and Cl. The analyses of F used the PHA (pulse height analysis) setting after Zhang et al. (2016); the
interference of the FeL$\alpha$ line on the FK$\alpha$ peak was corrected using the overlay function of the Cameca software.
The Smithsonian reference materials VG-2 glass, VG-A99 glass and Kakanui hornblende (Jarosewich et al.,
1980) were analyzed along with the samples for precision and accuracy control. Accuracy is better than 6% for
S and Cl and >20% for F; reproducibility is typically better than 10%. In addition, the composition of two
sulfide droplets was semiquantitatively estimated by EDX (energy-dispersive X-ray) spectroscopy.
A Scanning Electron Microscope (SEM) was used to obtain high-resolution back-scattered electron
(BSE) images of two sulfide droplets found in the tephra sample LM-2309 (Las Manchas, 23 September). The
BSE images were acquired using a JEOL JSM-7610F gun emission scanning electron microscope installed at
the Institute of Earth System Sciences, Leibniz Universität Hannover, Germany, using an accelerating voltage of
15kV and a working distance of 15 mm. Bruker ESPRIT software was used for image acquisition.
**3. FTIR and DOAS analysis: Specific $SO_2$, HCl, HF, $CO_2$ and CO retrievals**
**3.1. Spectral analysis from the combined EM27/SUN-DOAS system**

### 3.1.1 EM27/SUN retrievals (CO$_2$, CO, HCl, HF)

The processing of EM27/SUN measurements was performed using the open-source PROFFAST pylotv1.2 packages developed by the KIT and used by the COCCON community. The COCCON standard retrieval procedure used for the analysis of atmospheric CO$_2$, CO, CH$_4$ and H$_2$O species is fully described in Frey et al. (2019), Alberti et al. (2022), Herkommer (2024a,b) and Feld et al. (2024). Here, we provide details only on the specific retrieval strategies that we developed for volcanological applications. The PROFFAST package includes a preprocess code generating the required spectra by a Fast Fourier Transform. The processing incorporates various quality checks, as a signal threshold, intensity variations during recording, requirement of proper spectral abscissa scaling, and generates spectra only from raw measurements passing all checks (the remaining ones being flagged). We used the Instrumental Line Shape (ILS) parameters reported in Alberti et al. (2022) following the COCCON standard recommendations. Calibrated spectra are then analyzed using the PROFFAST radiative transfer and inversion models to derive the total columns by scaling the a priori Volume Mixing Ratio (VMR) profiles iteratively until adjusting the simulated spectra to the measured spectra. Surface pressures are derived from the in situ high precision sensor measurements (PCE-THB-40 at FUE and SETRA-470 at IZO). All the EM27/SUN retrievals presented in this study were performed using the HITRAN 2020 spectroscopic linelists (Gordon et al., 2022). We used meteorological data and a priori VMR profiles based on the sub-daily available GGG2020 TCCON meteorological data (MAP files downloaded from the Caltech server and based on National Centers for Environmental Prediction (NCEP) reanalysis). We adapted the a priori VMR profiles for the target species depending on whether the gas is purely volcanic (low atmospheric abundance) or also has an atmospheric background. The spectral windows and retrieval strategies used for each species are presented in Table 2 and detailed below.

For the analysis of HCl and HF species, we utilized a priori VMR profiles with high concentrations ($1\times10^{-4}$ ppm) up to the altitude of the volcanic plume (~6 km a.s.l., based on IGN/AEMET; Milford et al., 2023), and VMR concentrations for the upper levels derived from the Whole Atmosphere Community Climate Model (WACCM v.6, https://www2.acom.ucar.edu/gcm/waccm, last access: february 2025) average profiles provided by the National Center for Atmospheric Research (NCAR; James Hannigan, personal communication, 2014), which are commonly used by the NDACC community. In this case, we adapted the PROFFAST retrieval inputs so that only the tropospheric portion (up to the altitude of the volcanic plume) was scaled, keeping the stratospheric part as constant. This approach was previously employed to measure volcanic emissions of HCl and HF from Mt. Etna, also relying on low-resolution EM27/SUN spectra (Butz et al., 2017), but utilizing the PROFFIT package for the retrieval. We used new specifically optimised spectral windows (Table 2, HCl_v2 and HF_v2) for the analysis of these two species to be able to detect even very low concentrations, as those detected at the IZO station, 140 km from the eruptive fissure. The analysis was also conducted using the same spectral ranges as Butz et al. (2017) (HCl_v1 and HF_v1 in Table 2) to evaluate the consistency and improvements introduced by the new strategies for our application. Appendix A gives a full comparison between the results obtained using the new and Butz et al. (2017) retrievals, as well as with those from the high-resolution spectra analysis (see section 3.2) for side-by-side measurements.

For the retrieval of volcanic CO and CO$_2$, due to their high atmospheric abundance and variability, we used the COCCON standard retrievals (scaling of the whole profile and use of the COCCON spectral windows and TCCON priori VMRs) and then removed the atmospheric background to derive the volcanic contribution. The column-averaged dry-air mole fraction of CO$_2$ and CO *(XCO$_2$ and XCO)* were estimated using the O$_2$ total columns according to Wunch et al. (2011) ($Xgas = 0.2095 \times Col\ gas \div Col\ O_2$) after applying air mass independent and dependent correction factors (AICF and ADCF). We have slightly modified the standard procedure for performing the O$_2$ retrieval by adding HF as species to be retrieved, using a specific a priori VMR profile based on the WACCM v.6 climatology. However, the HF profile was adjusted to have a constant and significantly higher concentration ($1 \times 10^{-4}$ ppm) up to the maximum plume altitude. For the other interfering gases, we used the a priori VMRs derived from the TCCON GGG2020 MAP files.

To remove the background atmospheric concentrations of XCO$_2$ and XCO, we used the daily-averaged IZO X$_{gas}$ time series to model the long-term natural variability with a third-degree polynomial, which was then interpolated and subtracted from the FUE XCO$_2$ and XCO time series. Examples of XCO$_2$ and XCO background

fits are given in Fig. A3 and A4, respectively. For $CO_2$, an additional intraday variability had to be taken into
account. It was simulated by averaging and fitting some intraday IZO $XCO_2$ time series which were not affected
by the volcanic plume. Intraday simulations were performed for each day, using the average fit and adjusting the
offset. The accuracy of the method was assessed by comparing the simulated $XCO_2$ background at the station
impacted by the volcanic plume with the measured $XCO_2$ background at the other station when it was not
affected by the plume (Fig. A3). The average and maximum absolute difference arising from this procedure
were found to be 0.1 and 0.8 ppm in extreme cases. Finally, the $\Delta CO_2$ and $\Delta CO$ volcanic enhancements were
determined from the $X_{gas}$ enhancements by multiplying them by the dry air columns derived from the surface
pressure measurements and $H_2O$ total columns (Wunch et al., 2011).
**Table 2: Retrieval parameters used for the EM27/SUN and DOAS spectral analysis. "Sim" corresponds to the**
**interfering species only considered for the forward simulations. "*" refers to similar spectral windows as Butz et al.**
**(2017).**

| Gas | Instrument | Spectral Window (cm$^{-1}$) | Interfering Gases | Strategy |
|---|---|---|---|---|
| HCl_v1 HCl_v2 | EM27/SUN | 5684.0 - 5795.0* 5703.5 - 5779.0 | $H_2O$, HDO, $CH_4$ $H_2O$, HDO, $CH_4$ | High (1×10$^{-4}$ ppm) a priori HCl VMR between 0 - 5.8 km beyond: WACCM v.6 |
| HF_v1 HF_v2 | EM27/SUN | 7765.0 - 8005.0* 3995.0 - 4043.0 | $H_2O$, $CO_2$ (Sim), $O_2$ $H_2O$, HDO, $CH_4$ | High (1×10$^{-4}$ ppm) a priori HF VMR between 0 - 5.8 km beyond: WACCM v.6 |
| $CO_2$ | EM27/SUN | 6173.0 - 6390.0 | $H_2O$, $CH_4$ (Sim) | COCCON + post-process background correction |
| CO | EM27/SUN | 4208.7 - 4318.8 | $H_2O$, HDO, $CH_4$, $N_2O$ (Sim), HF (Sim) | COCCON + post-process background correction |
| $O_2$ | EM27/SUN | 7765.0 - 8005.0 | $H_2O$, $CO_2$ (Sim), HF | High (1×10$^{-4}$ ppm) a priori HF VRM between 0 - 5.8 km beyond: WACCM v.6 |
| $SO_2$ | direct-Sun DOAS | 312.0 nm – 326.8 nm | $O_3$ | Levenberg-Marquardt (LM) algorithm |

### 3.1.2 DOAS retrievals ($SO_2$)
Solar DOAS spectra were processed using the QDOAS v2.111 software (Dankaert et al., 2014),
applying a Levenberg-Marquardt (LM) algorithm to retrieve the Slant Column Densities. We used the same
analysis strategy as described in Taquet et al. (2023), with the key parameters summarized in Table 2.
Wavelength calibration and slit function were determined by laboratory close-path measurement using a low-
density mercury lamp, and further adjusted based on the position and widening of the Fraunhofer lines during
the QDOAS processing. $SO_2$ was retrieved in the 312.0–326.8 nm spectral window according to Butz et al.
(2017). The high resolution solar spectrum from Chance and Kurucz (2010) was used as the reference spectrum.
We used the cross-section at 298 K from Vandaele et al. (2009) for $SO_2$ and the cross-section at 221 K from
Burrows et al. (1999) for the interfering gas $O_3$. A third-order polynomial function was included in the fitting
routine to remove the broadband extinction. The I0 effect, due to the limited resolution of the spectrometers
(Platt et Stutz, 2008), was corrected using the QDOAS I0-correction algorithm applied for six fixed $SO_2$ slant
column values of 0.0, $1.0 \times 10^{18}$, $2.0 \times 10^{18}$, $3.0 \times 10^{18}$, $4.0 \times 10^{18}$, $5.0 \times 10^{18}$ molec/cm$^2$ (the latter is close to the
maximum uncorrected slant column). Then, each corrected value is determined by interpolating the corrected

| 400 | slant columns values. Unlike radiance scattered light measurements, the direct-sun configuration remains |
| 401 | unaffected by the Ring effect (Herman et al., 2009), which therefore was not considered in the retrieval. Finally, |
| 402 | $SO_2$ slant columns were converted into vertical columns by dividing them by the SZA-dependent air mass factor |
| 403 | (1/cos (SZA)) to be combined with the FTIR data. |

## 3.2 IFS-125HR analysis (HCl, HF and $SO_2$)

The HCl and HF retrieval strategy from the IFS-125HR spectra is based on the NDACC-IRWG recommendations (Infrared Working Group, IRWG, 2014), and on the adapted retrievals for volcanological applications reported in Taquet et al. (2019) and Stremme et al. (2023). However, they have been optimised here to properly capture tropospheric volcanic contributions up to 140 km from the eruptive fissure. Consistently with the NDACC approach, both species were retrieved using the non-linear least-squares fitting algorithm PROFFIT (Profile Fit, Hase et al., 2004), and considering the specified spectral regions and interfering gases given in Table 3. The inversion procedure is solved using a first-order Tikhonov–Phillips regularization (L1, Rodgers, 2000) on a logarithmic scale, where the VMR a priori profiles for the interfering gases are taken from WACCM v.6 climatological profiles. The NCEP 12:00 UTC daily temperature and pressure profiles are employed for the radiative transfer simulations.

The most significant changes with respect to NDACC involved the a priori VMR profiles considered for the target gases, vertical L1 regularization, and the spectroscopic database. Similarly to the EM27/SUN analysis, we adopt modified HF and HCl a priori VMR profiles with high concentrations ($1\times10^{-4}$ ppm) up to the maximum plume altitude (~6 km a.s.l.), which are completed for the IFS-125HR using WACCMv.6 information beyond this altitude. In addition, the 2020 HITRAN spectroscopic linelists were utilized for all gases. Finally, in contrast to the NDACC approach, where the lowermost and uppermost altitude levels are fixed to the a priori to ensure stability in the retrieval, in this study, the first level is left unconstrained to provide flexibility in the retrieval process in the lower troposphere.

In the case of $SO_2$, a harmonized and standardized FTIR strategy is not available within NDACC. Therefore, in this work, we employ the strategy developed by García et al. (2022), which has been successfully applied to various NDACC FTIR sites affected by volcanic $SO_2$ emissions (Smale et al., 2023; García et al., 2025). This approach is based on the study by Taquet et al. (2019), which presents $SO_2$ total column amounts from the measured solar absorption spectra in the 2500 cm$^{-1}$ region using a scaling retrieval and the inversion code PROFFIT. Similarly to HF and HCl volcanic products, the $SO_2$ a priori VMR profiles are adapted in the lower troposphere, while climatological WACCMv.6 profiles are considered for all interfering gases (Table 3). Appendix A provides a summary of the comparison between the standard NDACC HCl and HF products and those developed in this study, the new IFS-125HR $SO_2$ retrievals, as well as the comparison between all the IFS-125HR and EM27/SUN products.

**Table 3: Retrieval parameters used for the IFS-125HR analysis. "Sim" corresponds to the interfering species only considered for the forward simulations. The spectral windows are acquired using the NDACC filter SC (S3) for HCl, with the NDACC filter SA (S1) for HF, and with the NDACC filter SF (S6) for $SO_2$. Therefore, they are almost coincident, but not simultaneous observations.**

| Gas | Spectral Window (cm$^{-1}$) | Interfering Gases | Strategy |
|---|---|---|---|
| HCl | 2727.73-2727.83<br>2775.60-2775.90<br>2821.40-2821.75<br>2925.75-2926.10 | $H_2O$ (Sim), HDO (Sim), $O_3$, $CH_4$ (Sim), OCS, $NO_2$, $N_2O$ (Sim) | High ($1\times10^{-4}$ ppm) HCl a priori VMR between 0 - 5.6 km, above: WACCM v.6 |
| HF | 4000.90-4001.05<br>4038.85-4039.08 | $H_2O$, $O_3$ (Sim), $CH_4$ (Sim) | High HF ($1\times10^{-4}$ ppm) a priori VMR between 0 - 5.6 km, above: WACCM v.6 |
| $SO_2$ | 2480.00-2520.00 | $H_2O$, $CO_2$, $O_3$, $CH_4$, $N_2O$ | High $SO_2$ ($1\times10^{-2}$ ppm) a priori VMR between 0 - 5.6 km, above: WACCM v.6 |

## 4. Results

### 4.1. Evolution of the volcanic plume composition during the Tajogaite eruption

The temporal variability of the Tajogaite plume composition is examined through the time series of the ratios, some of them involving species with contrasting exsolution depths. Daily $\Delta CO_2/SO_2$, $HCl/SO_2$, $HF/SO_2$, $HCl/\Delta CO_2$, $HF/\Delta CO_2$, $\Delta CO/SO_2$ and $\Delta CO/\Delta CO_2$ molecular ratios were estimated from the daily correlation plots of the total column time series, following the methodology as detailed in Taquet et al. (2019, 2023) and are reported in Fig. 3. The same method used for column-averaged ratios was applied to calculate the surface concentration ratios from GAW and MultiGAS measurements (also presented in Fig. 3). The background contribution of atmospheric species ($CO_2$ and $CO$) to these measurements was removed using daily polynomial curves fitted from the surface measurements without contribution of volcanic emissions (i.e. $SO_2$ <0.05 ppm). Additionally, we reported in the same figure our MultiGAS $\Delta CO_2/SO_2$ measurements, obtained on 29 September, 2 and 7 October from Las Manchas (~500 m a.s.l., SW from the eruptive fissure, Fig. 1) and from the El Jable viewpoint (2100 m a.s.l., E of the eruptive fissure, Fig. 1), ranging between 1.7 and 14.3. The scarcity of FTIR measurements from early November until the end of the eruption, across all measurement techniques, is mainly due to poor or unsuitable weather conditions.

Our column-averaged $\Delta CO_2/SO_2$ molecular ratios range between $9 \pm 6$ and $63 \pm 28$ (9-24 at IZO and 14-63 at FUE) during the eruption. These values are consistent with the surface measurements at IZO (ratios from $5.6 \pm 0.1$ to $18.3 \pm 0.7$) and with our MultiGAS measurements at La Palma (1.7 to 14.3). These values are also consistent with the proximal measurements reported in the literature including Open Path FTIR (Burton et al., 2023 and Asensio-Ramos et al., 2025) and MultiGAS (Burton et al., 2023; Ericksen et al., 2024) measurements, ranging between 2 and 52 (shaded area in Fig. 3). All the measured $\Delta CO_2/SO_2$ ratios define an increasing trend, at least until 2 November 2021 and show more scatter after this date (Fig. 3).

$HCl/SO_2$ molecular ratios range between $0.02 \pm 0.002$ and $0.17 \pm 0.01$ (from 0.02 to 0.05 at IZO and from 0.02 to 0.17 at FUE) and show short-term variations around a nearly constant daily average of ($0.05 \pm 0.03$) throughout the entire eruptive period. These ratios are consistent with the values of $SO_2/HCl$ of 16.8 and 8 ($HCl/SO_2$=0.06 and 0.12, respectively) reported in Burton et al. (2023), which corresponds to a lava fountaining plume and spattering event (Fig. 3). It is also consistent with the more recently published ratios ranging between 0.04 and 0.2 (Asensio-Ramos et al., 2025; Fig. 3). $HF/SO_2$ molecular ratios vary between $0.0012 \pm 0.0002$ and $0.081 \pm 0.007$ (from $0.001 \pm 0.001$ to $0.082 \pm 0.007$ at FUE and from $0.007 \pm 0.002$ to $0.037 \pm 0.025$ at IZO) and show a similar day-to-day variability to that observed for the $HCl/SO_2$ ratios through the eruptive period. $HCl/\Delta CO_2$ molecular ratios exhibit values from $(6 \pm 1) \times 10^{-4}$ and $(4.1 \pm 0.1) \times 10^{-3}$ at FUE and from $(2 \pm 1) \times 10^{-3}$ to $(3 \pm 1) \times 10^{-3}$ at IZO, while the $HF/\Delta CO_2$ ratios range from $(0.5 \pm 0.1) \times 10^{-4}$ to $(4.5 \pm 0.1) \times 10^{-3}$ at FUE and from $(2.6 \pm 0.3) \times 10^{-4}$ to $(2.7 \pm 0.2) \times 10^{-3}$ at IZO. Like $HCl/SO_2$ and $HF/SO_2$, the $HCl/\Delta CO_2$ and $HF/\Delta CO_2$ ratios exhibit similar day-to-day variability. Their fluctuations include short-term decreasing trends, as observed between 2 and 14 October 2021 and between 21 October and 4 November 2021. The $\Delta CO/SO_2$ FTIR ratios span from $0.13 \pm 0.01$ to $0.66 \pm 0.03$ at FUE and from $0.02 \pm 0.01$ to $0.17 \pm 0.07$ at IZO, and are relatively stable around the average of 0.24 with one extreme event, observed between 1 and 4 November 2021.

During the initial phase of the eruption, prior to the eruptive pause on 27 September 2021, our ratios were comparable to those observed throughout the rest of the eruptive period, with $\Delta CO_2/SO_2$ ranging between $5.6 \pm 0.1$ and $9 \pm 1.1$, $HCl/SO_2$ between $0.031 \pm 0.005$ and $0.049 \pm 0.007$, and $HF/SO_2$ between $0.009 \pm 0.003$ and $0.022 \pm 0.006$. A significant and abrupt increase in all species-to-$SO_2$ ratios is observed on 2-3 November 2021, which also coincides with a minor peak in the HCl and HF-to-$\Delta CO_2$ ratios. This event represents a notable and enduring change in gas ratio variability involving $CO_2$, (i.e. $\Delta CO_2/SO_2$ and $HCl/\Delta CO_2$) and coincides with a sudden decrease in the amplitude of seismic tremor (VLP and LP, Fig. 3 and Bonadonna et al., 2022). Prior to this date, the variability in the $\Delta CO_2/SO_2$ ratio closely followed the increasing trend of VLP tremor amplitude, while afterwards it declined and exhibited a noticeable short-term variability until the end of the eruption. This noticeable change depicts two periods in our dataset (here after phase I and II), whose relationship with the previously described events and timeframes of the eruption (Bonadonna et al., 2022; Ubide et al., 2023; Milford et al., 2023) will be discussed in section 5. For $HCl/\Delta CO_2$ and $HF/\Delta CO_2$, the ratios are

significantly lower during phase II (average of 0.0012 ± 0.0005 and 0.0007 ± 0.0004, respectively) than during phase I (average of 0.0027 ± 0.0009 and 0.0014 ± 0.001, respectively). For other species, only a brief spike is noted at this time, with ratios returning to Phase I levels at the onset of Phase II.

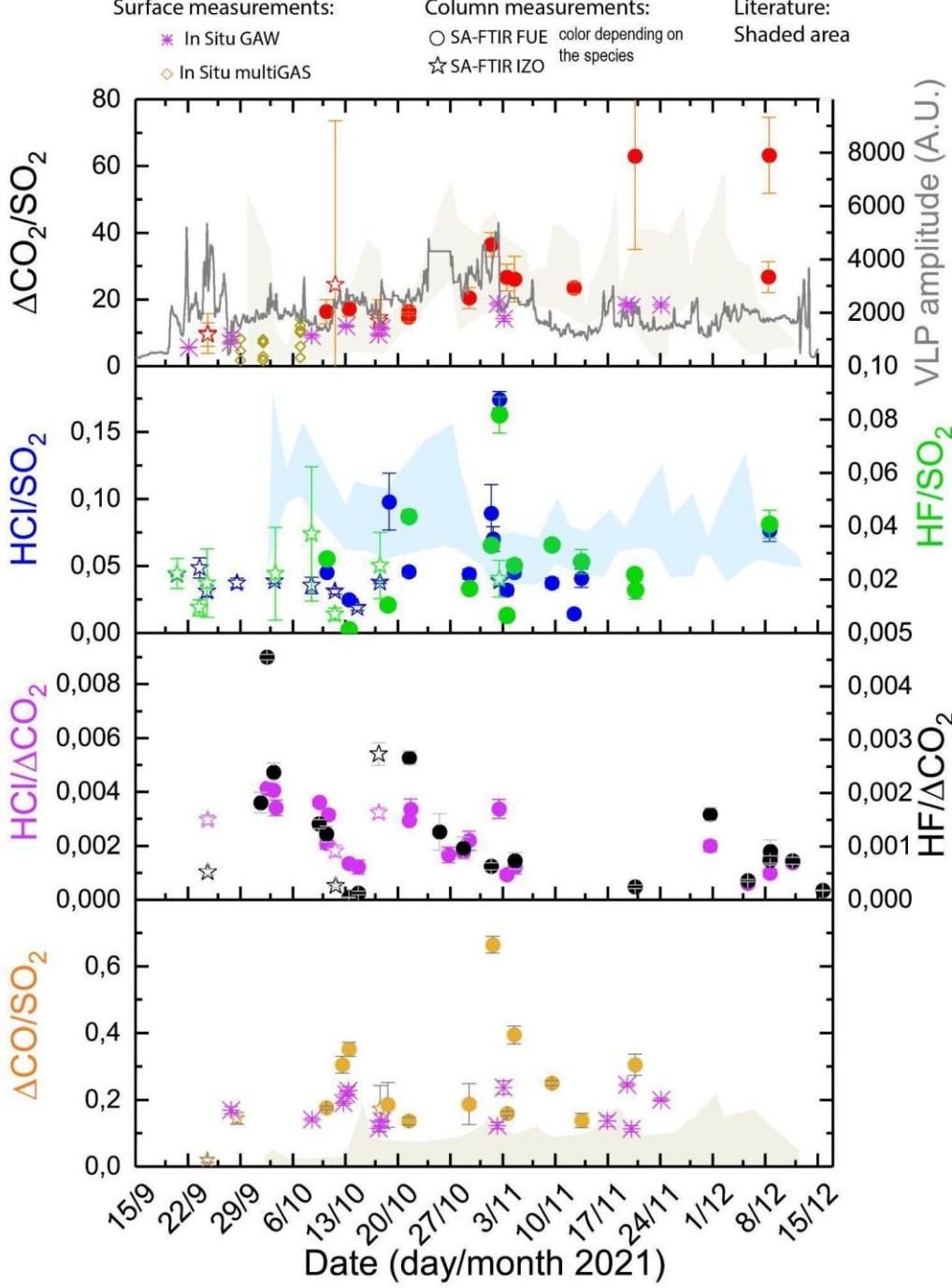

**Figure 3: Variability of the Tajogaite volcanic plume composition during the eruption. Daily molecular ratios are calculated from the daily species-to-$SO_2$ or species-to-$CO_2$ correlation plots of the total columns (SA: solar absorption FTIR and DOAS measurements) and surface (GAW and MultiGAS analysis) time series. Only the ratios with a $R^2 > 0.6$ in the correlation plots are reported here to exclude those with poor reliability. Data from literature is presented as shaded areas, including the ratios reported by Burton et al. (2023), Ericksen et al. (2024) and Asensio-Ramos et al. (2025). The latter were derived from MultiGAS and Open-Path FTIR measurements. Very Long Period (VLP; 0.4-0.6Hz) tremor amplitude (upper panel, gray line) is taken from Bonadonna et al. (2022).**

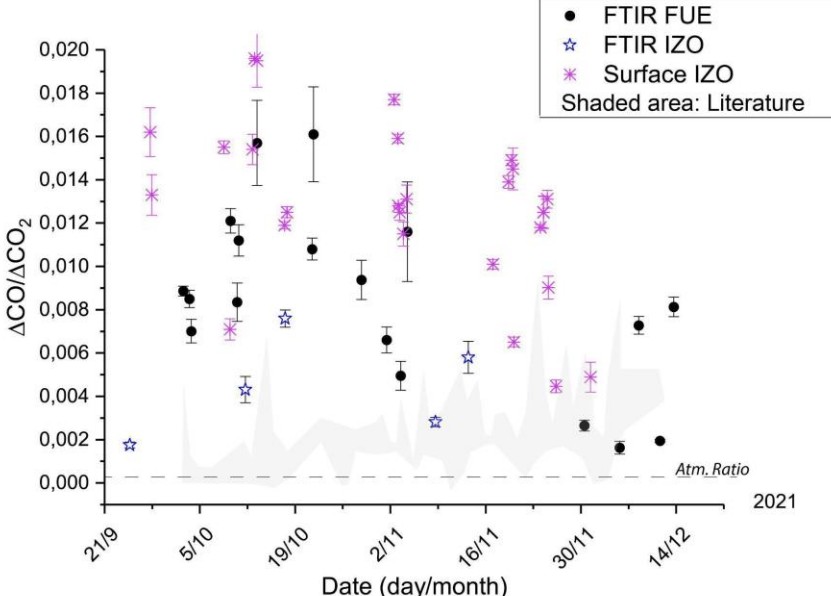

**Figure 4: Time series of the $\Delta CO/\Delta CO_2$ ratio at both FUE and IZO stations. The ratios at IZO presented here are**
**derived from in-situ (purple) and FTIR (blue stars) measurements. Shaded areas present the data from the literature**
**including Álvarez et al. (2023) measured by solar absorption FTIR and Asensio-Ramos et al. (2025) derived from OP-**
**FTIR measurements. The dashed black line represents the long term atmospheric ratio (Atm. Ratio) measured at**
**IZO (derived from Garcia et al., 2022).**

Figure 4 presents the time series of $\Delta CO/\Delta CO_2$ ratios derived from FTIR solar absorption
measurements at the FUE and IZO stations throughout the eruption, alongside with in situ surface measurements
at IZO (GAW data). The $\Delta CO/\Delta CO_2$ values observed at both sites and using both techniques are of the same
order of magnitude, and exceed by more than one order of magnitude the average atmospheric background ratio
at IZO (~0.0002). At FUE, the FTIR-derived ratios show a progressive increase from 0.0016 to 0.016 during the
first 30 days of the eruption, followed by a decrease to lower values before mid-November. The surface
$\Delta CO/\Delta CO_2$ ratios at IZO fall within a similar range to those derived from FTIR at FUE, with some coinciding
values in very good agreement. On average, the surface ratios at IZO are higher than the FTIR-derived ones at
the same site. This discrepancy may be explained not only by the strong short-term variability in the $\Delta CO/\Delta CO_2$
ratios (only a few data points are coincident), but also by the fact that, although all these points coincide with the
presence of $SO_2$ (indicating the presence of volcanic plume), the correlation between $\Delta CO$ and $SO_2$ is relatively
weak ($R^2 < 0.6$), suggesting additional sources contributing to the CO enhancements. Furthermore, satellite
imagery suggests that, on these days, the line of sight of the IZO FTIR instrument may have intersected aged
volcanic plumes, potentially altering the retrieved $\Delta CO/\Delta CO_2$ ratios due to both geometric and compositional
effects. The difference between the surface $\Delta CO/\Delta CO_2$ ratios observed at FUE and IZO and those (shaded area)
reported by Asensio-Ramos et al. (2025) is discussed in Section 5.
**4.2. $SO_2$, $CO_2$ and halogen-derived volcanic emission fluxes and total emissions**

$SO_2$ volcanic emission fluxes were estimated whenever the weather conditions made it possible
following the method described in section 2.3 and reported in Fig. 5. The $SO_2$ volcanic emission fluxes retrieved
during this eruption exhibited a remarkably strong correlation ($R^2=0.92$, Fig. D3) with the daily $SO_2$ masses
(taken from MOUNTS website; Valade et al., 2019). To fill the long-term gaps in our $SO_2$ fluxes time-series, a
less reliable mass-derived product was included, derived from the linear relation between the $SO_2$ volcanic
emission fluxes and daily mass (Fig. 5A, empty circles). This was only applied to days with minimal
accumulation. The $SO_2$ volcanic emission fluxes time series exhibit a decreasing exponential trend (red curve),
with an equation of the form $y = a \times e^{-bx}$ and a coefficient of determination $R^2= 0.63$. Most mass-derived
products were found to closely follow the overall trend (Fig. 5A, red curve), indicating that, despite inherent
uncertainties, these estimates are likely robust enough to assess long-term variability in this case study. This also
suggests that short-term variations in wind direction or partial plume coverage in satellite images (initial
filtering criteria) may have a limited impact on the observed global trend.

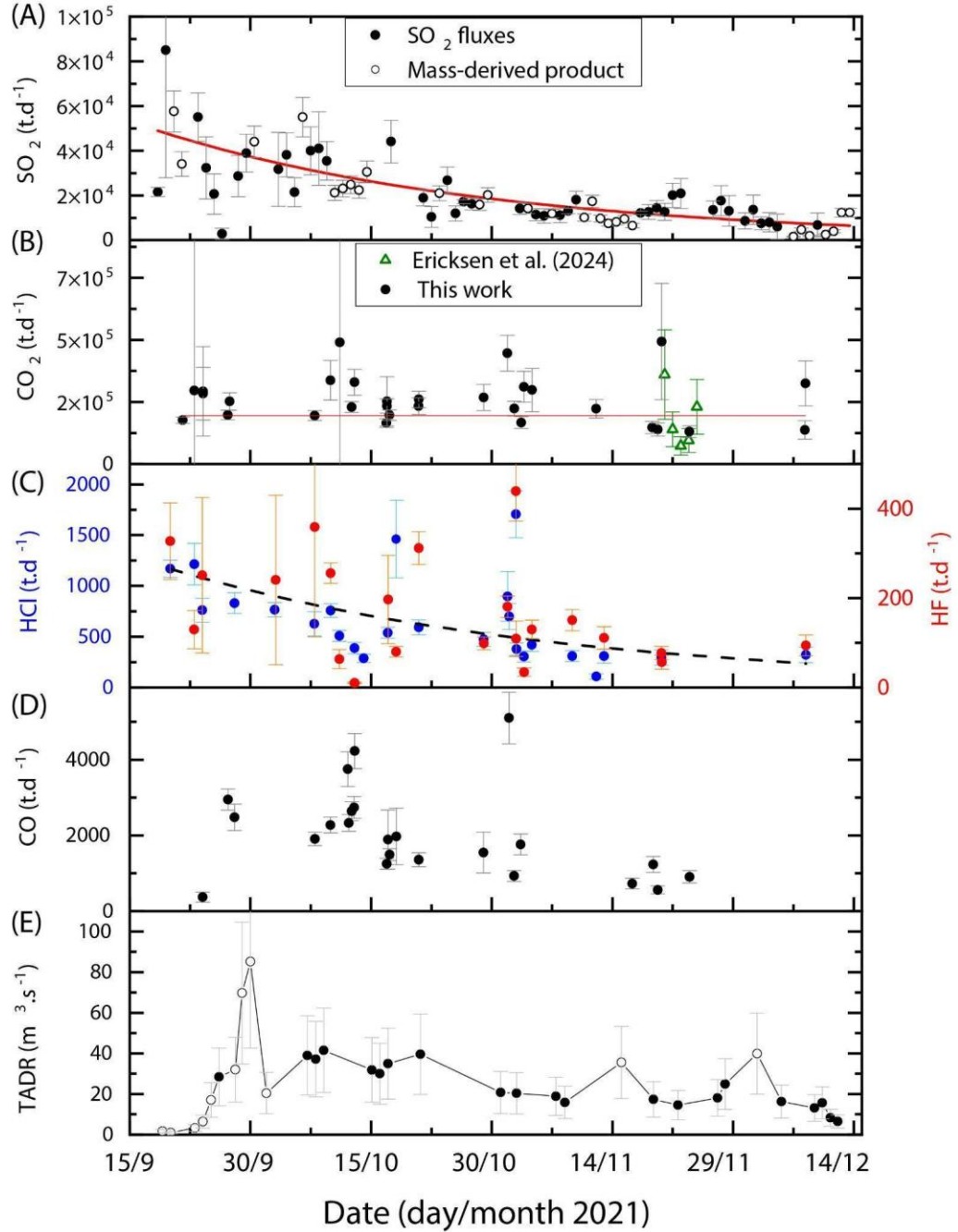

**Figure 5: (A) to (D) Emission fluxes of $SO_2$, $CO_2$, HCl, HF and CO and (E) corrected TADR, following Plank et al.**
**(2023), during the eruption. The thick red line in (A) is the exponential fit to the $SO_2$ emission fluxes time series. The**
**red line in (B) is the linear regression for the dataset. The black dashed line in (C) is the exponential fit to the HCl**
**time series. Black points in (E) are part of the TADR-$SO_2$ emission flux correlation.**
Volcanic emission fluxes for the other species were estimated by using daily species-to-$SO_2$ ratios and
either (i) interpolating the exponentially decreasing fit of $SO_2$ fluxes or (ii) performing a linear interpolation of
the $SO_2$ emission fluxes time series. The HCl, HF, $CO_2$ and CO volcanic emission fluxes are shown in Fig. 5 B-
D, concurrently with the Time-Averaged Discharge Rate (TADR, Fig. 5E) time series of Plank et al. (2023),
multiplied by a factor of 2, as suggested by the authors to take the underestimation of the lava volume into
account.
A significant observation is the long-term decrease in the volcanic emission fluxes of $SO_2$, HCl, HF,
and CO, which aligns with the TADR trend throughout the eruption, in contrast to the nearly stable trend of
CO$_2$. The similarity of the trends of the daily average SO$_2$ emission fluxes and the TADR is further supported by
an excellent correlation (Pearson coefficient R=0.94; see Fig. E1), defining a slope of 14.1±1.2 kg of SO$_2$ per
thermal m$^3$ of discharged lava (lava volumes estimated using the radiant flux). This relationship includes 21/27
of the available TADR-fluxes pairs) and is mainly valid from 7 October 2021 onwards (Fig. E2, full circles).
The points corresponding to the onset of the eruption (outliers represented as hollow circles in Fig. E2) have
either higher SO$_2$ fluxes for a given TADR until the 25/09 or higher discharge rates after the 27/09 eruptive
break and until 30/09 at least (the next pair is that of 07/10, belonging to the correlation).
Another important observation is that the SO$_2$ flux peak recorded during the first week of the eruption,
accounting for approximately 20% of the total SO$_2$ emissions, occurs during a period of apparently low TADR
and around ten days prior to the first peak with maximum values of TADR for the eruption. The relationship
between the SO$_2$ volcanic emission fluxes and the TADR is examined in the light of the petrological data in
section 5.
Furthermore, the early November peaks in the HF, HCl, and CO emission fluxes time series, which align with
those observed in several ratios time series (Fig. 3), correspond to the inflection point in the overall flux decline,
occurring near the end of Phase I, as defined by Milford et al. (2023). Since the CO$_2$ volcanic emission fluxes
appear to be nearly constant throughout the entire eruptive period, we can interpret the lower HCl and HF-to-
CO$_2$ ratios of phase II as the result of globally lower fluxes during this period, in line with the pressure decrease
in the reservoir (Charco et al., 2024).
Table 4 presents the average volcanic emission fluxes for each species over the entire eruption distinguishing
between the results from the two previously described methods. Total emissions were estimated by combining a
Monte Carlo approach to account for uncertainties with trapezoidal integration to compute the area under the
curve, and are also reported in Table 4. The average fluxes over the entire eruptive period and the estimated total
emissions of SO$_2$, HCl, HF, and CO$_2$ (Table 4) provide insight into the scale of the emissions of this eruption
with respect to other emission sources.
**Table 4: Estimate of total emissions during the eruption from gas to SO$_2$ ratios and SO$_2$ emission fluxes. The emission**
**fluxes estimates were performed using (1) an exponential fit for the SO$_2$ emission fluxes interpolation and (2) using**
**direct linear interpolation of daily SO$_2$ emission fluxes estimates (results between brackets). Total emissions to the**
**atmosphere are then derived combining the Monte Carlo and trapezoid integration methods.**

| Species | Average specie to SO$_2$ mass ratios | Average volcanic emission fluxes (kg.s$^{-1}$) | Total emissions (Mt) Estimates using exponential fit for SO$_2$ volcanic emission fluxes interpolation (Estimate using direct interpolation of SO$_2$ fluxes) |
|---|---|---|---|
| **SO$_2$** | 1.0 | 300 ± 230 | 1.81 ± 0.18 (1.86 ± 0.09) |
| **CO$_2$** | All studies: 12 ± 10 This study: 14 ± 9 | 2981 ± 1105 | 19.4 ± 1.8 (20.5 ± 1.9) |
| **HCl** | 0.03 ± 0.02 | 7 ± 4 | 0.05 ± 0.01 (0.043 ± 0.003) |
| **HF** | 0.0074 ± 0.0053 | 1.9 ± 1.3 | 0.013 ± 0.002 (0.013 ± 0.002) |
| **CO** | 0.09 ± 0.05 | 23 ± 14 | 0.123 ± 0.005 (0.138 ± 0.009) |

The total SO$_2$ emissions of 1.81 ± 0.18 Mt, derived from our exponentially decreasing fit, is similar to that
reported in Milford et al. (2023) using the daily SO$_2$ volcanic emissions derived from TROPOMI data (credit:
ESA, MOUNTS). These total SO$_2$ emissions are comparable to the emissions of the submarine 2011 Tagoro
eruption at El Hierro, that released between 1.8 and 2.9 Mt SO$_2$ into the ocean (estimated using the petrologic
method; see Longpré et al., 2017).

During the Tajogaite eruption, the highest $SO_2$ emission fluxes occurred during the first ten days of the eruption (median of 37 kt/day during this period), and then had a lower median of about 20 kt/day. These $SO_2$ emission rates are the same order of magnitude as the most recent basaltic eruptions such as for instance Piton de La Fournaise in 2020 (average: 0.9 kt/day; max: 25 kt/day, Hayer et al., 2023) in La Reunion island, Bárðarbunga in 2014-2015 (average of 50 kt/day over 6 months, Pfeffer et al., 2018) in Iceland, and lower than that found at Kilauea in 2018 (average of 200 kt/day; Kern et al., 2020), but the latter two exhibiting much higher eruptive TADR. For Tajogaite eruption, the high $SO_2$ fluxes result from the high sulfur content of parental magma, as reflected by the average content of 3360 ppm in our MIs (Supplementary data), similar to the value of 3500 ppm reported in Burton et al. (2023) and Dayton et al. (2024).

For $CO_2$, we obtained a steady average emission flux of 260 ± 24 kt/day, and total emissions of 19 ± 2 Mt over the course of the eruption. This result aligns closely with the estimates of 28 ± 14 Mt reported by Burton et al. (2023). These emissions represent 15% of global subaerial volcanic and tectonic annual emissions (Fischer and Aiuppa, 2020) or the equivalent of the annual $CO_2$ budget of Ocean Island Basalt (OIB) volcanism, as estimated by LoForte et al. (2024). The high $CO_2$ emissions with respect to the low extruded magma volume during Tajogaite eruption, compared to other effusive eruptions, are explained by the extraordinarily carbon-rich magma, as it is reflected in both fluid and melt inclusions (up to 2 wt% $CO_2$ in MIs; Dayton et al., 2024). This is a characteristic of Macaronesian magmas and possibly of global OIB (Burton et al., 2023; LoForte et al., 2024; Van Gerve et al., 2024).

Daily total CO emissions emitted during the eruption, averaging 2 kt/day, were exceptionally high, with a cumulative total of 0.12 ± 0.01 Mt. Only few volcanic CO emissions are reported in the literature, such as 0.15 kt/day at Erebus volcano (Wardell et al., 2004), 0.007 kt/day at Oldoinyo Lengai (Oppenheimer et al., 2002), 0.16 to 0.27 kt/day at Nyiragongo volcano (Sawyer et al., 2008a), 0.0007 kt/day at Erta Ale (Sawyer at al., 2008b) and are about one order of magnitude lower than our estimates during the Tajogaite eruption.

Finally, our estimated HCl and HF total emissions are about 50 ± 10 kt and 13 ± 2 kt, respectively, with an average of 604 ± 340 t/day and 173 ± 86 t/day. These emissions are in the same order of magnitude as that observed for other basaltic volcanoes, such as Etna (300-1300 t/day of HCl during the 2008-2009 eruption reported in Spina et al., 2023; 800 t/day of HCl and 200 t/day of HF in 1997 reported by Oppenheimer et al., 1998), Bárðarbunga volcano (500 t/day and 280 t/day for HCl and HF, respectively, reported in Galeczka et al., 2018). HCl and HF emissions from Tajogaite eruption are more than an order of magnitude higher than those observed at Kilauea volcano, which reported 12-22 t/day of HCl and 6-9 t/day of HF in 2008 and 2009 (Mather et al., 2012).

## 5. Discussion

### 5.1. Comparison of $CO_2$, CO, HCl and HF to $SO_2$ ratios from different measurement methods and sites

One of our key results is the remarkably strong consistency between the measured volcanic gas species-to-$SO_2$ ratios, whatever the measurement site, the technique and the instrument used (Fig. 3). The measurements conducted at the IZO station gave the excellent opportunity to assess the robustness of our estimated ratios, using both EM27/SUN and IFS-125HR instruments and their consistency with surface measurements. We found an excellent agreement between the HCl and HF total columns (with volcanic plume contribution) derived from the IFS-125HR and EM27/SUN products (see Appendix A for details).

We found a good comparability for the available $\Delta CO_2/SO_2$ and $\Delta CO/SO_2$ between surface and column measurements, reflecting an efficient vertical mixing. This also suggests that when the volcanic plume is detected by the surface measurements at the IZO station, the ground level concentrations are representative of the average volcanic plume composition. Since the IZO station is often located above the base height of the trade wind inversion (TWI) layer (Milford et al., 2023), volcanic plumes detected at IZO were typically transported rapidly through the low free troposphere. The progressive decrease in plume injection height throughout the eruption, combined with seasonal changes in the vertical stratification of the atmosphere (TWI height), resulted in sparse detections of the plume at the IZO station after mid-November 2021 (Milford et al., 2023). This led to a reduction of the coincident surface and total column observations.

Moreover, the comparison of ratios at different distances from the eruptive vents (i) at IZO (140 km) and (ii) near the active vent measured by OP-FTIR or MultiGAS (this work; Burton et al., 2023; Ericksen et al., 2024) allows qualitative assessment of the impact of in-plume reactions on our measurements. The ratios taken from Burton et al. (2023) were derived from either in situ ground-based or drone-borne MultiGAS measurements within the plume close to the volcanic vents, or, after 02/10/2021, from Open-Path FTIR measurements pointing to the eruptive column and using the lava fountain as a source. Those reported by Ericksen et al. (2024) are limited to ground-based MultiGAS measurements. In any case, the gas measured by these authors corresponds to the plume less than 1 km from the volcanic vents. Since $CO_2$ is a non-reactive species, a significant conversion of $SO_2$ into sulfate aerosols ($H_2SO_4$) during the transport between La Palma and IZO should increase the $\Delta CO_2/SO_2$ ratio. Hence, if significant conversion of $SO_2$ to sulfates occurred during the transport, the IZO ratios should be higher than those measured closer to the volcano. To examine this aspect, we estimated the plume age for each recorded event using the Hysplit transport model, in both retro-trajectories and forward simulation configuration mode. For meteorological data, we utilized 72-hour extended files containing high-resolution meteorological information derived from the WRF-ARW model as input. This model runs twice a day, using initial and boundary conditions from ECMWF's HRES-IFS data, with a resolution of $0.09° \times 0.09°$ (for further details, refer to Appendix C). Table 5 shows the coinciding values of the $\Delta CO_2/SO_2$ ratios measured at less than 1 km from the eruptive fissure (Burton et al., 2023) and at IZO (this work) and an estimate of plume age for each event. Despite the limited number of coincident events at the two sites, no clear dependence of this ratio on distance was observed for plumes with an age of 12 hours or less. Certain similarity was found, at least until the beginning of November, even in cases of relatively old plumes (~12h), suggesting a swift transport between La Palma and Tenerife islands and negligible in-plume reactions, at least indistinguishable within the uncertainties of the ratios. In the troposphere, the $SO_2$ to $SO_4^=$ oxidation rates vary significantly, from a few percent per hour by in-cloud droplet processes (driven by aqueous phase oxidation e.g. $H_2O_2$) to a few percent per day (in dry air, driven by OH radicals) (Seinfeld and Pandis, 1998). Our results suggest that this latter (slow dry oxidation) process may be the prevailing one during the transport in the dry free-troposphere, from La Palma to IZO. This interpretation is supported by the sulfate aerosols measured in situ in La Palma (Rodríguez et al., submitted) and at IZO, when the volcanic plume reaches the station, plotted in Fig. 6. Figure 6A reports the statistical distribution of the ratio (in percent, %) of particulate sulfur (S(p), i.e. sulfate $SO_4^=$) over total sulfur (i.e. gas sulfur as sulfur dioxide (S(g)) plus S(p)) measured in the aerosols smaller than 10 microns ($PM_{10}$) at IZO and at La Palma during the eruption. Figure 6B shows the correlation plot of S(g) as a function of S(g)+S(p). We observe a higher maximum conversion rate at IZO (45%) than in La Palma (20%), as expected. However, 80% of the dataset (Fig. 6A and B) presents a conversion rate of their sulfur content to $SO_4$ below 15% and 7% at IZO and La Palma, respectively.

**Table 5: Comparison of the $\Delta CO_2/SO_2$ ratio values at two different distances from the Tajogaite eruptive center and estimate of plume age at IZO station. FTIR ratios are given between brackets to distinguish them from surface ratios.**

| Date | Burton et al. (2023) Crater | IZO (140 km) Surface ratios (FTIR ratios) | Plume age at IZO (hour) |
|---|---|---|---|
| 27/09/2021 | 6.8 | 7.0 ± 0.5 | ~8h |
| 07/10/2021 - 08/10/2021 | 13 | 9.0 ± 0.5 | ~3h |
| 13/10/2021 | 9;11 | 12.0 ± 0.3 | ~12h |
| 16/10/2021 - 17/10/2021 | 29 | 10 ± 0.5 (13 ± 1) | ~12h |
| 23/11/2021 - 24/11/2021 | 38.3 | 18.3 ± 0.7 | ~12h |

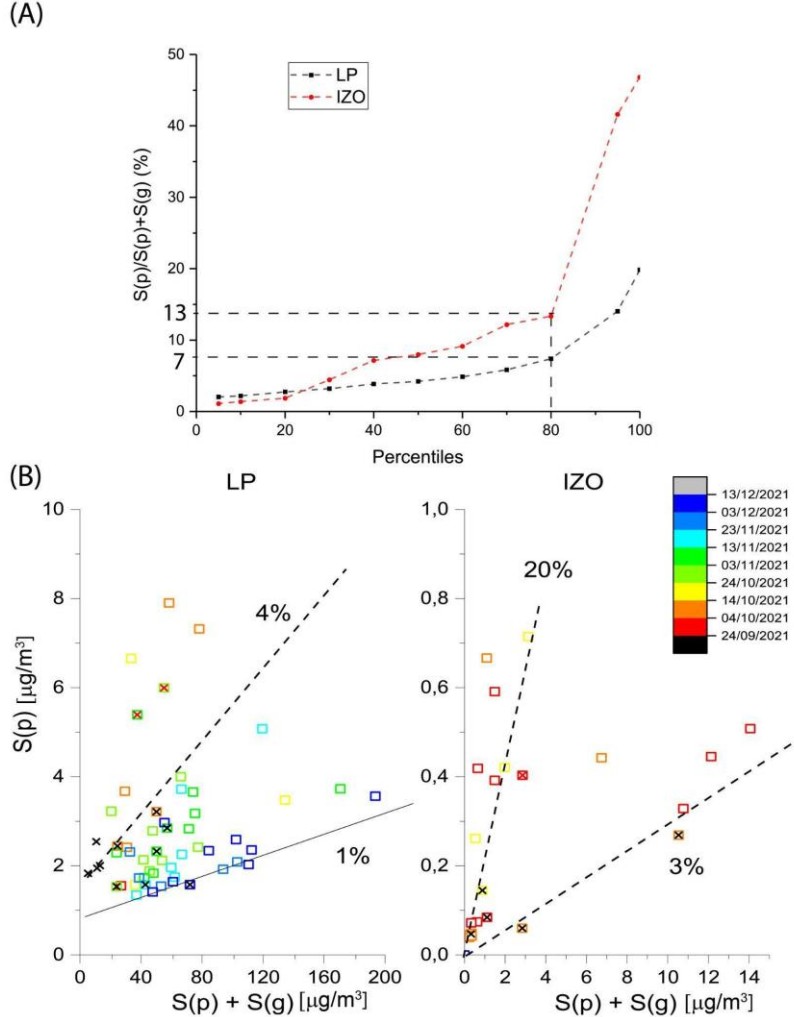

(A)

(B)

**Figure 6: Statistical distribution of the S(p) over S(p)+S(g) ratio (in percentage, p and g refers to particle and gas**
**respectively) measured at IZO (Tenerife) and at El Paso (La Palma) during the Tajogaite eruption. (A) shows the**
**statistical distribution of the conversion rate estimated from the La Palma and IZO aerosols measurements. (B)**
**reports S(p) as a function of S(p)+S(g) from the IZO and La Palma PM$_{10}$ analysis with the time as color scale. Crosses**
**inside the square denote data points coinciding with FTIR measurements.**

Furthermore, the time distribution of the S(p)/(S(p)+S(g) ratio (Fig. 6) suggests a higher conversion rate of SO$_2$
to sulfate during the first part of the eruption (until the beginning of November) compared to the second period.
This trend appears closely tied to the volcanic plume's altitude relative to the Trade Wind Inversion (TWI), as
described by Milford et al. (2023). During the first period of the eruption (until early November), plumes from
explosive activity and fountaining vents often rose above the TWI, and surface measurements at La Palma likely
captured older, dilute, more oxidized emissions from effusive vents trapped into the TWI. Conversely, from the
beginning of November, the entire plume, comprising both explosive and effusive components, was more
frequently trapped below the TWI, leading to the detection of younger, more concentrated, and less oxidized
emissions at ground level. In any case, the plumes reaching IZO are most likely dominated by explosive
emissions which, despite substantial transport times, exhibit oxidation rates below 15%. Such low conversion
rates would not produce resolvable differences in our gas-to-SO$_2$ ratios. The last two events in Table 5 present
some difference between both sites. On 16 October 2021, the FTIR and surface ratios at IZO are comparable,
highlighting their robustness, however, they are a factor of 2-3 lower than those reported by Burton et al. (2023).
We remark that for these days, the measurement target reported by these authors mention the base of the lava
fountain instead of the spattering vents or passive degassing, as for the other three dates, implying different
conditions and processes.

Finally, the $\Delta CO/\Delta CO_2$ ratios measured at FUE station (Fig. 4) and those recorded at IZO from surface measurements are on average higher and with higher variability than that recently reported in Asensio-Ramos et al. (2025) from open-path measurements (Fig. 4). This difference is likely due to the different measurement methods (solar absorption vs. open-path measurements using hot lava as source), implying different loci of measurements and gas contribution along their respective line of sight. Tajogaite volcano presented notable differences in eruptive behaviour between the different vents along the volcanic fissure, the higher elevated ones being more explosive than the lower ones. Recent studies suggest that eruptive dynamics may affect the abundance of redox-sensitive species (e.g.: Oppenheimer et al., 2018, Moussallam et al., 2019). Furthermore, we note that most of the Asensio-Ramos et al. (2025) measurement sites until the beginning of November (i.e: when our highest $\Delta CO/\Delta CO_2$ ratios were recorded) were located at the NNW from the eruptive fissure. With winds dominantly blowing towards the S and SW during this period, this configuration avoided a significant contribution of biomass and building burning plume to their measurements. It is not the case for the FUE measurements that were more likely to be affected by this contribution provoked by the advance of the lava flows. This hypothesis is also supported by the similarity of the $\Delta CO/\Delta CO_2$ time series at FUE with the time series of the areas covered daily by the advancing lava flows (Appendix D), reflecting the extent of burnt vegetation. The typical values reported in the literature for the wildfires (Yokelson et al., 2007; Akagi et al., 2014; Vasileva et al., 2017; Álvarez et al., 2023) are generally higher than our values, by at least a factor of 5 likely explained by the different contributors of the measured plume, i.e. a mixing of volcanic plume and vegetation/infrastructures burning in the case of the 2021 La Palma eruption.

## 5.2 New insight into the eruption dynamics

The ratios and emission flux time series as well as total emission estimates presented here provide some information about the degassing processes during the Tajogaite eruption.

Our time series of $SO_2$ volcanic emission fluxes confirms the decreasing trend observed from the $SO_2$ daily mass time series from Mounts (http://www.mounts-project.com, Valade et al., 2019) and reported in Milford et al. (2023). The concurrent decrease of $SO_2$ emissions together with that of decreasing tephra accumulation rates and decreasing plume height was suggested to reflect the decrease of the pressure in the plumbing system (Milford et al., 2023). This was confirmed by the co-eruptive deflation trend observed and inverted by Charco et al. (2024), possibly related to the pressure drop due to drainage of the reservoir. The relatively good fit of the $SO_2$ fluxes data obtained using an exponential function further supports this interpretation.

We found a good correlation between the $SO_2$ volcanic emission flux time series and the TADR (slope: 14.1 ± 1.2 kg of $SO_2$ m$^{-3}$ of lava and R=0.94). A similar correlation between $SO_2$ emissions and effusive volumes has previously been observed during the 2021 Fagradalsfjall eruption (Pfeffer et al., 2024). The few outliers to this correlation (empty circles, Fig. E2) occurred during three distinct periods: (1) the initial days of the eruption, coinciding with the peak in $SO_2$ emissions (2) just after the 27/09 eruptive pause, at the onset of sharp increase in effusion rates and (3) following the opening of the late November vents, north of the main vent alignment. These outliers correspond to abrupt changes in the output rate, likely associated with transient perturbations of the surface thermal structure-conditions known to affect the reliability of TADR estimations based on radiant density models (Coppola et al., 2016). Interestingly, applying the TADR values derived from the Pleiades-based volume estimates of Belart and Pinel (2022), which are averaged over 6-7 days, would bring at least three of these outliers back in line with the main trend. This suggests that apparent short-term imbalances between $SO_2$ emissions and effusion rates may be rapidly compensated, resulting in a coherent degassing-effusion relationship over multi-day timescales. This is particularly evident at the beginning of the eruption, where the Belart and Pinel (2022) estimates yield significantly higher TADR values than those of Plank et al. (2023) (Fig. E2).

Beyond these transient deviations, the correlation between $SO_2$ flux and TADR remains remarkably consistent throughout the eruption, suggesting that the emitted $SO_2$ predominantly reflects syn-eruptive magma degassing. This coherence, maintained over nearly three months of activity, indicates that the degassing regime remained stable once the eruption was fully underway. The early deviation from this trend, characterized by an apparent excess of $SO_2$ emissions relative to effusion, may reflect the release of sulfur that had already exsolved in the shallow system prior to the eruption and its rapid release, followed, after the eruptive pause, by the evacuation

of the partly degassed magma. While this interpretation is consistent with the observed trends (Fig. E2), it
remains tentative, given the absence of composition data for the earliest days of the eruption.
This correlation confirms that the emitted $SO_2$ only proceeds from the ascending magma. We observed a similar
behavior for HCl, HF and CO emission fluxes, which contrasts with the almost constant $CO_2$ flux throughout the
85 days of the eruption. This observation is fully consistent with the degassing model proposed by Burton et al.
(2023), which suggests a decoupling between $CO_2$ and $SO_2$ degassing processes. According to this model, a
$CO_2$-rich volatile phase, already exsolved in the upper mantle reservoir, could account for a large fraction of the
emitted $CO_2$ (up to ~80% according to Dayton et al., 2024), sustaining nearly constant $CO_2$ fluxes through the
system. This difference is partially reflected in the time series of the $\Delta CO_2/SO_2$ ratio that steadily increases from
the beginning of the eruption to the end of phase I, mimicking the trend of the VLP tremor amplitude. Such co-
evolution abruptly ends at the beginning of November, from when the ratio becomes more variable. The $CO_2$
volcanic emission fluxes being constant within uncertainties during the whole eruption and the $SO_2$ volcanic
emission fluxes being mainly controlled by the magma discharge rate, the steady increase of the C/S ratio during
the first part of the eruption thus reflects the progressive decrease of the proportion of shallow (discharge)
component relatively to the deep reservoir $CO_2$-rich fluids. In the frame of overall lower $SO_2$ fluxes due to
waning activity, the variability of the ratios of the phase II reflect the control of low $SO_2$ contents in the plume
and short-term variability of the $SO_2$ emissions.
The early November transition between phase I and phase II follows the apparition of new vents at the end of
October (Muñoz et al., 2022), interpreted as further propagation/opening of the underlying dike intrusion. This
transition shortly anticipates an abrupt and enduring drop in tremor amplitude (both VLP and LP frequency
bands; Bonadonna et al., 2022), geochemical changes (Ubide et al., 2023; Dayton et al., 2024) and hydrologic
and hydrochemical changes in the aquifer. The latter comprises e.g. an influx of pure (most likely endogenous)
$CO_2$ (Jimenez et al., 2024) that drastically increased the groundwater $HCO_3^-$ content at several sampling points
from 27 October 2021 (Amonte et al., 2022; Garcia-Gil et al., 2023b) or the establishment of a direct
relationship between the level in several groundwater wells and the tremor amplitude around 7 November 2021
(Garcia-Gil et al., 2023a). VLP tremor amplitudes are especially sensitive to variations in magma ascent
dynamics and conduit geometry (D'Auria et Martini, 2009; Bonadonna et al., 2022). Similar drops in VLP
tremor amplitude were observed at other volcanoes, such as at Piton de la Fournaise (Duputel et al., 2023) where
it was interpreted in terms of reduction of dyke dimension, heralding the end of the eruption. All these
observations suggest that these events at the beginning of November constitute a turning point in the eruption
implying significant structural changes in the plumbing system.
This turning point is particularly evident with the split described in the time series in the Sr isotopic
compositions of the matrix, and interpreted as the consequence of a deep-origin melt injection replenishing the
feeder system (Ubide et al., 2023). This interpretation further relies on this compositional change occurring in
close time relationship with an increase in the magnitude of seismicity, VLP tremor amplitude and a short-term
(5 days) rebound in the time series of daily $SO_2$ masses. We emphasize that the short-term increase in daily $SO_2$
masses observed between 28 October and 2 November 2021 should be interpreted with caution. First of all, at
the depth of injection, $SO_2$ being mostly soluble in magma until a few hundred meters depth (Burton et al.,
2023), any increase in $SO_2$ emissions would be due to an increase in lava discharge rate at the surface. Then, this
apparent peak coincides with a period of low wind speeds and a reversal in wind direction at 700 hPa (ERA5
data), which likely caused plume stagnation and gas accumulation. These meteorological conditions can lead to
an overestimation of $SO_2$ masses derived from satellite data. Therefore, we do not interpret this increase as a
definitive sign of enhanced volcanic degassing. The deep-origin melt injection at this period is further not
supported by the absence of corresponding signals in the GPS baseline time series (Charco et al., 2024), TADR
data (Plank et al., 2023), and our $CO_2$ fluxes and $CO_2/SO_2$ ratios.
The observed multiparametric transition in the eruption dynamics at the beginning of November could be
alternatively explained by a significant alteration of the magma pathway between the surface and the top of the
magma chamber. With the waning of the eruption, the ascent rate decreased and the conduit became more
unstable (Muñoz et al., 2022), with the opening of new vents from mid November (Gonzalez, 2022; Walter et
al., 2023), resulting in interaction with the aquifer, changes in the tremor amplitude, mixing ratio and/or
composition of endmembers and the return of radiogenic signatures.

### 5.3 Volatile mass balances and implications

Once released from the magma, volcanic gases suffer a number of processes such as oxidation, scavenging and dissolution in aqueous fluids that can alter their original composition before their detection. Integrating petrological constraints helps understanding volcanic degassing processes linking deep degassing to atmospheric observations and refining our understanding of element cycling and the environmental impact of volcanic plumes. We report here such an exercise estimating expected emissions estimated from petrological data, and compare them with our estimates derived from atmospheric measurements.

### 5.3.1 "Effective S degassing" and $SO_2$ mass balance

Combining our $SO_2$ volcanic emission fluxes and new petrological data, complementing literature, allow us to estimate a S degassing balance for the Tajogaite eruption. We used a similar Monte Carlo approach as proposed in Dayton et al. (2024), but refining the degassing balance as follows. We use an erupted lava volume of $(177 \pm 5.8) \times 10^6$ $Mm^3$ from Civico et al. (2022), a distal tephra volume of $(22.8 \pm 1.8) \times 10^6$ $Mm^3$ from Bonadonna et al. (2023) and a cone volume of $(36.5 \pm 0.3) \times 10^6$ $Mm^3$ from Civico et al. (2022). The total erupted mass is obtained applying a similar approach to Dayton et al. (2024), using densities of $2403 \pm 170$ kg.m$^{-3}$ for lava flows, based on an average percentage of vesicles for the erupted lava, 1800 kg.m$^{-3}$ for the cone and $1200 \pm 120$ kg.m$^{-3}$ for the tephra blanket (Bonadonna et al., 2022), resulting in a total erupted mass of $5.2 \times 10^8$ tons. S degassing from the magma is usually estimated from petrological data (difference between MI and matrix glass S contents), as in the mass balance of Burton et al. (2023) and Dayton et al. (2024) for the Tajogaite eruption. The observed correlation between the TADR and our $SO_2$ volcanic emission fluxes allows us to directly relate the degassed volume and the emitted S mass, with $14.1 \pm 1.2$ kg $SO_2$ emitted per "thermal" cubic meter of lava (lava volumes estimated using the radiant flux). We corrected this thermal volume for the tephra volume (blanket and cone) representing ~33% of the total emitted volume, because this does not participate significantly in the radiant flux. This resulted in $9.4 \pm 0.8$ kg degassed $SO_2$ per cubic meter of emitted lava, which converts into $2611 \pm 285$ ppm effective S degassing, considering above density and a correction of the crystal mass fraction (25% following Dayton et al., 2024). This value is very similar to that obtained by Dayton et al. (2024) using the difference between the S content of inclusions ($3062 \pm 500$ ppm) and matrix glasses ($345 \pm 53$ ppm). Note that the matrix S contents we present (average 534 ppm; N=52; σ=130 ppm; Supplementary Table S1) are consistent with previously published datasets for the eruption (average of 403 ppm; N=438; σ=10 ppm; Burton et al., 2023; Longpré et al., 2025). These data are nevertheless substantially higher than the value reported by Dayton et al. (2024). Using these values in the MonteCarlo degassing simulation of Dayton et al. (2024), the full degassing of 0.25 km$^3$ of magma would produce emissions of $1.93 \pm 0.21$ Mt $SO_2$. This is compatible with the TROPOMI-derived total $SO_2$ emissions ($1.81 \pm 0.18$ Mt).
A possibly unaccounted repository for initial S in the degassing balance could be the rare sulfide droplets, previously described to be present in the eruptive products matrix (Fig. B1; Day et al., 2022; Pankhurst et al., 2022) but also, more recently in clinopyroxene (CPx) cores and in magnetites (Andujar et al., 2025). These droplets separated from the silicate melt upon reaching the sulfide saturation during a pre-eruptive crystallization episode (Day et al., 2022), as confirmed by our own saturation calculations using the ONeil (2021) SCSS model (see Appendix B2). Importantly for the sulfur budget, although part of the primitive magma S content, as recorded in MI, the sulfur they contain is not included in matrix glass analyses (since it is physically segregated) and is not released as gas during eruption. The sulfide abundance could range between 0.03 vol.% (QEMSCAN quantification in Pankhurst et al., 2022) and 0.066 vol.% (0.001 mass fraction in the crystallizing assemblage in the models of Day et al., 2022). Assuming a density of 4500 kg·m$^{-3}$ (Saumur et al., 2015) and an average sulfur content of ~35% in the analyzed sulfides (Fig. B1), this range of abundance would represent a potential sulfide cargo in the erupted lava until day 20 (Day et al., 2022) of ~30 to 60 kt of non-degassed sulfur (equivalent to ~60 to 120 kt of $SO_2$). Accounting for this contribution would further improve the agreement between the petrologic budget (1.81-1.87 Mt of $SO_2$) and satellite-based estimates ($1.81 \pm 0.18$ Mt of $SO_2$).
Surprisingly, applying the same approach for the first week of the eruption (LU1 in Bonadonna et al., 2023) encompassing the TROPOMI-derived $SO_2$ emission peak, we observe a mismatch of a factor of 3 between the expected $SO_2$ degassing and that measured by TROPOMI. This arises from the very low thermal

lava volume (4.3 Mm$^3$ estimated using the radiant flux, corrected with the factor of 2 proposed by Plank et al.,
2023), which can be due to the transient time required for the surface thermal structure to become steady
(Coppola et al., 2016). Alternatively, using the cumulative volume of $43.0 \pm 6.1$ Mm$^3$ on the 26-09-2021
reported by Belard and Pinel (2022) and derived from multiple Pléiades stereoscopic surveys during the first
period of the eruption) and assuming a volume of $15 \pm 0.12$ Mm$^3$ for the edifice (Romero et al., 2022), we found
cumulated $SO_2$ emissions of about $580 \pm 66$ kt, which is closer to the TROPOMI-derived estimates for this
period (about 560 kt).

### 5.3.2 CO$_2$ mass balance and estimation of the reservoir volume

Applying the same MonteCarlo approach used for sulfur and assuming full $CO_2$ degassing, we estimate
that $\sim4.4 \pm 0.8$ Mt of $CO_2$ would have been released from the erupted material alone. This is consistent with the
estimate of $5.4 \pm 1.0$ Mt by Dayton et al. (2024). However, plume measurements indicate significantly higher
total $CO_2$ emissions during the eruption, amounting to $19.4 \pm 1.8$ Mt. This discrepancy, combined with the near-
constant fluxes throughout the eruption, supports the presence of a $CO_2$-rich fluid phase in the reservoir
(Hansteen et al., 1998; Burton et al., 2023) coexisting with a $CO_2$-saturated melt, capable of contributing an
additional 15 Mt of $CO_2$. Based on FI densities reported by Dayton et al. (2023), we estimate that this additional
15 Mt of $CO_2$ corresponds to a fluid volume of $\sim25$-17 Mm$^3$ at the pressure of the shallow (deflating) reservoir
pressure and at that of the deeper reservoir, respectively.
The $\sim1\%$ pressure drop relative to the pressure at the beginning of the eruption observed by Charco et
al. (2024) provides an opportunity to derive a first-order constraint on the volume of the deflating reservoir.
Assuming this pressure loss is attributed to a volume change due to magma extraction, we can estimate the total
volume of "hydraulically" connected magma/mush feeding the eruption.
To estimate the total volume (magma+fluid) extracted from the reservoir, 1) we corrected the eruptive products
volume for vesicularity (Dense Rock Equivalent or DRE volume, taking as a reference a melt density of $\sim2700$
kg.m$^{-3}$; see previous section and Dayton et al., 2024), 2) we added the volume of the magma-filled dykes and sill
network (as described by De Luca et al., 2022) and 3) finally, we corrected for the effect of magma
compressibility. According to Rivalta and Segall (2008), the volume ratios (intrusion/associated reservoir
deflation) necessary to estimate magma compressibility range between 1.2 and 7.7. For the Tajogaite eruption,
the most likely value is $\sim5$ (reservoir from 10 to 15 km deep, saturation depth >25 km; see Fig. 3 in Rivalta and
Segall, 2008). Using such values for correcting our magma volume and adding our extracted (additional) fluid
volume estimate allows estimating a total volume (magma + fluids) extracted from the reservoir of $\sim60$ Mm$^3$
(from 45 to 200 Mm$^3$ for the full range of volume ratios). Considering the extraction of this volume produced
the pressure drop in the deflating reservoir, we roughly estimate the volume of magma/mush to equate, at least
to 6 km$^3$ (4-20 km$^3$ range). This estimate provides a first-order volume of magma/mush that could have been
"hydraulically" connected to the surface during the eruption. It includes at least the shallow reservoir, but may
also encompass deeper zones of the plumbing system if they were effectively connected during the eruptive
episode.

### 5.3.3 Halogens mass balance

Fluorine and chlorine generally have high solubility in magmas and only begin to exsolve at shallow
depths, close to the fragmentation level (e.g.: Aiuppa, 2009). This is likely the case for the 2021 La Palma
eruption, where rapid magma ascent (Romero et al., 2022; Boneschi et al., 2024) limited halogen degassing due
to kinetic constraints. As a result, the melt retained most of its original halogen content, and the difference
between melt inclusions and matrix glass Cl and F contents is hardly resolvable from analytical uncertainty
(Dayton et al., 2024). We thus assessed the consistency of our fluxes using another approach, estimating the
expected Cl and F degassed amounts from the total observed emissions.
The adsorption of halogen-derived salts onto ash surfaces is likely to be a non-negligible sink for
hydrogen halides of the volcanic plume (Bagnato et al., 2013) and should be considered in our balance. We thus
propose a rough estimate of the scavenged halogen mass using the median (and standard error) content of Cl
($335 \pm 34$ ppm) and F ($422 \pm 49$ ppm) from a compilation (N=57) of published lixiviation experiments
(Ruggieri et al. 2023; Sanchez-España et al., 2023; Rodriguez et al., 2025, submitted) and the mass of tephra
emitted throughout the eruption (Bonadonna et al., 2022), including the cone (Civico et al., 2022). We obtain
estimates of $31 \pm 8$ kt HCl and $39 \pm 11$ kt HF possibly scavenged from the plume, that we need to sum to our
measured HCl and HF budgets ($49 \pm 12$ kt and $13 \pm 2$ kt of HCl and HF, respectively), giving surface emissions
of $80 \pm 15$ and $52 \pm 11$ kt for HCl and HF, respectively.
Using the average Cl and F contents in MIs of Dayton et al. (2024), these emissions can be explained
with Cl and F losses of ~195 and 130 ppm from the melt, respectively. This is 35% and 9% of the initial melt
content in Cl and F, respectively. This Cl difference should be resolvable analytically, but the F difference is
indeed within the analytical uncertainty of electron microprobe for volcanic glasses (Rose-Koga et al., 2021)
and at the limit of that for the Secondary Ion Mass Spectrometry analyses of Dayton et al. (2024). We propose a
complementary estimation of the Cl loss from the melt using petrological data of the MI and matrix glasses of
the eruption (Burton et al., 2023; Dayton et al., 2024; Longpré et al., 2025). The determination of the amount of
Cl degassing from the melt is indeed obscured by the magma evolution in the plumbing system, as shown by the
bivariate diagram between $K_2O$ and the Cl contents (Fig. B3), where the matrix glass Cl contents are
consistently higher than that of MI, impeding simple quantifications by difference as for S balance. In this
diagram, MIs define a trend (Pearson's R=0.943) that can be used to estimate the average Cl amount degassed
from magma. We find an error-weighted mean Cl content difference between the simulated undegassed magma
compositions and the matrix glasses of $189 \pm 10$ ppm (95%; N=633; $\sigma$=135), within uncertainties of our
degassing balance approach. The total HCl emissions that would arise from such degassing from the volume of
eruptive products is $77 \pm 7$ kt, indistinguishable from our HCl balance of $80 \pm 15$ kt within uncertainties. This
approach is not possible for F due to significant variability in MI F contents.

### 5.4 Potential atmospheric implications of Tajogaite eruption emissions

Volcanic emissions of greenhouse gases and reactive species represent critical inputs for climate
models, as they contribute to baseline radiative forcing, perturb the oxidative capacity of both the troposphere
and stratosphere, influence aerosol–cloud microphysical interactions, and play a significant role in the
geochemical cycling of key elements such as sulfur, carbon, and halogens between the Earth's surface and
atmosphere (Von Glasow et al., 2009 and references therein). Accurate quantification of these natural fluxes is
essential for distinguishing anthropogenic signals from background variability in atmospheric composition. The
Tajogaite eruption provides a striking example of how a single volcanic event can temporarily dominate
regional atmospheric budgets. Its $SO_2$ emissions were approximately 15 times greater than Spain's total
anthropogenic $SO_2$ emissions for the year 2021 (123 kt; MITECO, 2023), and even exceeded the total EU
anthropogenic $SO_2$ emissions for that year (1.4 Tg; EEA, 2023). Assuming a conservative 20% conversion rate
of S to sulfate aerosols (see Section 5.1 and Fig. 6), the eruption is estimated to have produced approximately
0.5 Mt of sulfate particles. However, since the plume remained below 8 km altitude, well within the troposphere,
these aerosols were likely short-lived and regionally confined, with limited potential to affect atmospheric
radiation budgets, and a negligible climatic forcing from aerosol loading. In terms of carbon emissions, the $CO_2$
released by the eruption amounted to approximately 10% of Spain's anthropogenic $CO_2$ emissions for 2021
(https://www.miteco.gob.es/es/calidad-y-evaluacion-ambiental/temas/sistema-espanol-de-inventario-sei-
/informe-interactivo-inventario-nacional-emisiones-atmosfera.html). Emissions of CO were also substantial,
corresponding to about 7% of the 2021 national anthropogenic CO inventory (1.64 Mt; MITECO, 2023).
Halogen emissions were particularly notable. The total atmospheric HCl output was around ten times higher
than the annual UK emissions since 2017 (UK National Atmospheric Inventory) and represented roughly 20%
of total European anthropogenic HCl emissions in 2014 (220 kt; Zhang et al., 2022), which are primarily
associated with the energy sector (38%) and open waste burning (23%). Similarly, the eruption's HF
atmospheric emissions exceeded UK national totals for the same period by an order of magnitude. In contrast to
the purely atmospheric pathway, a significant fraction of the halogens was likely scavenged from the plume by
ash particles. This process, which accounts for an estimated $31 \pm 8$ kt of HCl and $39 \pm 11$ kt of HF, provides a
distinct mechanism for their re-entry into the geosphere through ash deposition. Subsequently, these ash-bound
halogens are remobilized by initial rainfall events (Medina et al., 2025), where they can enter and be transported
through natural elemental cycles of Cl and F in soil, aquifer, and marine environments.

Volcanic emissions of chlorine are known to significantly influence tropospheric ozone ($O_3$), as these
halogens participate in catalytic cycles that destroy $O_3$, particularly in the presence of sunlight and moisture
(Gerlach, 2004). Studying these emissions allows assessing the chemical forcing of volcanoes on the
troposphere, test atmospheric chemistry models using real events, and improve our understanding of the climatic
and chemical role of volcanic eruptions, even moderate ones like Tajogaite. In this study, we looked for signs of
such an impact in the local $O_3$ total column from FTIR spectroscopy but found no clear evidence. However, Fig.
7 displays the time series of $O_3$ partial pressure up to 8000 m, retrieved from electrochemical concentration cell
(ECC) ozonesonde measurements conducted by AEMET (García et al., 2021) from Puerto de la Cruz, Tenerife.
These are shown together with cumulative $SO_2$, HF, and HCl emissions up to 18 November 2021, corresponding
to the period with the most continuous and densely sampled flux measurements. A noticeable coincidence was
observed between the two sharp increases in the cumulative HCl (and HF) emissions, occurring on the
18/10/2021 and 02/11/2021 and local ozone depletion (with $O_3$ values near zero) at plume altitudes. This
coincidence is not observed for low HCl/$SO_2$ ratios.

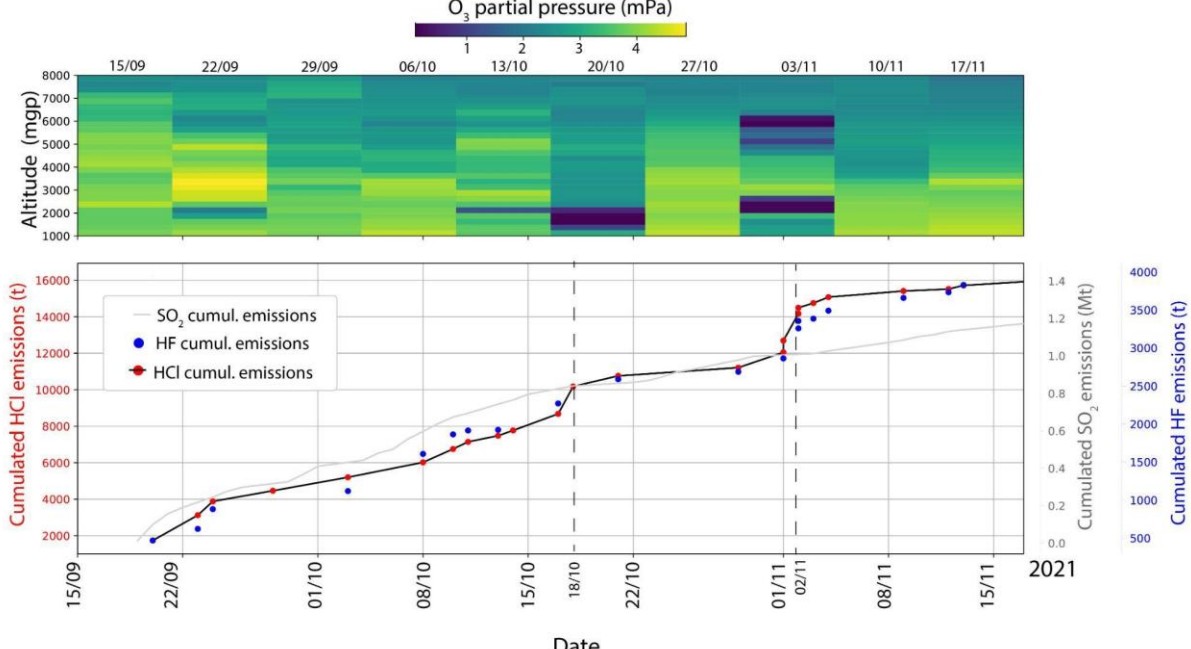

**Figure 7: Relationship between the partial pressure of $O_3$ at the volcanic plume altitude (from AEMET radiosonde**
**data) and the volcanic gas fluxes. $SO_2$, HCl, and HF emissions are shown as cumulative curves to highlight key**
**temporal variations. The $O_3$ partial pressure is derived from radiosonde measurements conducted at the AEMET**
**Tenerife station.**

Although geometric constraints and weather conditions affected the continuity of our cumulative flux estimates,
this preliminary observation suggests that halogen-induced $O_3$ loss may occur locally, at least where the plume
was present, even if only transiently. The observed $O_3$ loss appears to be short-lived, with concentrations
recovering shortly after, arguing against a persistent or widespread effect. To better assess the intensity and
duration of this impact, more continuous time series and refined flux retrievals are required. Nonetheless, this
initial evidence from the Tajogaite eruption provides a valuable basis for future investigations.

**6. Summary and Conclusion**

In this study, we explored the variability of the chemical composition of the Tajogaite eruption
volcanic gas plume by combining ground-based FTIR and UV direct-sun measurements with surface gas
observations at two sites: Fuencaliente, on La Palma, and the Izaña high-altitude Atmospheric Observatory, a
reference station for atmospheric studies located in Tenerife. New retrieval methods are presented to derive the
HF and HCl volcanic contribution in the total columns obtained from the solar FTIR spectra for both low
(EM27/SUN) and high (IFS-125HR) spectral resolution measurements performed up to 140 km from the
eruptive fissure. The good agreement between the different products (total columns and ratios) obtained from

the different instruments (FTIR, DOAS and surface measurements) demonstrates the robustness of our results, even at such distant and low-concentration locations as the 140 km-far IZO Observatory. Our compositional ratios measured during the eruption are also consistent with the limited data reported in the literature (Asensio-Ramos et al., 2025; Ericksen et al., 2024; Burton et al., 2023), including for previous basaltic eruptions in the world (e.g. Aiuppa et al., 2009). We derived $SO_2$ volcanic emission fluxes from the TROPOMI data and assessed the long-term variability of the emission fluxes of the other volcanic species, based on our compositional data. We found total emissions of $CO_2$, $SO_2$, HCl, HF and CO of 19.4±1.8, 1.8±0.2, 0.05±0.01, 0.013±0.002 and 0.123±0.005 Mt, respectively. These emissions were found to be non-negligible in the annual Spanish national and European inventory balance compared to anthropogenic emissions. Furthermore, while the $SO_2$ and halogen halides emission fluxes decreased throughout the eruption along with the lava emission fluxes, the $CO_2$ emission fluxes were found to be almost constant, implying a comparatively increasing discharge with respect to the daily emitted lava volumes. This is consistent with a significant amount of $CO_2$ being already exsolved in the reservoir, as previously observed by Burton et al. (2023), Dayton et al. (2023) and Dayton et al. (2024). Global degassing balances were performed for C, S, Cl and F, showing a good consistency between the plume measurements and the petrological data. This study highlights the potential of employing existing global atmospheric FTIR, DOAS and surface measurement networks to explore remotely (>100 km) the variability of volcanic plumes chemical composition and its implications at different timescales. By demonstrating their effectiveness in tracking volcanic emissions in real time, our findings underscore the value of these networks for both operational volcano monitoring and scientific investigations during and after eruptive crises. Such measurements are crucial for assessing the role of volcanic emissions as natural sources in the global cycling of carbon, sulfur, and halogens. This study emphasises the value of solar absorption measurements for volcanology, atmospheric research, and air-quality monitoring during eruptions, and suggests their potential application during major eruptions even when access is more restricted.

## 7. Appendices

### Appendix A

**Comparison between the new HF and HCl products derived from the IFS-125HR and EM27/SUN measurements**

The appendix A provides a summary of the comparison between the standard NDACC FTIR HCl and HF products and those developed in this study (Fig. A1), the new IFS-125HR $SO_2$ retrievals (Fig. A1c), as well as the comparison between all the IFS-125HR and EM27/SUN products (Fig. A2). As illustrated by the comparison (Table A1), the standard and optimised approaches show an excellent agreement under background conditions with a mean bias of approximately 3% and 15% for HCl and HF, respectively, while the scatter is limited to 4% for both trace gases. These values fall within the expected uncertainty estimations of the IFS-125HR products (García et al., 2021). However, for volcanic emissions, the NDACC approaches, in contrast to our optimized approach, are not able to capture the volcanic HCl and HF contributions in the lower/middle troposphere, resulting in a mean difference of 88% and 100% for HCl and HF, respectively. Column enhancements as large as $6.00 \times 10^{20}$ and $2.05 \times 10^{20}$ molec/m$^2$ for HCl and HF, respectively, were reported during the volcanic process, which accounts for the high variability observed between mean, median and scatter values under volcanic emissions.

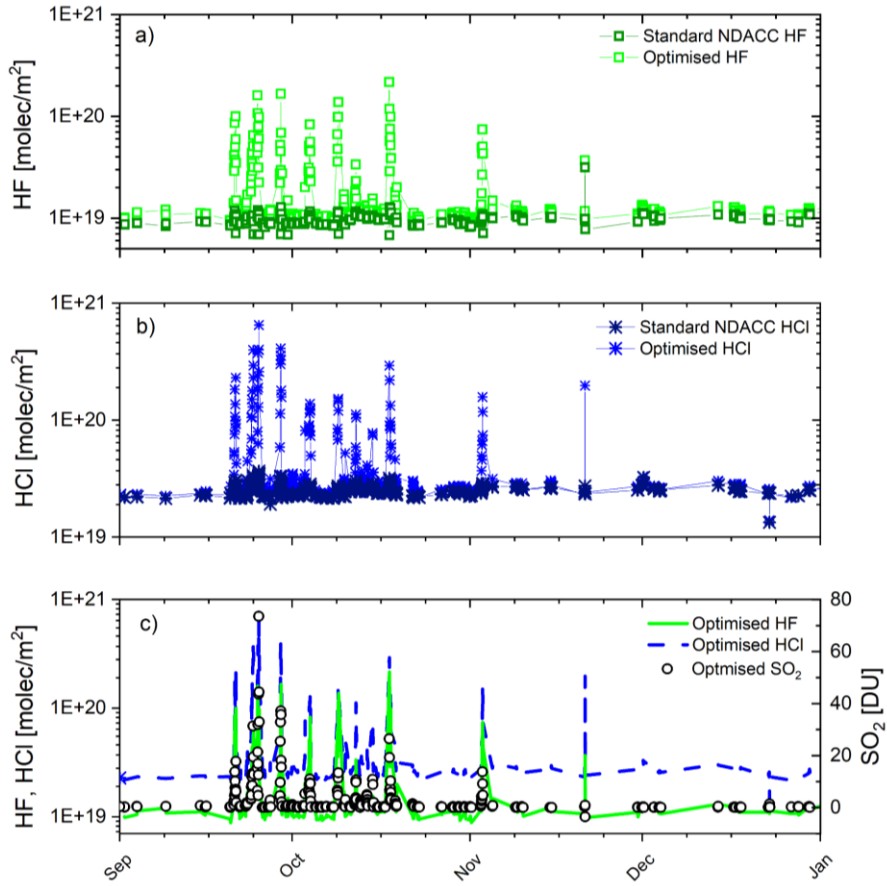

1005

**Figure A1: Time series of the standard NDACC and optimised HF (a) and HCl (b) total column amounts measured at IZO from the IFS-125HR instrument between 1 September and 31 December 2021. (c) Time series of the optimised HF, HCl and SO$_2$ IFS-125HR products at IZO for the same period.**

Figure A1 also presents the SO$_2$ total column amounts retrieved from the IFS-125HR measurements at IZO. The excellent agreement found between the SO$_2$, HCl and HF retrievals consistently capturing volcanic plumes probes the reliability and quality of the optimised IFS-125HR products, which has been also documented by side-by-side Pandora and FTIR SO$_2$ observations (Taquet et al. 2023). As found for the NDACC FTIR sites of IZO and Altzomoni (García et al., 2022), the Pandora and FTIR comparison shows an excellent correlation for the whole SO$_2$ range observed (Pearson correlation coefficients larger than 0.99) and the scatter between techniques is comparable to background signal (less than 0.7 and 2.0 DU for IZO and Altzomoni, respectively). For further details about SO$_2$ IFS-125HR retrieval refer to García et al. (2022).

**Table A1: Summary of the comparison between the standard NDACC and optimised HF and HCl total column amounts measured at IZO from the IFS-125HR instrument for the period (1) 1-15 September and 1-31 December 2021 under background conditions; and (2) between 19 September and 31 November affected by volcanic emissions. N stands for the number of measurements and STD corresponds to the standard deviation of the data distribution.**

| | Background conditions (1-15 September & 1-31 December, N=68 for HCl and N=67 for HF) | | | Volcanic emissions (19 September - 31 November, N=414 for HCl and N=405 for HF) | | |
|---|---|---|---|---|---|---|
| IFS-125HR Products | Mean | Median | STD | Mean | Median | STD |
| NDACC HCl [molec.m$^{-2}$] | 2.47E19 | 2.50E19 | 1.46E18 | 2.47E19 | 2.40E19 | 2.69E18 |
| Optimised HCl | 2.55E19 | 2.57E19 | 2.02E18 | 5.07E19 | 2.7E19 | 6.86E19 |

| [molec.m⁻²] | | | | | | |
|---|---|---|---|---|---|---|
| NDACC - Optimised HCl [molec.m⁻²] | 8.11E17 | 7.49E17 | 9.79E17 | 2.60E19 | 2.88E18 | 6.66E19 |
| NDACC - Optimised HCl [%] | 3.2 | 3.0 | 3.9 | 88 | 12 | 204 |
| NDACC HF [molec.m⁻²] | 9.92E18 | 9.77E18 | 6.85E17 | 9.51E18 | 9.19E18 | 1.46E18 |
| Optimised HF [molec.m⁻²] | 1.14E19 | 1.14E19 | 6.95E17 | 1.95E19 | 1.12E19 | 2.34E19 |
| NDACC - Optimised HF [molec.m⁻²] | 1.50E18 | 1.45E18 | 3.64E17 | 1.00E19 | 1.68E18 | 2.32E19 |
| NDACC - Optimised HF [%] | 15.2 | 14.9 | 4.1 | 100 | 18 | 288 |

Figure A2 shows the comparison between the new HCl and HF products (HCl_v2 and HF_v2) derived from the
EM27/SUN measurements, as well as those (HCl_v1 and HF_v1) estimated using the same retrieval as Butz et
al. (2017) and the new products derived from the IFS-125HR measurements at IZO. Fit parameters are
presented in Table A2. An excellent correlation was found for the newly developed EM27/SUN products
(HCl_v2 and HF_v2), highlighting the improvements of the retrieval methods especially in case of far
measurement sites.

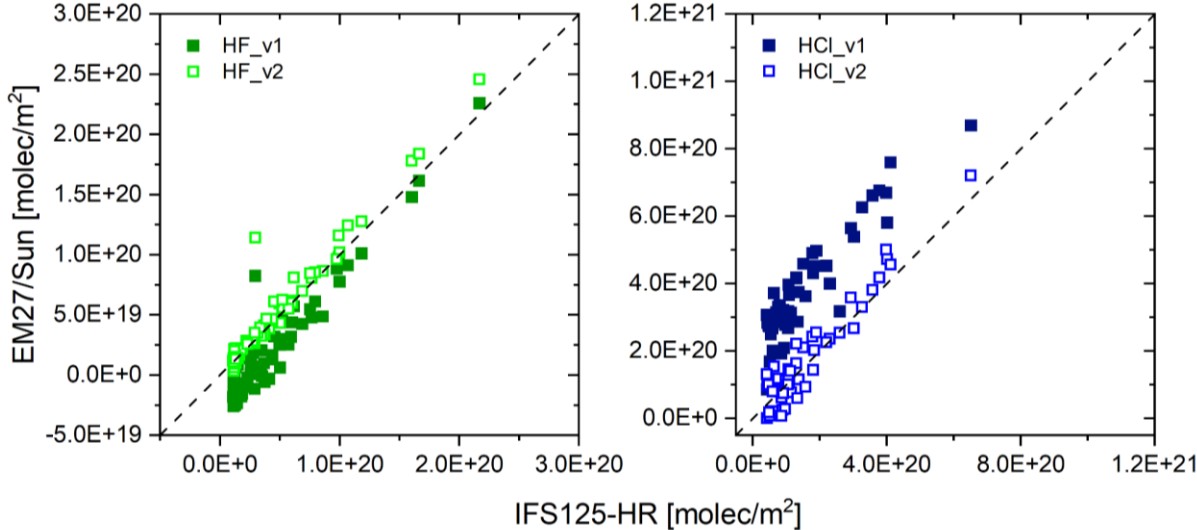

**1028 Figure A2: Intercomparison between the IFS-125HR and EM27/SUN HF and HCl total columns obtained from side-**
**1029 by-side measurements during the Tajogaite eruption. The diagonal (y=x) is plotted as a dashed line. The fit**
**1030 parameters from linear regression are given in Table A2.**

**1031 Table A2: Fit parameters obtained from the linear regression between the EM27/SUN HF and HCl products and**
**1032 those from the IFS-125HR.**

| EM27/Sun (10 min average centered in the IFS125-HR measurements) vs. IFS-125HR | Linear regression using least squares fitting method |
|---|---|
| HF_v1 | Slope=1.11±0.03 |

| | Offset=(-2.7±0.2)×10$^{19}$ molec/m$^2$ <br> R=0.91 |
|---|---|
| HF_v2 | Slope=1.11±0.025 <br> Offset=(-2.0±11.8)×10$^{17}$ molec/m$^2$ <br> R=0.98 |
| HCl_v1 | Slope=1.14±0.07 <br> Offset=(20.4±1.5)×10$^{19}$ molec/m$^2$ <br> R=0.92 |
| HCl_v2 | Slope=1.11±0.05 <br> Offset=(-0.43±1.07)×10$^{19}$ molec/m$^2$ <br> R=0.95 |

1033

**Procedure for removing CO and CO$_2$ background to the total column estimates and extracting volcanic contribution**

CO and CO$_2$ analysis from the total column measurements require the simulation and removal of the background concentration. Figures A3 and A4 show the procedure we employed for this study, through two examples from the FUE and IZO dataset. The time series is first detrended from the annual cycle, using IZO long-term time series and a third degree polynomial. The intraday variability of CO$_2$ background is then simulated using the average of the XCO$_2$ intraday time series from spectra without volcanic plume contribution. The background contribution is simulated for each day, fitting an offset. An example of the total procedure is illustrated in Fig. A3 and A4 for CO$_2$ and CO, respectively. The simulated background at FUE (in red, A) was compared with IZO measurements (in blue, A) when it was not affected by the volcanic plume. The resulting intraday time series of $\Delta$XCO$_2$ (Fig A3C and A3D) and $\Delta$XCO (Fig A4C and A4D) in presence of volcanic plume are well correlated with the $\Delta$HCl, which can be considered as a tracer of the volcanic plume.

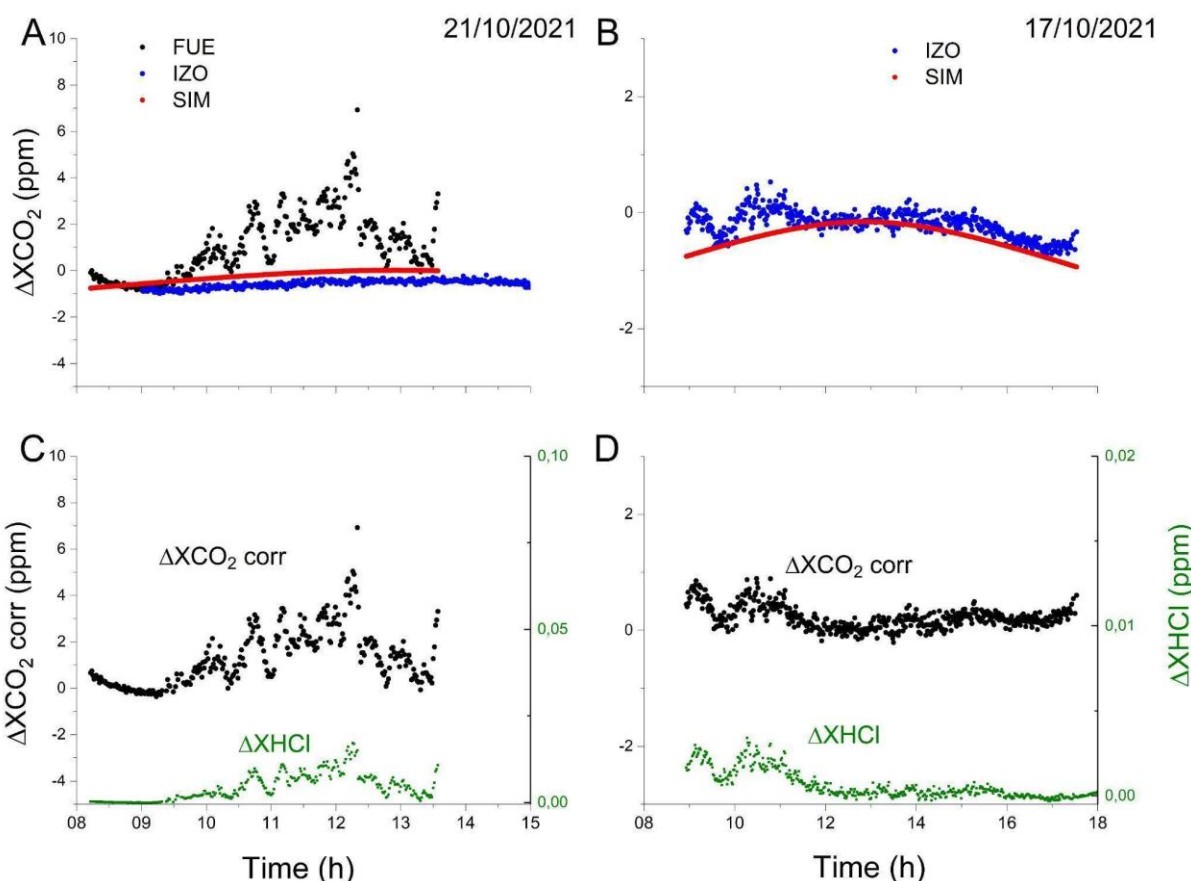

**Figure A3: Procedure for removing the atmospheric background contribution to estimate the CO$_2$ and CO abundance in the volcanic plume. A and B show a typical example of uncorrected $\Delta$XCO$_2$ at FUE (in black) and IZO (in blue) and the corresponding simulated background (red). The background simulations (in red) obtained using the average diurnal pattern without presence of volcanic plume and adjusting offset is compared with the measurements**

**taken at the IZO station (blue) on the same day. The corrected $\Delta XCO_2$ is presented in black in (C) and (D)**
**concurrently with the $\Delta XHCl$, which can be considered as a tracer of the volcanic plume.**

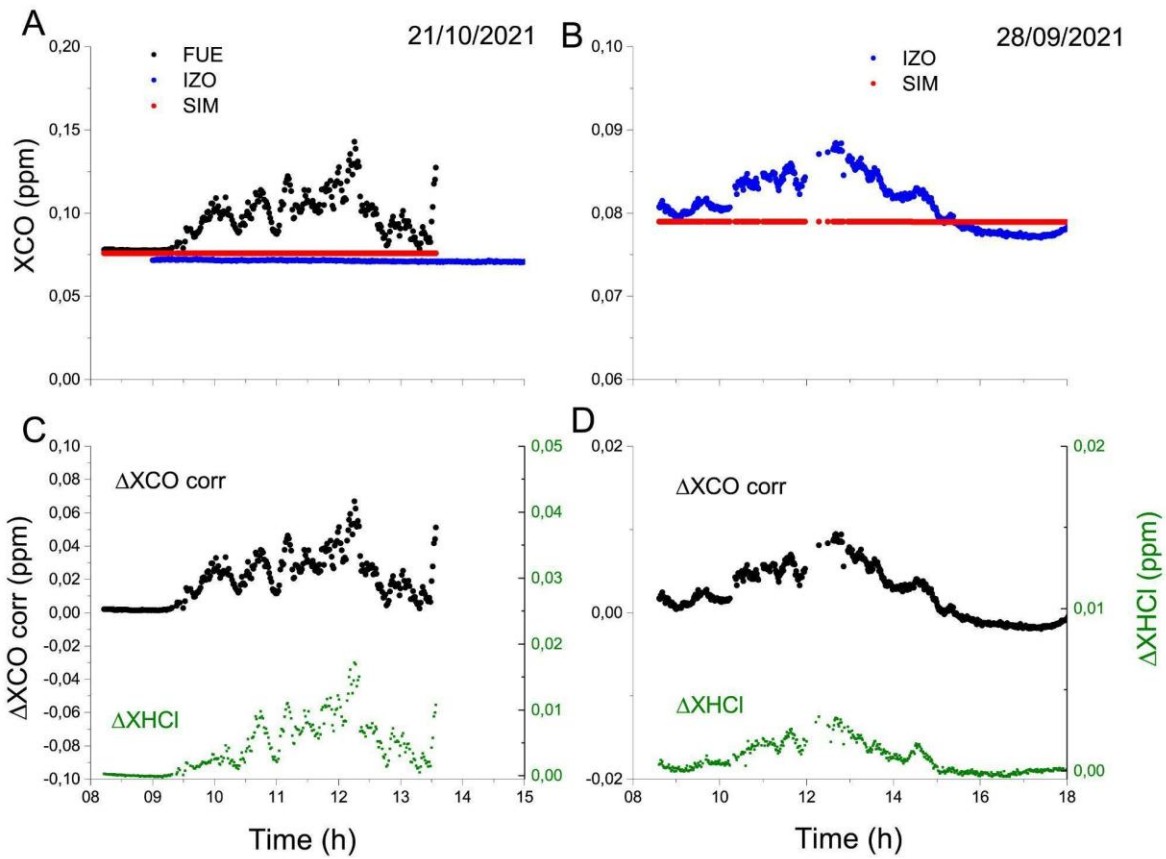


**Figure A4: Same as A3 for CO. For CO the procedure only consisted in removing the long-term trend, estimated**
**from the long-term IZO daily average time series.**

## Appendix B

Appendix B presents the tephra compositions acquired with Scanning Electron Microscope and Electron Micro-Probe and elements of the petrologic approaches used in for the estimation of the volatile emissions.

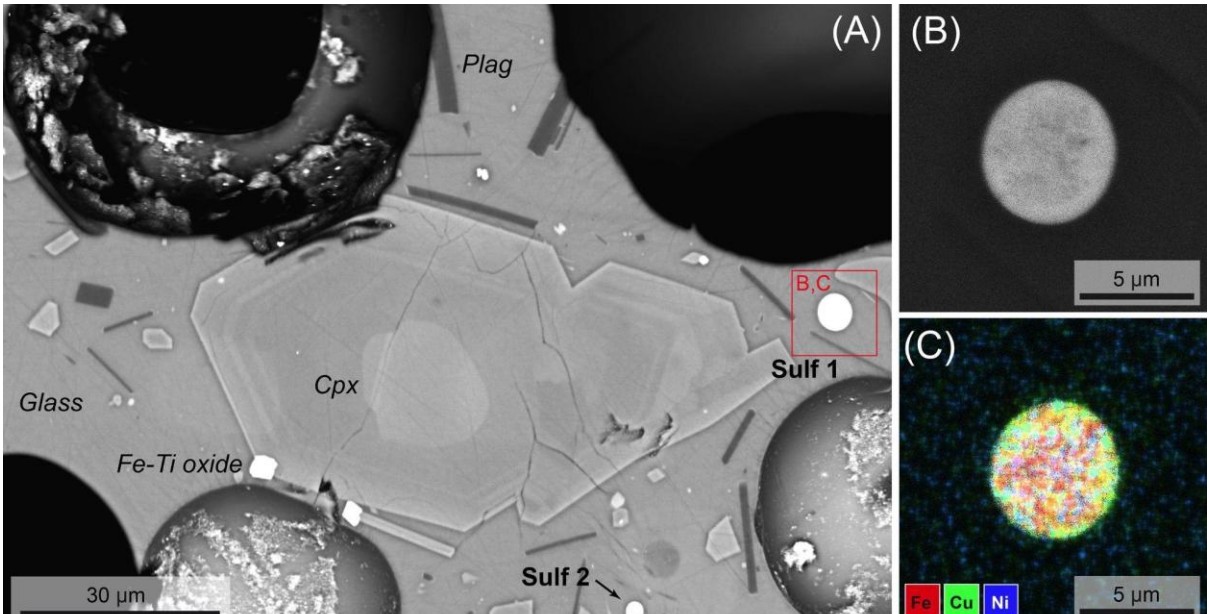

| | S | Fe | Ni | Cu |
|---|---|---|---|---|
| Sulf1 | 35.7 | 54.2 | 2.9 | 7.3 |
| Sulf2 | 34.6 | 53 | 2.8 | 9.7 |

**Figure B1: (A)** Backscattered electron (BSE) image of a section of LM-2309 tephra sample (Gonzalez-Garcia et al., 2023) displaying two sulfide droplets (Sulf-1 and Sulf-2). **(B)** Detail BSE image of Sulf-1, and **(C)** EDX compositional map of Sulf-1, showing zoning in Fe-Cu-Ni sulfides. The images were acquired using (A) a JEOL JSM-7610F gun emission scanning electron microscope operating at 15 kV (IESW, Hannover) (A), and a TESCAN Vega 4 operating at 20kV with EDX Bruker detectors (UCM, Madrid) (B, C). The table below shows their compositions (in wt%), determined by energy dispersive spectroscopy with a Cameca SX-100 electron microprobe (Uni. Bremen).

**Figure B2:** The upper panel shows the results of the sulfur content at sulfide saturation (SCSS) calculations performed using the model of ONeill (2021) implemented in the open-source Python3 tool PySulfSat (Wieser and Gleeson, 2023). The starting composition is one of the most primitive MI of the literature dataset for the eruption (LM0 G29 Dayton et al. 2024), to which a Petrolog3 (Danyushevsky and Plechov, 2011) crystallization model (with olivine ±clinopyroxene + spinel as crystallizing phase, following Day et al. 2022) is applied at a magma stalling at 3.5 kbars and a $fO_2$ buffer of NNO+0.4, following Andujar et al. (2025). Given these conditions, the melt is expected to contain a significant proportion of sulfur as sulfate ($S^{6+}$), rather than sulfide ($S^{2-}$). Therefore, we used the SCSSt model of Jugo et al. (2010), which accounts for mixed sulfur speciation, to evaluate saturation. Only a few inclusions slightly exceed the SCSSt curve, consistent with the rarity of sulfide globules in the eruptive products and with the interpretation that sulfide saturation was only reached locally or after some crystallization (Day et al., 2022). The bottom panel shows the modeled composition (Fe/Fe+Ni+Cu) of the sulfide phase precipitating along the liquid line of descent, which is matching the measured compositions between ~4 and 5.8 wt% MgO (after 5-15% crystallization). This range is reported as the orange portion of the liquid line of descent in the upper panel.

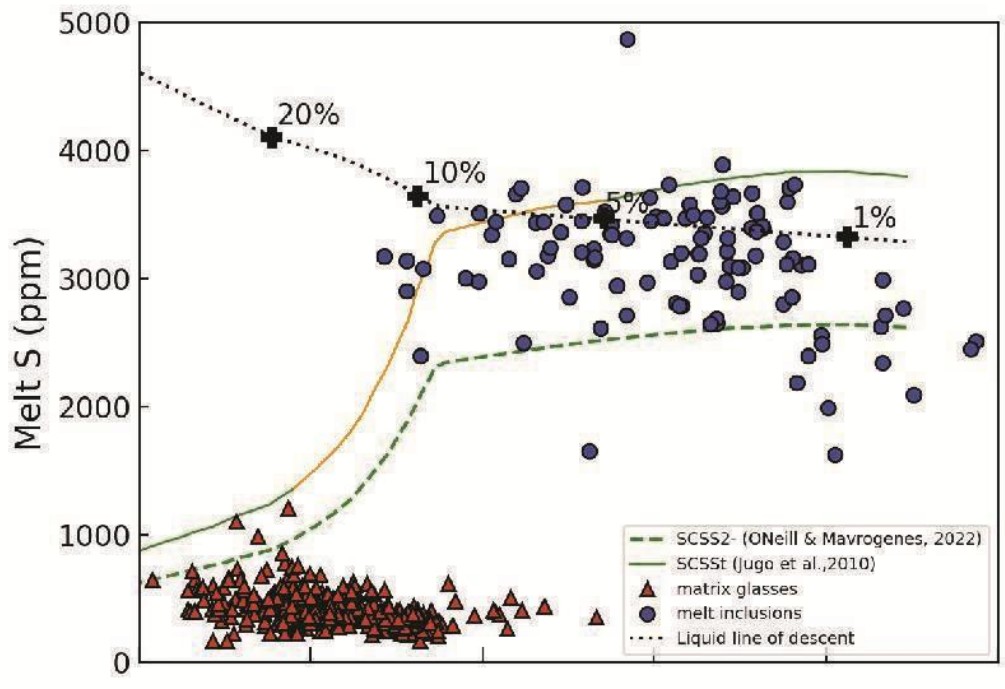

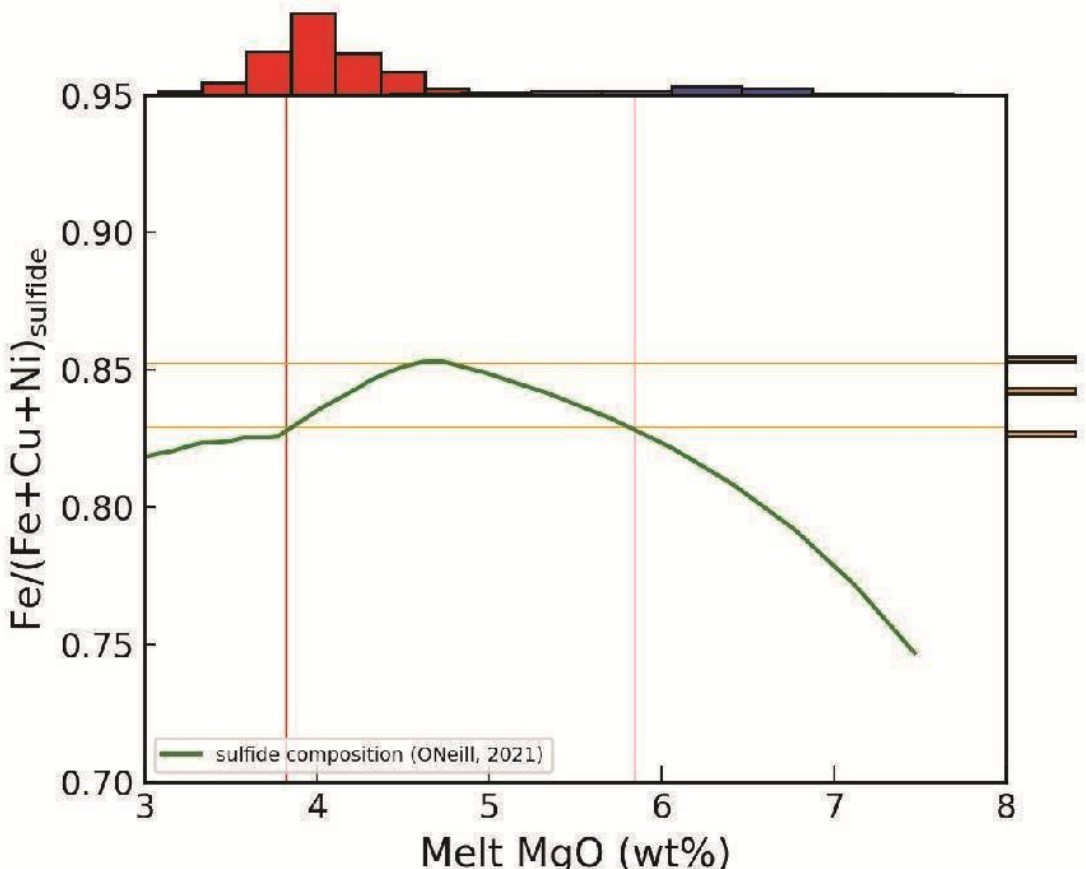

1080

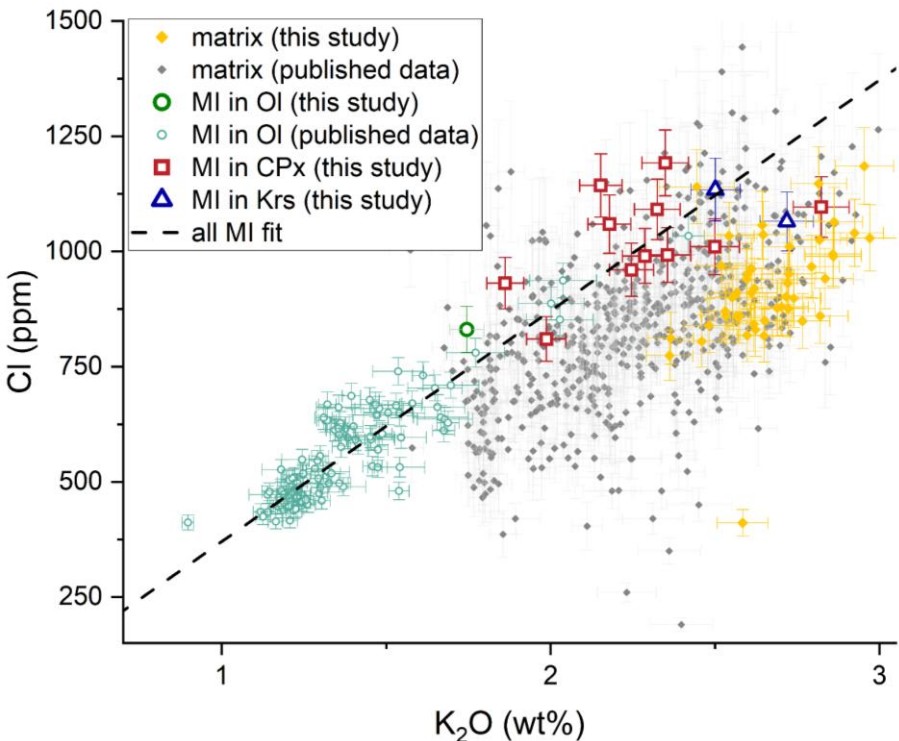

**Figure B3: New and previously published Cl content in MIs (green hollow circles) and matrix glass (orange diamonds) with melt evolution, represented by the incompatible K₂O. MI compositions are from this study, Burton et al. (2023) and Dayton et al. (2024). Ol stands for olivine, CPx, for clinopyroxene and Krs for Kaersutite. Matrix glasses have been measured on tephra samples and are from this study, Burton et al. (2023), Ubide et al., 2023, Dayton et al. (2024), Longpré et al. (2025). The dashed line is the linear regression through the MIs dataset and represents the increase in melt Cl content during crystal fractionation. We calculated the degassed Cl amount as the error-weighted mean difference between each matrix Cl content and the regression line.**

## Appendix C

Appendix C details the procedure that we employed to estimate the plume age for the dates reported in Table 5, using the Hysplit forward simulations.

We used high resolution meteorological data derived from the WRF-AWR (Advanced Research Weather Research and Forecasting) model (Powers et al., 2017; Skamarock et al., 2019). The WRF-ARW model is run operationally twice daily, utilizing initial and boundary conditions from HRES-IFS (High Resolution Integrated Forecast System) data provided by ECMWF (European Centre for Medium-Range Weather Forecasts) at a resolution of $0.09° \times 0.09°$. The model configuration includes three nested domains with horizontal resolutions of 6 km, 2 km, and 1 km, respectively, and 31 vertical levels, operating in non-hydrostatic mode. Each simulation produces forecasts extending up to 72 hours. The outputs of the WRF-ARW model are converted into the required format for the HYSPLIT model (Hybrid Single-Particle Lagrangian Integrated Trajectory) using the ARW2ARL program. This process produces meteorological data formatted for use in trajectory and dispersion simulations. To prepare the data for HYSPLIT, the WRF-ARW outputs are processed to generate ARL files with a 12-hour temporal span. These files are designed to overlap every 12 hours, ensuring continuous hourly meteorological data coverage. This approach provides a seamless dataset necessary for accurate and uninterrupted backward trajectory and forward simulation calculations. Forward simulations were performed using Hysplit with a standard configuration over a minimum total calculation time of 48h and an hourly time resolution. The plume altitude was taken from the IGN/AEMET data and Milford et al. (2023) for each studied date.

## Appendix D

Appendix D presents the comparison between the $\Delta XCO/\Delta XCO_2$ measured at FUE using the direct-sun FTIR measurements and the area covered daily by the lava flows, derived from the daily Copernicus EMSR546

mapping. A good agreement is found between the two dataset, indicating a possible contribution of the burning
infrastructure and vegetation in our FUE FTIR measurements. It contrasted with the ratios reported by Asensio-
Ramos et al. (2025) derived from open-path measurements performed at the N-NW from the eruptive fissure,
less affected by this contribution.

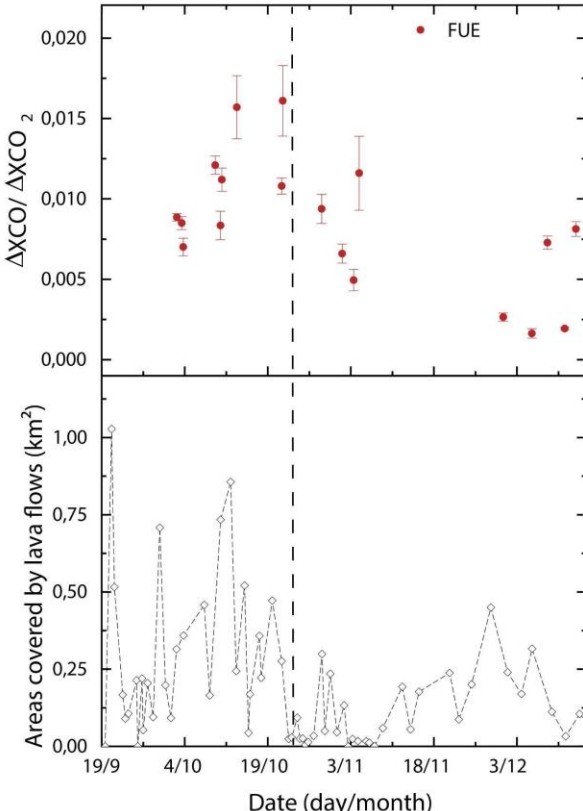


**Figure D1: Comparison between the time series of $\Delta XCO/\Delta XCO_2$ ratios obtained over the whole eruptive period at**
**FUE and the daily covered area by the lava flows derived from the daily Copernicus EMSR546 mapping**
**(COPERNICUS EMERGENCY MANAGEMENT SERVICE | Copernicus EMS - Mapping).**

 **Appendix E**

Appendix E describes the relationship observed between the retrieved daily $SO_2$ emission fluxes and the lava
Time Averaged Discharge Rate (TADR). The TADR estimates the lava volume responsible for the radiant flux
measured by satellite (Coppola et al., 2016). We exploit the fact that this volume is also the source of $SO_2$
emissions (Fig. E1), providing a direct quantification of the amount of S actually degassing ("effective S
degassing"), which is usually indirectly derived a posteriori, by the difference between the S content of the
primitive magma (melt inclusions) and that remaining in the matrix of the (degassed) eruptive products. This
quantity is shown constant through the eruption (E2).

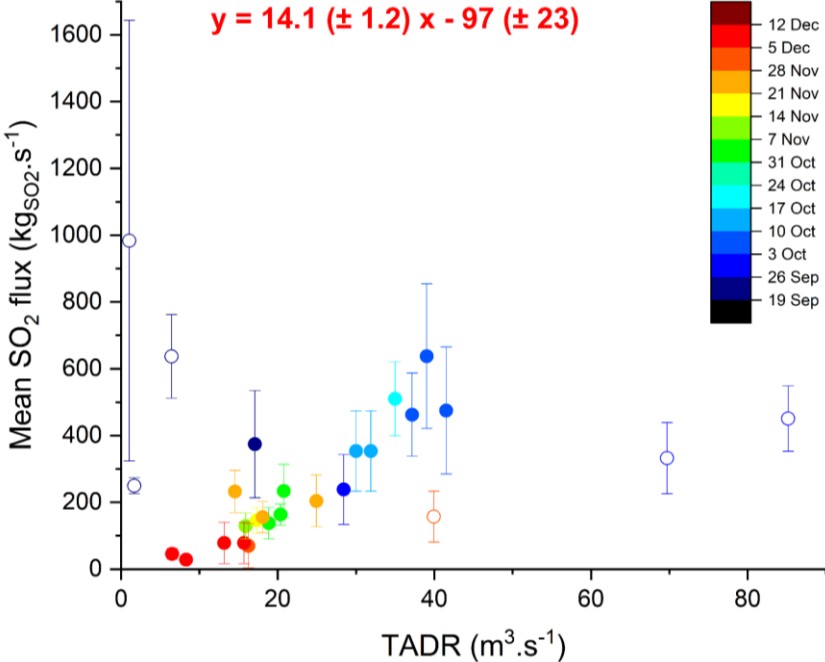

**Figure E1: Correlation between the TADR and $SO_2$ volcanic emission fluxes illustrating an average "effective S**
**degassing" of 14.1 ± 1.2 $kg_{SO2}$ per (thermal) cubic meter of lava discharged to the surface. Hollow points correspond**
**to the outliers to the dataset.**

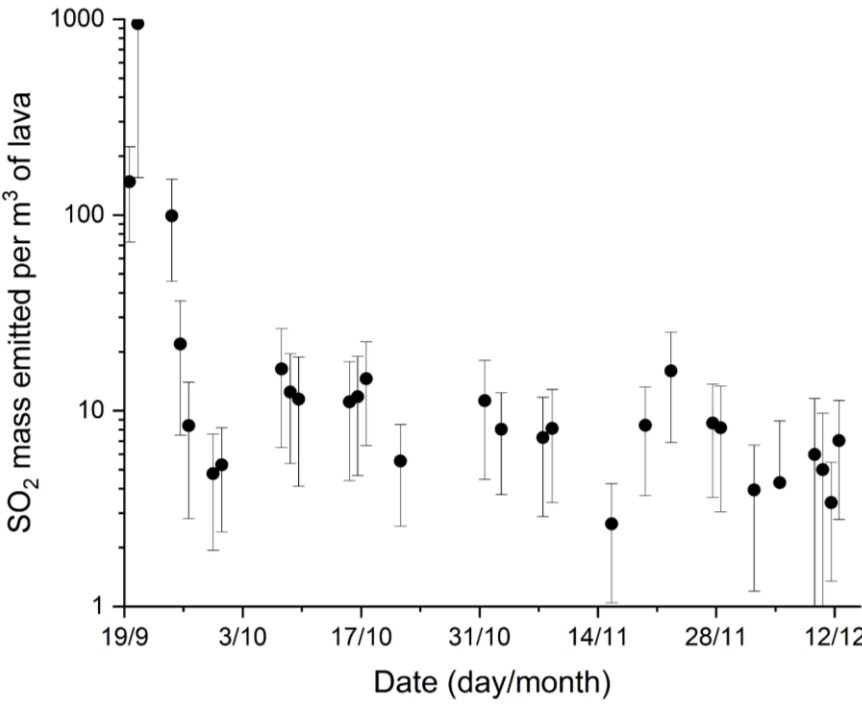

**Figure E2: Time series of "effective S degassing".**

## 8. Data availability

FTIR data used in this study are available upon request. In situ surface data at Izaña Atmospheric Observatory contribute to the WMO-GAW Program and are available at the World Data Centre for Greenhouse Gases (WDCGG, https://gaw.kishou.go.jp/). TROPOMI data (Copernicus Sentinel-5P) are publicly available from Sentinel-5P data hub at https://sentinels.copernicus.eu/web/sentinel/data-products/-/asset_publisher/fp37fc19FN8F/. Petrological dataset is available as Supplementary Material.

## 9. Authors contribution

All of the co-authors contributed to the preparation and writing of the manuscript. OG, TB, NT conceptualised the study. OG led the development of the FTIR program at the Izaña Atmospheric Observatory and its long-term operation. OG, RR, AA, VC were in charge of the implementation and operation of the Fuencaliente (La Palma) station during the eruption. They also assured the operation and maintenance of measurements at IZO. OG, NT, WS, ES, SL contributed to the FTIR and DOAS and data analysis. SL and PRS are responsible for the GAW surface measurements at IZO and their processing. NT, TB, RC, WS, OG contributed to the implementation and operation of the combined DOAS-EM27/SUN measurements at Fuencaliente. NT, TB, RC performed the MultiGAS measurements and ash sampling. RC processed the MultiGAS data. DGG, AK performed the SEM and EPMA analyses and helped for the interpretation and discussion of results. CA, MIG contribute to realisation of the Hysplit modeling of the volcanic plume dispersion to estimate the plume age. SR, JLD performed the chemical analysis of the $PM_{10}$ and contributed to their interpretation. MIG, SR, PGS, TB, NT contribute to the discussion about the $PM_{10}$ measurements. FH helped with the FTIR operating maintenance and data processing. He developed the PROFFAST and PROFFIT retrieval codes and provides continuous support to the group with respect to its use and spectrometer operation. FH and OG led the German–Spanish collaboration and provided precious help with respect to the EM27/SUN measurements within the framework of the COCCON network.

## 10. Acknowledgments

We acknowledge the two reviewers for their constructive comments which contribute to significantly improving the manuscript. The AEMET team and TB acknowledge the San Antonio volcano visitor's center in Fuencaliente and its personnel for authorizing and facilitating the instrumentation deployment there. The AEMET team also would like to thank all the researchers and technical personnel for the maintenance and operation of the instrumentation at the Izaña Atmospheric Observatory. The CSIC team and RC acknowledge the administration of the IPNA-CSIC and the CSIC deployment plan during the eruption and its coordination by Manuel Nogales. NT is grateful to the IPNA-CSIC and its director, Juan Ignacio Padron Peña, for allowing her temporary research internship during the eruption. TB, NT, RC, DGG and AK are grateful to PEVOLCA for granting permission for access to the exclusion zone during sampling and to F.M. Medina from the Cabildo Insular de La Palma for the sampling authorizations and for facilitating the fieldwork campaigns. TB and NT acknowledge Pablo González (Group of Volcanology of IPNA-CSIC) and A. Barreto (Aerosols group from the CIAI-AEMET) for the fruitful discussions and M. Charco (IGEO-CSIC, Madrid) for providing the corrected lava emission volumes data. Authors are grateful to S. Valade for the $SO_2$ masses from Mounts Project. NT, RC, and TB acknowledge C. Fayt, M. Van Roozendael, and A. Merlaud from the BIRA-IASB institute for providing and helping with the use of QDOAS

software. The CSIC team and R.C. acknowledge the Cabildo Insular de La Palma and its personnel for their assistance in the field. They also warmly thank J.G. Barreto from Spar La Palma and TICOM solutions and his personnel for their logistic support and assistance in the field. DGG acknowledges X. Arroyo for support at the UCM SEM laboratory, Madrid.

**11. Financial Support**

This study was partially funded by the European Union – NextGenerationEU within the actions P02.C05.I03.P51.S000.42 and P02.C05.I03.P51.S000.43. This study is part of the projects AERO-EXTREME (PID2021-125669NB-I00), funded by the Spanish National Research Agency (Agencia Estatal de Investigación) and the European Regional Development Funds. This study has received funding from the European Union's Horizon Europe Research and Innovation program under Grant Agreement 101189654. DGG acknowledges financial support from the Alexander von Humboldt Foundation through a Humboldt Fellowship for Postdoctoral Researchers. This work benefited from funding from CSIC and the Ministry of Science and Innovation through the CSIC-PIE project PIE20223PAL009. The aerosol sampling in La Palma during the eruption and part of the chemical analysis were performed within the framework of the project CSIC-LAPALMA-06, funded by CSIC and the Ministry of Science and Innovation.

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
