# Peer review of "New insights into 2021 La Palma eruption degassing processes from direct-sun spectroscopic measurements"

_EGUsphere, 2025_

## Author Comment (AC1)

**Response to Referees**

*Atmospheric Chemistry and Physics*

*Taquet et al.: "New insights into the 2021 La Palma eruption degassing processes from direct-sun spectroscopic measurements"*

We thank both reviewers for their very constructive comments, which really helped to prepare an improved revised manuscript.

**Response to Reviewer #2: Nicole Bobrowski**

The work of N. Taquet et al., "New insights into the 2021 La Palma eruption degassing processes from direct-sun spectroscopic measurements" present remote sensing and in-situ measurements over the entire period of the La Palma eruption, which took place in autumn 2021. Without doubt this is a wonderful data set, my congratulation – it is really an exceptional degassing data set of the La Palma eruption and will for sure be useful also to others for instance for more in depth modelling of this eruption. It also shows the strength and opportunities of such standardised measurement network instruments (like NDACC) outside their daily business. The authors give a large detailed description of the methods and show the potential of continuously working instruments even in far distances of volcanic emission sources and the strength of combining different technologies.

However, it is an atmospheric chemistry and physics journal so I would have liked to see a more extensive discussion about the impact of that volcanic emission on the local and regional (or even global) scale. You nicely determined the masses etc. but only compare them very shortly by mentioning few other sources, but what effect, impact have those gases measured here on the atmosphere, what changes on the chemistry or physic might they cause, what are the implications ... some back on the envelop calculation or at least consideration on the impact on pH in aerosol, clouds, on lifetimes of other atmospheric species, or if you assume the try oxidation for SO2 to sulphate can you estimate that amount of OH necessary from your data? … you even have particle and sulphate measurement please use those and please extend a bit more on the data interpretation. There is a much longer discussion interpretation of the data in a volcanological context than in an atmospheric one, which gives the impression that another journal would have been more adapted, or let's say a different community of readers could profit more of it (?) However, for some reason you decided for ACP, great journal and yes also your data fit, it is just the discussion and interpretation which remains a bit poor in the context of this journal topics.

**Reply:** We thank the reviewer for raising relevant concerns regarding the journal's scope, as our study lies at the interface between volcanology and atmospheric sciences. Nevertheless, our choice to submit our manuscript to ACP was justified by our interest to highlight the potential of standardized atmospheric monitoring networks (such as GAW, NDACC and COCCON) to contribute valuable data during exceptional events like volcanic eruptions. This dataset is of direct relevance to the ACP readership interested in quantifyinging emissions from point sources using FTIR spectroscopy, in quantifying the contribution of natural sources of atmospheric trace gases and aerosols, and assessing their temporal variability. We provide multi-species and time-resolved time series of volcanic gas emissions derived from surface (GAW) and ground-based FTIR (COCCON and NDACC) measurements  and satellite observations during the Tajogaite eruption. Some additional aspects relative to in-plume reaction and aerosols formation are also reported. The reviewer's suggestion encouraged us to extend the discussion on atmospheric implications, which has significantly improved the overall quality and depth of the study. We have specifically added section 5.4 in the discussion, entitled "Potential atmospheric implications of Tajogaite eruption emissions", which examines the relative contribution of Tajogaite's emissions compared to anthropogenic sources, and also explores their potential link to tropospheric $O_3$ levels.

As suggested by the reviewer, based on our observational dataset and an estimated maximum $SO_2$-to-$SO_4$ conversion rate at IZO, we performed a rough calculation of the total sulfate ($SO_4$) produced during the eruption. Assuming a 20% conversion efficiency of the total emitted S (maximum conversion rate based on the IZO dataset), this corresponds to a minimum of 0.5 Mt of sulfate released into the atmosphere

(information included in the section 5.4). Achieving such a level of oxidation would require around 95 kt of OH radicals. To assess the plausibility of this conversion under atmospheric conditions, we can roughly estimate the OH availability for one day within an atmospheric volume corresponding to the volcanic plume. Considering an average $SO_2$ flux of 21,000 t/day and using the TROPOMI images and radiosonde data to approximate the plume volume between La Palma and Tenerife, found to be ~21,000 $km^3$ (150 km × 1 km × 140 km). Even under high-end scenarios of atmospheric OH concentrations ($1\times10^6$ molecules $cm^{-3}$), the number of OH radicals present in this volume falls several orders of magnitude short of the amount required to oxidize 20% of the emitted $SO_2$. The $SO_2$-to-sulfate conversion rate was derived from IZO measurements and likely reflects not only in-plume oxidation processes but also early-stage transformations occurring upstream, including near-vent chemistry (Surl et al., 2021; Van Glasow et al., 2009). In addition, background or long-range transported sulfate may contribute to the observed $SO_4$ burden, leading to a possible overestimation of the conversion rate at IZO. Although these estimates are very rough and neglect many in-plume chemical processes, they nevertheless suggest that, regardless of the exact origin of the sulfate measured at IZO, a conversion rate of this magnitude implies substantial OH consumption within the plume and the implication of other pathways to overcome the apparent 15% S conversion threshold. Such OH depletion could transiently reduce the local oxidative capacity of the atmosphere and disrupt the chemical balance, affecting the oxidation of other trace gases, while halogen chemistry catalyzes significant local tropospheric ozone destruction within the plume. We analyzed other components such as $CH_4$ when the volcanic plume was detected by our FTIR measurements but did not observe clear impacts. We chose not to include these estimations in the manuscript due to the lack of sufficient data to further investigate these potential in-plume reactions.

However, we investigated the potential impacts of volcanic emissions on local tropospheric ozone. Volcanic chlorine compounds are known to trigger catalytic ozone destruction under conditions of sunlight and high humidity. Although our FTIR total column $O_3$ measurements did not show clear evidence of ozone depletion during the eruption, we further analyzed ozonesonde profiles from the electrochemical concentration cell (ECC) sonde measurements performed by AEMET in Tenerife, and launched from Puerto de La Cruz, ~140 km from the volcano (Figure R6). We focused our analysis on the period from early September to late November 2021, during which the FTIR gas time series are the densest. These profiles revealed two distinct ozone depletion events at altitudes corresponding to the volcanic plume (Figure R6). Interestingly, these depletion events coincided with sharp increases in cumulative halogen emissions (HCl and HF), while no corresponding peak was observed in $SO_2$ fluxes (Figure 7 in the new version of the manuscript). While the distance between the volcano and the observation site limits definitive attribution, the temporal coincidence between ozone depletion and halogen emission peaks suggests a possible halogen-related impact on tropospheric ozone during these episodes. These results are now incorporated into the revised discussion (see Figure 7 and section 5.4 of the new version of the manuscript). This additional observation underscores the value of combining emission gas and aerosol data with in situ and remote sensing atmospheric observations to assess the chemical effects of volcanic plumes, even during moderate events such as the Tajogaite eruption.

[Figure]

*Figure R6: $O_3$ profiles from 17/09 to 23/11/2021 measured by electrochemical concentration cells (ECC) launched from Puerto de la Cruz, in Tenerife.*

There are several smaller comments I just added in the pdf, including typos, small changes or suggestions and few questions to make the text clearer for all readers. Please take those into account by revising your manuscript.

**Reply:** We thank the reviewer for all the detailed suggestions provided in the PDF file, which have been incorporated into the manuscript.

Some question, arguments, which are in my opinion a little more important, I like to point out here and would like to see answers on it before publishing. The suggestion and questions are in the order they appear in the manuscript.

**Reply:** We provided below detailed point-by-point responses (in red) to all reviewer comments and suggestions.

Figure 1 should be extended also viewing the various measurement locations at La Palma which are mentioned in the text and which are part of discussions about the comparability, etc. so maybe realised by another inset. Also in Figure 1, the wind velocities might be displayed always with the same colour for the same height. (page 3)

Done: We modified Figure 1 with a new subset presenting a zoom of La Palma island, including the different measurement locations (FTIR and aerosols stations and multigas measurement locations). We also modified the colour scale of the wind rose diagram as suggested by the reviewer.

Instrumental descriptions the FTS instruments are nicely referenced and specified, the instrument used for the UV spectroscopy ("the DOAS") a little less – could you please add which fibre you used (400 micrometres, mono?) and what about the slit wide? Regarding the software could you add a reference or specify a bit, I made some suggestion in the manuscript, but certainly I'm not 100% sure (page 5, line 203ff)

We thank the reviewer for pointing out these missing elements and have added the corresponding information to the manuscript (l. 206-208 and 221-223):

"The DOAS instrument had a 50 µm wide slit entrance, and allows recording spectra in the 270-425 nm spectral range with a spectral resolution of 0.4 nm. We used a 200 µm wide quartz-made optical fibre."

*"DOAS direct-sun absorption spectra were routinely recorded using the MobileDOAS software (unpublished acquisition program developed for mobile DOAS measurements by C. Fayt and A. Merlaud from the BIRA-IASB institute) with an integration time of about 30 seconds, with on average 20 scans."*

.

For your MultiGAS measurements could you be a bit more specific how large was the time lag your determined between your sensors, and what about your smoothing parameters you mentioned. (line 255 ff)

The time lag was adjusted to maximize the coefficient of determination ($R^2$) between the two gas concentration time series, with typical values ranging from 5 to 9 seconds. This information has been added to the manuscript (lines 292-294): *"Time series of concentrations of the different gas species were cross-correlated by adjusting the time-lag (usually between 5 and 9 seconds) and smoothing parameter until the best R-squared correlation coefficient was obtained. The measurements presented here have R-squared higher than 0.75."*

Regarding the background correction for CO2 and CO – you show some nice examples for CO2 but none for CO – could you please add one in the supplement A, Figure A3

We added the figure A4 with examples of CO.

The analysis details of the DOAS instruments could be added to table 2 (ground based and satellite)

Done: We added the information in Table 2.

Could you please add some details for the satellite based SO2 fluxes – e.g. the distance you chose for putting the traverse or did you use multiple distances from the source and then calculating a mean? where did you get the plume height from? (line 372 ff)

The $SO_2$ flux presented in the study represents the average of several traverses (usually a few tens and, in some occasions, up to two hundreds, depending on the coherence of the plume and the wind field that transports it). The plume altitudes were derived from visual observation, photographs, distal webcam images (from Roque de los Muchachos) and HYSPLIT trajectory simulations, picking the injection altitude that best reproduces the general plume direction observed on the Tropomi image, and were found consistent with the AEMET/IGN estimates for coincident dates.

We added the information in the manuscript (l.276-281): *"The plume altitude was estimated from visual observations such as photographs, distal webcam images (from Roque de los Muchachos) and HYSPLIT trajectory simulations, picking the injection altitude that best reproduces the general plume direction observed on the TROPOMI image, and confirmed with the AEMET/IGN estimates for the coincident days. The SO2 fluxes were finally estimated using the average of several traverses (usually a few tens and, in some occasions, up to two hundreds, depending on the coherence of the plume and the wind field that transports it)."*

Figure 3 (page 11), the ratio HCl/CO2 should be probable circles instead of rectangles?

Done: We replace the rectangles with circles.

Figure 4 (page 12), please add here also the literature data to the plot you mention in the text and adapt the style of the Figure consistent with Figure 3. Please also confirm the background corrected ratios for CO and CO2?

We have adapted Figure 4 by adding literature data as shaded areas, also in accordance with Reviewer 1's comments. We confirm that all $CO/CO_2$ ratios presented here are background-corrected. To clarify this throughout the manuscript, we have standardized the notation by systematically using $\Delta CO/\Delta CO_2$ (or other background-corrected species) and replaced the previous notation accordingly.

You correlate TADR and SO2 by excluding more than 1/3 of the data, what was the criteria to exclude data – if I overlooked it, sorry but I haven't seen an explanation for the choice of which data to include and which to leave out rather than randomly taken into account just data that fit?

We thank the reviewer for this comment and acknowledge that our procedure was not clear enough. We revised this section and found that the count of the total number of pairs was considering 5 TADR values for which no coincident $SO_2$ flux data was disponible. Figure E1 presents all the revised data pairs. All the data pairs taken after 07/10 (full circles) are well correlated (21/27 data corresponding to 77% with Pearson Correlation coefficient=0.94). The observed outliers (empty circles) correspond to:

(i) three out of the first five days of the eruption (until 24/09). This period corresponds to the main $SO_2$ emission peak .

(ii) 29/09 and 30/09, corresponding to the first two data after the eruption break (27/09), after which the TADR peak occurred.

(iii) 02/12, corresponding following the opening of the Northern vents occurring on the 28/11/2021 and generating new spread lava flows in not previously affected areas.

We have rephrased this section in the manuscript (lines 720–728) to provide a clearer description.

*"The few outliers to this correlation (empty circles, Fig. E2) occurred during three distinct periods: (1) the initial days of the eruption, coinciding with the peak in $SO_2$ emissions (2) just after the 27/09 eruptive pause, at the onset of sharp increase in effusion rates and (3) following the opening of the late November vents, north of the main vent alignment. These outliers correspond to abrupt changes in the output rate, likely associated with transient perturbations of the surface thermal structure-conditions known to affect the reliability of TADR estimations based on radiant density models (Coppola et al., 2016). Interestingly, applying the TADR values derived from the Pleiades-based volume estimates of Belart and Pinel (2022), which are averaged over 6-7 days, would bring at least three of these outliers back in line with the main trend. This suggests that apparent short-term imbalances between $SO_2$ emissions and effusion rates may be rapidly compensated, resulting in a coherent degassing-effusion relationship over multi-day timescales. This is particularly evident at the beginning of the eruption, where the Belart and Pinel (2022) estimates yield significantly higher TADR values than those of Plank et al. (2023) (Fig. E2)."*

CO2 emission is constant during the eruption and you also write that it is independent from volcanic emissions (you even say in your summary and conclusion: "CO2 emissions are decoupled from the volcanic emissions" (which confused me a bit) so do we have also before and after the eruption such high CO2 emissions? If yes, please report that, if not please explain better why we have them when they are independent from volcanic emissions

We thank the reviewer for pointing that our formulation was not clear and confusing. Pre-eruptive $CO_2$ fluxes were ~250 times lower than the eruptive fluxes (Padron et al., 2015). The observed high $CO_2$ emissions indeed fueled the eruption but outweighed the expected emissions from the erupted lava volume alone. This indicates that $CO_2$ was emitted from a magma reservoir situated above its exsolution threshold.

We have rephrased this part of the manuscript (l.974-979):

*"Furthermore, while the $SO_2$ and halogen halides emission fluxes decreased throughout the eruption along with the lava emission fluxes, the $CO_2$ emission fluxes were found to be almost constant, implying a comparatively increasing discharge with respect to the daily emitted lava volumes. This is consistent with a significant amount of $CO_2$ being already exsolved in the reservoir, as previously observed by Burton et al. (2023), Dayton et al. (2023) and Dayton et al. (2024)."*

The total emission estimates are done by a Monte Carlo approach – if you would simply integrate the surface under the curve over the time for each molecule how large would be the difference to your current results. Wouldn't this be much more straight forward? What is the advantage of your method? (line 521)

Table 4 reports the total emissions using two different approaches: (1) the integration below the exponential curve, (2) the integration of the surface under the daily measured fluxes curve (trapezoidal rule). In both cases, we used the Monte Carlo approach to take into account the uncertainties (measurement errors for daily fluxes and fit parameter uncertainties for the analytical model). The Monte Carlo approach allows us to rigorously propagate these uncertainties by generating many realizations of the input data: sampling the fit parameters a and b in method (1) or random draws of daily fluxes in method (2). Therefore, although the Monte Carlo simulation does not generate important changes in the average flux estimates (relative difference <0.02% for $SO_2$), it allows estimating their uncertainty better.

It is true that there are not that many CO measurements of volcanic emission, but it is not true that only Wardell et al presented a CO measurement during an eruption. I mean those authors measured at Erebus which has a continuous lava lake, but also the measurements at Nyiragongo or Erta Ale were taken during

Lava lake measurements, or also the more recent measurements at Iceland from Scott et al were taken during an eruption. Please correct. (line 559 ff)

We thank the reviewer for pointing out this additional literature and we corrected the sentence in the manuscript (l. 57-600), including more references. Note that CO emissions are not mentioned in Scott et al. (2022).

*"Only few volcanic CO emissions are reported in the literature, such as 0.15 kt/day at Erebus volcano (Wardell et al., 2004), 0.007 kt/day at Oldoinyo Lengai (Oppenheimer et al., 2002), 0.16 to 0.27 kt/day at Nyiragongo volcano (Sawyer et al., 2008a), 0.0007 kt/day at Erta Ale (Sawyer at al., 2008b) and are about one order of magnitude lower than our estimates during the Tajogaite eruption."*

Figure 6B please add units on the axis's, interesting that there is a higher conversion at the beginning of the eruption – can you explain that? (page 17)

We thank the reviewer for pointing out the missing units and for the insightful remarks. We have added the missing units in Figure 6B. Concerning the temporal evolution of the Sp/(Sp+Sg) ratios, several hypotheses can be considered. Figure R8 illustrates the time series of $SO_4$ and $SO_2$ concentrations, the Sp/(Sp+Sg) ratio, as well as the altitudes of the volcanic plume and the Trade Wind Inversion (TWI), based on data from Milford et al. (2023). The variation of the Sp/(Sp+Sg) ratio over time appears to be closely related to the plume's altitude relative to the TWI. As discussed by Milford et al., the increase in surface-level $SO_2$ concentrations observed during the second part of the eruption (from early November onwards) is likely due to the plume being more frequently confined below the TWI. In such conditions, the measured plume at ground level likely consists of a mixture of explosive and effusive vent emissions retained beneath the inversion layer. In contrast, during the first phase of the eruption, the explosive plumes often rose above the TWI, and the plume detected at the surface at La Palma was probably dominated by effusive vent and diffuse emissions, potentially mixed with older, more oxidized plume remnants.

We added this possible explanation in the manuscript (l. 669-679).

*"Furthermore, the time distribution of the S(p)/(S(p)+S(g)) ratio (Fig. 6) suggests a higher conversion rate of $SO_2$ to sulfate during the first part of the eruption (until the beginning of November) compared to the second period. This trend appears closely tied to the volcanic plume's altitude relative to the Trade Wind Inversion (TWI), as described by Milford et al. (2023). During the first period of the eruption (until early November), plumes from explosive activity and fountaining vents often rose above the TWI, and surface measurements at La Palma likely captured older, dilute, more oxidized emissions from effusive vents trapped into the TWI. Conversely, from the beginning of November, the entire plume, comprising both explosive and effusive components, was more frequently trapped below the TWI, leading to the detection of younger, more concentrated, and less oxidized emissions at ground level. In any case, the plumes reaching IZO are most likely dominated by explosive emissions which, despite substantial transport times, exhibit oxidation rates below 15%."*

[Figure]

*Figure R8: Relationship between the SO$_2$-to-SO$_4$ conversion rate and the relative altitude of the plume with respect to the Trade Wind Inversion (TWI) during the eruption (TWI and plume altitude data from Milford et al., 2023).*

Why do you think your first estimate of 1,93±0,21 Mt of SO2 degassed during the eruption is in disagreement with the Tropomi estimate of 1,81±0,18 Mt SO2 – I would say it agrees perfectly within the given errors. (line 747 ff)

We thank the reviewer for pointing out this unclear sentence and we rephrased it (l. 816-817) : "This is compatible with the TROPOMI-derived total SO$_2$ emissions (1.81 ± 0.18 Mt)."

It is not really clear to me if your emitted CO2 volume fits roughly to your estimated pressure decrease, lava volume, etc? The conclusion of 5.3.2 is not clear to me, please could you rephrase and make it more clear.

We thank the reviewer for pointing out this confusing part of the manuscript and rephrased the section 5.3.2:

*"Applying the same MonteCarlo approach used for sulfur and assuming full CO$_2$ degassing, we estimate that ~4.4 ± 0.8 Mt of CO$_2$ would have been released from the erupted material alone. This is consistent with the estimate of 5.4 ± 1.0 Mt by Dayton et al. (2024). However, plume measurements indicate significantly higher total CO$_2$ emissions during the eruption, amounting to 19.4 ± 1.8 Mt. This discrepancy, combined with the near-constant fluxes throughout the eruption, supports the presence of a CO$_2$-rich fluid phase in the reservoir (Hansteen et al., 1998; Burton et al., 2023) coexisting with a CO$_2$-saturated melt, capable of contributing an additional 15 Mt of CO$_2$. Based on FI densities reported by Dayton et al. (2023), we estimate that this additional 15 Mt of CO$_2$ corresponds to a fluid volume of ~25-17 Mm$^3$ at the pressure of the shallow (deflating) reservoir pressure and at that of the deeper reservoir, respectively.*

*The ~1% pressure drop relative to the pressure at the beginning of the eruption observed by Charco et al. (2024) provides an opportunity to derive a first-order constraint on the volume of the deflating reservoir. Assuming this pressure loss is attributed to a volume change due to magma extraction, we can estimate the total volume of "hydraulically" connected magma/mush feeding the eruption.*

*To estimate the total volume (magma+fluid) extracted from the reservoir, 1) we corrected the eruptive products volume for vesicularity (Dense Rock Equivalent or DRE volume, taking as a reference a melt density of ~2700 kg.m-3; see previous section and Dayton et al., 2024), 2) we added the volume of the magma-filled dykes and sill network (as described by De Luca et al., 2022) and 3) finally, we corrected for the effect of magma compressibility. According to Rivalta and Segall (2008), the volume ratios (intrusion/associated reservoir deflation) necessary to estimate magma compressibility range between 1.2 and 7.7. For the Tajogaite eruption, the most likely value is ~5 (reservoir from 10 to 15 km deep, saturation depth >25 km; see Fig. 3 in Rivalta and Segall, 2008). Using such values for correcting our magma volume and adding our extracted (additional) fluid volume estimate allows estimating a total volume (magma + fluids) extracted from the reservoir of ~60 Mm3 (from 45 to 200 Mm$^3$ for the full range of volume ratios). Considering the extraction of this volume produced the pressure drop in the deflating reservoir, we roughly estimate the volume of magma/mush to equate, at least to 6 km$^3$ (4-20 km$^3$ range). This estimate provides a first-order volume of magma/mush that could have been "hydraulically" connected to the surface during the eruption. It includes at least the shallow reservoir, but may also encompass deeper zones of the plumbing system if they were effectively connected during the eruptive episode."*

3.3 you could not determine the F and Cl emission from the petrological data if I understood right? Because the difference of MI and Glass content is too small, could you please state if this is a general issue? Or are the values in you MI and glasses are particularly small? Thanks.

We thank the reviewer for this comment. We were actually able to estimate Cl emissions from the petrological data, but not those of F. This limitation is not due to particularly low concentrations in La Palma's magmas, which contain 551 ± 37 ppm Cl and 1559 ± 62 ppm F in melt inclusions, but rather to the very small difference between melt inclusions and matrix glasses (550 ± 37 ppm Cl and 1438 ± 47 ppm F), which falls within the range of analytical uncertainty. As noted by Dayton et al. (2024), "a HF and HCl budget cannot be calculated as fluorine and chlorine do not degas from the matrix glass in sufficient quantities to be resolvable from analytical noise."

This issue is partly general and partly specific to the La Palma eruption:

As discussed in Rose-Koga et al. (2021), uncertainties on halogens are generally high, especially F measurements by electron microprobe that can reach up to 20% relative. Although some of the published dataset was acquired using SIMS, which offers better precision, the difference between MI and glass remains too small to be resolved confidently, even with this technique.

In the case of La Palma, the inability to resolve halogen degassing petrologically is also linked to the rapid ascent of magma. Unlike $CO_2$, which begins to exsolve at greater depths and thus has time to degas significantly before fragmentation, halogens such as Cl and F have much higher solubilities and exsolve only at shallow depths, often very close to the fragmentation level. In fast-rising magmas, this means that halogens remain largely retained in the melt until quenching, resulting in glasses that preserve nearly their original halogen content. This behavior is not universal but can occur in other basaltic systems with similar ascent kinetics.

Therefore, while the small MI-glass difference is a general analytical challenge, the lack of resolvable halogen degassing in this case is also a consequence of the eruption's rapid dynamics.

We added some elements to the first paragraph of the halogens budget section to make this point clear (l.873-879):

"*Fluorine and chlorine generally have high solubility in magmas and only begin to exsolve at shallow depths, close to the fragmentation level (e.g.: Aiuppa, 2009). This is likely the case for the 2021 La Palma eruption, where rapid magma ascent (Romero et al., 2022; Boneschi et al., 2024) limited halogen degassing due to kinetic constraints. As a result, the melt retained most of its original halogen content, and the difference between melt inclusions and matrix glass Cl and F contents is hardly resolvable from analytical uncertainty (Dayton et al., 2024). We thus assessed the consistency of our fluxes using another approach, estimating the expected Cl and F degassed amounts from the total observed emissions.*"

Appendix D – Honestly this is very confusing appendix for me – I don't agree that your CO/CO2 data for the two station match very well and at the same time disagree with the one from Asensio-Ramos. Looking up the Figure D1 it seems that IZO and earlier data match better than IZO and FUE.

Please comment on this. Further on, please improve, discuss at all the data you are showing on the lower part of the same Figure in context with your data. Do you want to show that a mismatch might be larger areas are covered – more fires? Or what do you like to show here?

We thank the reviewer for highlighting this unclear point. We have added the in situ surface $\Delta CO/\Delta CO_2$ measurements at IZO in Figure 4, showing that the IZO in situ ratios are consistent with the FTIR ratios observed at FUE. As explained previously (see response to Reviewer 1 comments on Figure 4), the lower $\Delta CO/\Delta CO_2$ ratios retrieved by FTIR at IZO compared to the in situ surface values at the same site can be attributed to the contribution of different CO sources, which is consistent with the weak $\Delta CO$ vs. $SO_2$ correlation (r < 0.6). Satellite imagery confirms that, on days when a $\Delta CO$ vs. $\Delta CO_2$ correlation is observed in the FTIR data at IZO, the instrument's line of sight likely intersected both fresh and aged volcanic plumes. This overlap may influence the retrieved $\Delta CO/\Delta CO_2$ ratios due to a combination of geometric and compositional effects.

Nonetheless, our surface ratios at IZO and FTIR ratios at FUE in the presence of volcanic plume are significantly higher than those reported by Asensio-Ramos et al. (2025). As discussed in the manuscript, this discrepancy may stem from differences in measurement geometry and methodology. The plumes detected at FUE and IZO likely included a wildfire component, unlike those observed by Asensio-Ramos et al (2025). This hypothesis is further supported by the parallel between the $\Delta CO/\Delta CO_2$ ratio and the surface area affected by lava. The increase in lava coverage likely enhanced the extent of burnt vegetation, which coincides with the rise in the $\Delta CO/\Delta CO_2$ ratio measured at FUE (see Appendix D).

We have added several sentences to the manuscript to clarify and better support our hypothesis, and we also removed the IZO and literature data in the Appendix D to avoid any confusion.

l. 503-518: "*Figure 4 presents the time series of $\Delta CO/\Delta CO_2$ ratios derived from FTIR solar absorption measurements at the FUE and IZO stations throughout the eruption, alongside with in situ surface measurements at IZO (GAW data). The $\Delta CO/\Delta CO_2$ values observed at both sites and using both techniques are of the same order of magnitude, and exceed by more than one order of magnitude the average atmospheric background ratio at IZO (~0.0002). At FUE, the FTIR-derived ratios show a progressive increase from 0.0016 to 0.016 during the first 30 days of the eruption, followed by a decrease to lower values before mid-November. The surface $\Delta CO/\Delta CO_2$ ratios at IZO fall within a similar range to*

*those derived from FTIR at FUE, with some coinciding values in very good agreement. On average, the surface ratios at IZO are higher than the FTIR-derived ones at the same site. This discrepancy may be explained not only by the strong short-term variability in the $\Delta CO/\Delta CO_2$ ratios (only a few data points are coincident), but also by the fact that, although all these points coincide with the presence of $SO_2$ (indicating the presence of volcanic plume), the correlation between $\Delta CO$ and $SO_2$ is relatively weak ($R^2 < 0.6$), suggesting additional sources contributing to the CO enhancements. Furthermore, satellite imagery suggests that, on these days, the line of sight of the IZO FTIR instrument may have intersected aged volcanic plumes, potentially altering the retrieved $\Delta CO/\Delta CO_2$ ratios due to both geometric and compositional effects. The difference between the surface $\Delta CO/\Delta CO_2$ ratios observed at FUE and IZO and those (shaded area) reported by Asensio-Ramos et al. (2025) is discussed in Section 5.*"

l. 698-700: "*This hypothesis is also supported by the similarity of the $CO/CO_2$ time series at FUE with the time series of the areas covered daily by the advancing lava flows (Appendix D), reflecting the extent of burnt vegetation*"

**References:**

Surl, L., Roberts, T., and Bekki, S.: Observation and modelling of ozone-destructive halogen chemistry in a passively degassing volcanic plume, Atmos. Chem. Phys., 21, 12413–12441, https://doi.org/10.5194/acp-21-12413-2021, 2021.

Von Glasow, R., Bobrowski, N., and Kern, C.: The effects of volcanic eruptions on atmospheric chemistry, Chemical Geology, 263, 131–142, https://doi.org/10.1016/j.chemgeo.2008.08.020, 2009.

---

## Author Comment (AC2)

**Response to Referees**

*Atmospheric Chemistry and Physics*

*Taquet et al.: "New insights into the 2021 La Palma eruption degassing processes from surface and direct-sun spectroscopic measurements"*

We thank both reviewers for their very constructive comments, which really helped to prepare an improved revised manuscript.

**Response to Reviewer #1: Yves Moussallam**

**General comments**

There are a few things which I find amazing about this study. First, the authors are able to derive not only $SO_2$, HCl and HF from their solar-occultation FTIR measurements but also CO and $CO_2$. This is a major advance because unlike the other volcanic gas species listed, $CO_2$ has a very high (>400 ppm) background concentration, making the volcanic contribution over the large path length of solar occultation measurement (the entire atmosphere) too low to resolve prior to the latest generation of portable FTIR used here. Second the authors provide measurements of the gas compositions over the entire duration of the eruption which is a beautiful dataset. Third, the authors performed measurements at two sites, one close to the eruption on La Palma and one 140 km away on Tenerife.

I would encourage the authors to publish the code they used to analyse the FTIR spectra as they have made significant modifications in their retrieval strategy compared to the openly available code. I also encourage the authors to upload all the spectra they used on an open platform (this might be a journal requirement anyway).

**Reply:** The retrieval algorithms for EM27/SUN FTIR data are accessible via the KIT-COCCON website: https://www.imk-asf.kit.edu/english/COCCON.php. With respect to the specific retrievals used in this study, some aspects are still under consideration for a new specific contribution and will be made public after its publication. In any case, any modifications to the standard input files and all the datasets used in this study, can be obtained from the co-authors upon request.

I also encourage the authors to look for OCS in their FTIR spectra. Retrieving OCS is probably possible given the authors were able to retrieve volcanic CO. If you can retrieve OCS then you will have two redox species (e.g., https://www.nature.com/articles/s41561-018-0194-5) and may be able to tell a lot more about magmatic evolution during your observation period (see: https://comptes-rendus.academie-sciences.fr/geoscience/articles/10.5802/crgeos.158/).

**Reply:** We thank the reviewer for this insightful suggestion and the references. OCS can be retrieved in the mid-infrared spectral region (2030–2050 cm$^{-1}$), and in the context of this study, this range is only available using the high-resolution IFS125-HR measurements from the Izaña Atmospheric Observatory (Tenerife, 140 km from the Tajogaite volcano), which covers an extended MIR–NIR spectral range.

OCS is part of the standard NDACC products routinely retrieved from Izaña's IFS125-HR spectra (García et al., 2021), and we have examined the complete time series to investigate potential anomalies attributable to volcanic plumes. Over the course of the eruption, OCS strong enhancements (i.e., above natural variability) were observed on only two occasions. The first one (28/09/2021) coincided with elevated ΔCO and $SO_2$ levels (see Figure R1 below), and a ΔOCS/$SO_2$ ratio of 0.00015 was estimated (ΔOCS corresponds to the anomaly above the background). However, no Δ$CO_2$ anomaly was detected at that time (i.e. Δ$CO_2$ remained below the detection limit), preventing further investigation about the redox state of the magmatic system. During the second event, the OCS signal was too weak to establish any correlation with $SO_2$. For these reasons, we chose not to include these results in the manuscript.

[Figure]

*Figure R1: Correlation plot between ΔOCS, ΔCO$_2$, ΔSO$_2$ and ΔCO during the 28/09/2021 event during which the volcanic plume was detected at IZO.*

**Specific comments**

**Reply:**

Please add "e.g.," in front of all citations which are examples.

Done. We added "e.g.," in front of citations which are examples.

Line 51: "…*CO2 and H2O are among the deepest exsolved gas species, followed by SO2 and halogens in sub-surface*." This is not entirely/always true I suggest taking out this sentence.

We thank the reviewer for pointing out this unclear sentence and we replaced it with: "*CO$_2$ and H$_2$O are **usually** among the deepest exsolved gas species*" (l. 52 of the new version of the manuscript).

Line 59-64: Overly vague statements.

We rephrased these lines in the new version of the manuscript (l. 59-63) with:

*"Volcanic plume compositions, when combined with seismic and structural data, help constrain volatile fluxes, magma ascent rates, and the architecture of the magmatic plumbing system. Integrating gas measurements with petrological constraints from matrix, melt inclusions (MI), and fluid inclusions (FI) enables reconstruction of pre-eruptive volatile contents and degassing pathways, which are key to modeling eruption dynamics (e.g.: Ubide et al., 2023; Longpré et al., 2025)."*

Line 107: "…*Lava evolved from a'a' to fluid basaltic flows with changing composition*." You mean basanitic flows no?

We thank the reviewer for pointing out this unclear sentence and we replaced "basaltic" with "*basanitic*".

Line 170: "…*Base map was obtained from © Google Earth (©Google).*" Add the original sources of the satellite data. 0

We added the original source for the SO$_2$ satellite data in the legend for Figure 1: *"The base layer was sourced from Google Earth (© Google), while the SO$_2$ distribution map was derived from TROPOMI data accessed through the Sentinel Hub platform."*

Figure 2: A photograph of the real setup would be better here that the Schematic (or in addition to).

Done: We replaced the schematic with a photograph in Figure 2 corresponding to our measurement set-up (combined EM27/SUN - DOAS) at the FUE station.

Line 398: *"We also report Cl, F and S contents in tephra glasses that were measured during the analyses published in Gonzalez-Garcia et al. (2023)."* I don't understand this, if these are already published you are not reporting them here. Or where these analysed for major elements but not S, Cl, Fl ? Please explain.

The major elements and S, Cl and F were measured together, but only the major elements were published in Gonzalez-Garcia et al. (2023). We rephrased this sentence as follows (l. 303-305 of the new version):

*"We also report Cl, F, and S contents in tephra glasses that were measured alongside major elements during the analytical session described in Gonzalez-Garcia et al. (2023), although only the major element data were published in that study."*

Figure 3: Maybe add the data from Asensio-Ramos et al., (2025) too.

The figure R2 shows our data together with the HCl/SO$_2$ and $\Delta$CO/SO$_2$ ratios reported by Asensio-Ramos et al. (2025). As shown, including the full set of literature data might obscure the internal consistency of our dataset, which relies on only two, well-characterized measurement techniques (i.e., solar absorption FTIR measurements and in situ surface observations), both compatible with long term atmospheric monitoring networks. This could lead to misinterpretations. Therefore, in the figure 3 of the new version of the manuscript, we chose to represent the literature values only as shaded areas, in order to preserve the clarity of our time series.

[Figure]

*Figure R2: Comparison between the gas-species to $SO_2$ ratios found for this study and the literature data.*

As shown in Figure R2, our $\Delta CO_2/SO_2$ and $HCl/SO_2$ ratios fall within the range reported in the literature. The observed differences can be attributed to variations in measurement techniques, viewing geometries, and plume sampling locations, particularly for species like CO and $CO_2$, which exhibit substantial atmospheric background levels. Our measurements are based on solar absorption spectroscopy conducted at distances of up to 140 km from the volcanic source, but benefit from high signal-to-noise ratios and traceability through participation in international networks such as NDACC and COCCON. This enables robust background corrections, even at significant distances from the source. In comparison, the study by Asensio-Ramos et al. (2025) employs an Open Path approach, using lava as a source of infrared emission (emission–absorption spectra) over shorter optical paths. This technique involves both absorption and emission processes, and requires consideration of temperature gradients across the atmospheric layers. Differences in radiative transfer effects, methodological approach and retrieval strategies, and measurement geometries could easily account for the observed small discrepancies. The significantly higher $\Delta CO/CO_2$ and $\Delta CO/SO_2$ ratios observed in our study compared to those reported by Asensio-Ramos et al. (2025) likely stem from differences in CO enhancements, as our $\Delta CO/CO_2$ ratios (Figure 4 of the new version of the manuscript) are also substantially higher than theirs. As mentioned in the manuscript (l.697-704), this discrepancy can be explained by the location of their measurement sites, mostly NNW of the eruptive vent and upwind of biomass burning plumes, whereas our FUE and IZO sites were more exposed to CO emissions from vegetation and building fires triggered by advancing lava flows. Furthermore another possible explanation could be the different contribution of emissions from effusive vs. explosive vents in the optical path of the instrument. We added this alternative hypothesis in the manuscript (l.689-692): "*Tajogaite volcano presented notable differences in eruptive behaviour between the different vents along the volcanic fissure, the higher elevated ones being more explosive than the lower ones. Recent studies suggest that eruptive dynamics may affect the abundance of redox-sensitive species (e.g.: Oppenheimer et al. 2018, Moussalam et al. 2019).*"

Figure 3: You could have HF/SO2 and HF/CO2 plotted in their own panel to make the figure a bit clearer.

In our view, it is essential to display all the time series within a single figure to clearly highlight the co-variability of the ratios. Adding two additional panels to the figure would either exceed the page size limit or hinder the readability of the figure. Moreover, the $HF/SO_2$ and $HCl/SO_2$ ratios exhibit similar trends and variability, as do $HCl/CO_2$ and $HF/CO_2$. Therefore, we chose to group these ratios into two panels, offering a comprehensive overview of all species and their co-variations in a single plot. For these reasons, we prefer to retain the original version of Figure 3.

Figure 4 and CO/CO2: You say the data from the FUE and IZO observation cites are similar but it rather looks like the CO/CO2 ratios measured at IZO tend to be lower than the ones measured at FUE. This may be evidence of oxidation of the gas plume during transport.

We thank the reviewer for this very insightful comment. To investigate this further, we have added the in situ surface $\Delta CO/\Delta CO_2$ ratios measured at IZO (GAW measurements) in Figure 4 of the new version of the manuscript. These ratios fall within the same range as those observed at FUE and their similarity with the FUE FTIR ratios rules out the hypothesis of systematic oxidation during plume transport between IZO and FUE. This new data bring further constraint on the amplitude of the intraday and day-to-day variability of this ratio (up to 0.005 over 1 day, i.e. ~ ⅓ of the full observed variation range) at IZO. It is noteworthy that at IZO, the in situ surface $\Delta CO/\Delta CO_2$ ratios are systematically higher than those derived from FTIR measurements. The above-mentioned short-term variability could explain this difference (only 1/5 coincident measurements), but other considerations due to measurement geometries have to be taken into account. On the days when FTIR retrievals were possible (i.e., when the correlation coefficient exceeded 0.6), the correlations were generally weak and only marginally above the threshold, except on October 17. Despite the detection of $SO_2$ on these days, confirming the presence of volcanic plume in the line of sight of the instrument, the FTIR retrievals showed limited correlation with $SO_2$. In parallel, TROPOMI S5P imagery (available on the Mounts project website) indicates that the FTIR instruments at IZO may have been affected by aged plume components during those episodes. This suggests that the FTIR-derived $\Delta CO/\Delta CO_2$ ratios at Izaña likely reflect a mixture of fresh and aged plumes, while the in situ surface measurements are more indicative of more directly transported plumes.

We added the following lines in the manuscript (l. 503-518):

"*Figure 4 presents the time series of $\Delta CO/\Delta CO_2$ ratios derived from FTIR solar absorption measurements at the FUE and IZO stations throughout the eruption, alongside with in situ surface measurements at IZO (GAW data). The $\Delta CO/\Delta CO_2$ values observed at both sites and using both techniques are of the same order of magnitude, and exceed by more than one order of magnitude the average atmospheric background ratio at IZO (~0.0002). At FUE, the FTIR-derived ratios show a progressive increase from 0.0016 to 0.016 during the first 30 days of the eruption, followed by a decrease to lower values before mid-November. The surface $\Delta CO/\Delta CO_2$ ratios at IZO fall within a similar range to those derived from FTIR at FUE, with some coinciding values in very good agreement. On average, the surface ratios at IZO are higher than the FTIR-derived ones at the same site. This discrepancy may be explained not only by the strong short-term variability in the $\Delta CO/\Delta CO_2$ ratios (only a few data points are coincident), but also by the fact that, although all these points coincide with the presence of SO2 (indicating the presence of volcanic plume), the correlation between $\Delta CO$ and $SO_2$ is relatively weak ($R^2 < 0.6$), suggesting additional sources contributing to the CO enhancements. Furthermore, satellite imagery suggests that, on these days, the line of sight of the IZO FTIR instrument may have intersected aged volcanic plumes, potentially altering the retrieved $\Delta CO/\Delta CO_2$ ratios due to both geometric and compositional effects. The difference between the surface $\Delta CO/\Delta CO_2$ ratios observed at FUE and IZO and those (shaded area) reported by Asensio-Ramos et al. (2025) is discussed in Section 5.*"

The comparison between SO2 flux and TADR is interesting, I would suggest citing this article which found the same thing during the Fagradalsfjall eruption: https://www.sciencedirect.com/science/article/pii/S0377027324000568

The following sentence has been added at lines 715-716 to comment on this previous observation: *"A similar correlation between $SO_2$ emissions and effusive volumes has previously been observed during the 2021 Fagradalsfjall eruption (Pfeffer et al., 2024)."*

Line 745: The difference in S content of the glass may between your data and previous publications may be related to the type of sample (flow vs tephra) used in each study. Please specify if these are all from the same type of samples.

We thank the reviewer for this suggestion. All the analyses arise from the tephra matrix.

The most comprehensive matrix dataset published to date is that of Longpré et al. (2025) with >500 EMPA distributed over the eruption. All the individual analyses previously published and our new dataset is consistent with Longpré et al. (2025). Our analyses were mainly collected from samples of 23/09, during one of the most explosive phases of the eruption, which emitted the tephra with the highest matrix S contents (up to 800 ppm) of the eruption. Our average is thus consistently slightly higher than that of the average for full literature data, for which the vast majority of samples have 300-500 ppm.

In this sentence, our point was both comparing our new data with literature values and suggesting that the matrix value used by Dayton et al. (2024) for their S degassing calculation is probably a little low, resulting in higher emissions. We rephrased this section as follows (l.812-817):

*"Note that the matrix S contents we present (average 534 ppm; N=52; $\sigma$=130 ppm; Supplementary Table S1) are consistent with previously published datasets for the eruption (average of 403 ppm; N=438; $\sigma$=10 ppm; Burton et al., 2023; Longpré et al., 2025). These data are nevertheless substantially higher than the value reported by Dayton et al. (2024). Using these values in the MonteCarlo degassing simulation of Dayton et al. (2024), the full degassing of 0.25 km$^3$ of magma would produce emissions of 1.93 ± 0.21 Mt SO$_2$. This is compatible with the TROPOMI-derived total SO$_2$ emissions (1.81 ± 0.18 Mt)."*

Line 753: You say that sulfide droplets are "...*absent from the matrix.*" But then you show a picture of Sulfide droplets in the matrix glass (Figure B1). This is confusing please rephrase and improve the explanations. Do you see any sulfide inclusions in minerals? Can you plot the S content of Melt inclusions versus FeO to see when the melt reached sulfide saturation?

We thank the reviewer for spotting this confusing section. Rare sulfide droplets have indeed been observed in the matrix of Tajogaite eruption products by Day et al. (2022) and Pankhurst et al. (2022). We have only been able to observe sulfide bids in the matrix, but the recent paper of Andujar et al. (2025) describes "a few" sulfide inclusions in CPx cores and magnetites. Here we meant that the S making up these sulfides is neither in the matrix analyses nor degassed. We rephrased it and explained as follows (l. 818-832):

*"A possibly unaccounted repository for initial S in the degassing balance could be the rare sulfide droplets, previously described to be present in the eruptive products matrix (Fig. B1; Day et al., 2022; Pankhurst et al., 2022) but also, more recently in clinopyroxene (CPx) cores and in magnetites (Andujar et al., 2025). These droplets separated from the silicate melt upon reaching the sulfide saturation during a pre-eruptive crystallization episode (Day et al., 2022), as confirmed by our own saturation calculations using the ONeil (2021) SCSS model (see Appendix B2). Importantly for the sulfur budget, although part of the primitive magma S content, as recorded in MI, the sulfur they contain is not included in matrix glass analyses (since it is physically segregated) and is not released as gas during eruption. The sulfide abundance could range between 0.03 vol.% (QEMSCAN quantification in Pankhurst et al., 2022) and 0.066 vol.% (0.001 mass fraction in the crystallizing assemblage in the models of Day et al., 2022). Assuming a density of 4500 kg·m-3 (Saumur et al., 2015) and an average sulfur content of ~35% in the analyzed sulfides (Fig. B1), this range of abundance would represent a potential sulfide cargo in the erupted lava until day 20 (Day et al., 2022) of ~30 to 60 kt of non-degassed sulfur (equivalent to ~60 to 120 kt of SO$_2$). Accounting for this contribution would further improve the agreement between the petrologic budget (1.81-1.87 Mt of SO$_2$) and satellite-based estimates (1.81 ± 0.18 Mt of SO$_2$)."*

We present hereafter in Figure R3 the S vs FeO contents (corrected for post-entrapment crystallization) in published melt inclusions (hollow circles for Burton et al., 2023; grey dots for Dayton et al. 2024). Note that Dayton et al. (2024) state:

*"Our analyzed melt inclusions do not record evidence of sulfur saturation, identified by the presence of sulfide blebs (Hartley et al., 2017), except in two heterogeneously entrapped inclusions with anomalously*

*low $H_2O$ (~0.5 wt%) and high chlorine contents (~1,000 ppm)."* We remark the difference between both datasets, which we attribute to the PEC correction procedure (most probably $FeO^*_i$) used by the different authors.

[Figure]

*Figure R3: Correlation plot between S and FeO contents measured in the melt inclusions from literature data (Burton et al., 2023 and Dayton et al., 2024).*

The MgO content is less affected by this effect and its relationship with S shows a clear increase from 2500 to >3500 ppm between 8.5 and ~6.5 wt% MgO, then slightly decreasing down to a maximum S content of 3500 ppm at 4.5 wt% MgO, suggesting possible precipitation of sulfide, S being exsolved only in subsurface (Burton et al. 2023).

[Figure]

*Figure R4: S vs. MgO diagram for the published MI showing an increasing trend between 8.5 and ~6.5 wt.% MgO that stabilizes at lower MgO contents.*

To confirm this hypothesis, we further calculated the sulfur content at sulfide saturation using the PySulfSat tool (Wieser and Gleeson 2023) and added the following figure and description in the Appendix B.

[Figure]

*Figure R5: The upper panel shows the results of the sulfur content at sulfide saturation (SCSS) calculations performed using the model of ONeill (2021) implemented in the open-source Python3 tool PySulfSat (Wieser and Gleeson, 2023). The starting composition is one of the most primitive MI of the literature dataset for the eruption (LM0 G29 Dayton et al. 2024), to which a Petrolog3 (Danyushevsky and Plechov, 2011) crystallization model (with olivine±clinopyroxene + spinel as crystallizing phase, following Day et al. 2022) is applied at a magma stalling at 3.5 kbars and a $fO_2$ buffer of NNO+0.4, following Andujar et al. (2025). Given these conditions, the melt is expected to contain a significant proportion of sulfur as sulfate ($S^{6+}$), rather than sulfide ($S^{2-}$). Therefore, we used the SCSSt model of Jugo et al. (2010), which accounts for mixed sulfur speciation, to evaluate saturation. Only a few inclusions slightly exceed the SCSSt curve, consistent with the rarity of sulfide globules in the eruptive products and with the interpretation that sulfide saturation was only reached locally or after some crystallization (Day et al., 2022). The bottom panel shows the modeled composition (Fe/Fe+Ni+Cu) of the sulfide phase precipitating along the liquid line of descent, which is matching the measured compositions between ~4*

*and 5.8 wt% MgO (after 5-15% crystallization). This range is reported as the orange section of the liquid line of descent in the upper panel.*

Figure B2: Please specify the sample types (tephra vs flow).

We checked over again the original publications to be sure, but the full dataset represented in this figure arises from tephra. For clarity, we added a line in the caption of figure B3: *"Matrix glasses have been measured on tephra samples and are from this study, Burton et al. (2023), Ubide et al., 2023, Dayton et al. (2024), Longpré et al. (2025)."*

Figure D1: The data from Asensio-Ramos et al. (2025) could be added to your Figure 4 also.

Done: We added the data from Asensio-Ramos et al. (2025) as shaded areas to the Figure 4 of the new version of the manuscript, for consistency with Figure 3, and updated the legend accordingly. In addition, following Reviewer 2's comment, we removed the literature data from Figure D1 to clarify the message of the figure.